# Dynamics of Finite Width Kernel and Prediction Fluctuations in Mean Field Neural Networks

**Blake Bordelon  &  Cengiz Pehlevan**
John Paulson School of Engineering and Applied Sciences,
Center for Brain Science,
Kempner Institute for the Study of Natural & Artificial Intelligence,
Harvard University
Cambridge MA, 02138
`blake_bordelon@g.harvard.edu, cpehlevan@g.harvard.edu`

## Abstract

We analyze the dynamics of finite width effects in wide but finite feature learning neural networks. Starting from a dynamical mean field theory description of infinite width deep neural network kernel and prediction dynamics, we provide a characterization of the $\mathcal{O}(1/\sqrt{\text{width}})$ fluctuations of the DMFT order parameters over random initializations of the network weights. Our results, while perturbative in width, unlike prior analyses, are non-perturbative in the strength of feature learning. In the lazy limit of network training, all kernels are random but static in time and the prediction variance has a universal form. However, in the rich, feature learning regime, the fluctuations of the kernels and predictions are dynamically coupled with a variance that can be computed self-consistently. In two layer networks, we show how feature learning can dynamically reduce the variance of the final tangent kernel and final network predictions. We also show how initialization variance can slow down online learning in wide but finite networks. In deeper networks, kernel variance can dramatically accumulate through subsequent layers at large feature learning strengths, but feature learning continues to improve the signal-to-noise ratio of the feature kernels. In discrete time, we demonstrate that large learning rate phenomena such as edge of stability effects can be well captured by infinite width dynamics and that initialization variance can decrease dynamically. For CNNs trained on CIFAR-10, we empirically find significant corrections to both the bias and variance of network dynamics due to finite width.

## 1   Introduction

Learning dynamics of deep neural networks are challenging to analyze and understand theoretically, but recent progress has been made by studying the idealization of infinite-width networks. Two types of infinite-width limits have been especially fruitful. First, the kernel or lazy infinite-width limit, which arises in the standard or neural tangent kernel (NTK) parameterization, gives prediction dynamics which correspond to a linear model [1–5]. This limit is theoretically tractable but fails to capture adaptation of internal features in the neural network, which are thought to be crucial to the success of deep learning in practice. Alternatively, the mean field or $\mu$-parameterization allows feature learning at infinite width [6–9].

With a set of well-defined infinite-width limits, prior theoretical works have analyzed finite networks in the NTK parameterization perturbatively, revealing that finite width both enhances the amount of feature evolution (which is still small in this limit) but also introduces variance in the kernels and the predictions over random initializations [10–15]. Because of these competing effects, in some situations wider networks are better, and in others wider networks perform worse [16].

37th Conference on Neural Information Processing Systems (NeurIPS 2023).

In this paper, we analyze finite-width network learning dynamics in the mean field parameterization. In this parameterization, wide networks are empirically observed to outperform narrow networks [7, 17, 18]. Our results and framework provide a methodology for reasoning about detrimental finite-size effects in such feature-learning neural networks. We show that observable averages involving kernels and predictions obey a well-defined power series in inverse width even in rich training regimes. We generally observe that the leading finite-size corrections to both the bias and variance components of the square loss are increased for narrower networks, and diminish performance. Further, we show that richer networks are closer to their corresponding infinite-width mean field limit. For simple tasks and architectures the leading $\mathcal{O}(1/\text{width})$ corrections to the error can be descriptive, while for large sample size or more realistic tasks, higher order corrections appear to become relevant. Concretely, our contributions are listed below:

1. Starting from a dynamical mean field theory (DMFT) description of infinite-width nonlinear deep neural network training dynamics, we provide a complete recipe for computing fluctuation dynamics of DMFT order parameters over random network initializations during training. These include the variance of the training and test predictions and the $\mathcal{O}(1/\text{width})$ variance of feature and gradient kernels throughout training.

2. We first solve these equations for the lazy limit, where no feature learning occurs, recovering a simple differential equation which describes how prediction variance evolves during learning.

3. We solve for variance in the rich feature learning regime in two-layer networks and deep linear networks. We show richer nonlinear dynamics improve the signal-to-noise ratio (SNR) of kernels and predictions, leading to closer agreement with infinite-width mean field behavior.

4. We analyze in a two-layer model why larger training set sizes in the overparameterized regime enhance finite-width effects and how richer training can reduce this effect.

5. We show that large learning rate effects such as edge-of-stability [19–21] dynamics can be well captured by infinite width theory, with finite size variance accurately predicted by our theory.

6. We test our predictions in Convolutional Neural Networks (CNNs) trained on CIFAR-10 [22]. We observe that wider networks and richly trained networks have lower logit variance as predicted. However, the timescale of training dynamics is significantly altered by finite width even after ensembling. We argue that this is due to a detrimental correction to the mean dynamical NTK.

## 1.1 Related Works

Infinite-width networks at initialization converge to a Gaussian process with a covariance kernel that is computed with a layerwise recursion [23–26, 13]. In the large but finite width limit, these kernels do not concentrate at each layer, but rather propagate finite-size corrections forward through the network [27–30, 14]. During gradient-based training with the NTK parameterization, a hierarchy of differential equations have been utilized to compute small feature learning corrections to the kernel through training [10–13]. However the higher order tensors required to compute the theory are initialization dependent, and the theory breaks down for sufficiently rich feature learning dynamics. Various works on Bayesian deep networks have also considered fluctuations and perturbations in the kernels at finite width during inference [31, 32]. Other relevant work in this domain are [33–39].

An alternative to standard/NTK parameterization is the mean field or $\mu P$-limit where features evolve even at infinite width [6–9, 40–42]. Recent studies on two-layer mean field networks trained online with Gaussian data have revealed that finite networks have larger sensitivity to SGD noise [43, 44]. Here, we examine how finite-width neural networks are sensitive to initialization noise. Prior work has studied how the weight space distribution and predictions converge to mean field dynamics with a dynamical error $\mathcal{O}(1/\sqrt{\text{width}})$ [40, 45], however in the deep case this requires a probability distribution over couplings between adjacent layers. Our analysis, by contrast, focuses on a function and kernel space picture which decouples interactions between layers at infinite width. A starting point for our present analysis of finite-width effects was a previous set of studies [9, 46] which identified the DMFT action corresponding to randomly initialized deep NNs which generates the distribution over kernel and network prediction dynamics. These prior works discuss the possibility of using a finite-size perturbation series but crucially failed to recognize the role of the network prediction fluctuations on the kernel fluctuations which are necessary to close the self-consistent equations in the rich regime. Using the mean field action to calculate a perturbation expansion around DMFT is a long celebrated technique to obtain finite size corrections in physics [47–50] and has been utilized for random untrained recurrent networks [51, 52], and more recently to calculate variance of

feature kernels $\Phi^\ell$ at initialization $t = 0$ in deep MLPs or RNNs [53]. We extend these prior studies to the dynamics of training and to probe how feature learning alters finite size corrections.

## 2 Problem Setup

We consider wide neural networks where the number of neurons (or channels for a CNN) $N$ in each layer is large. For a multi-layer perceptron (MLP), the network is defined as a map from input $\boldsymbol{x}_\mu \in \mathbb{R}^D$ to hidden preactivations $\boldsymbol{h}_\mu^\ell \in \mathbb{R}^N$ in layers $\ell \in \{1, ..., L\}$ and finally output $f_\mu$

$$f_\mu = \frac{1}{\gamma N} \boldsymbol{w}^L \cdot \phi(\boldsymbol{h}_\mu^L), \quad \boldsymbol{h}_\mu^{\ell+1} = \frac{1}{\sqrt{N}} \boldsymbol{W}^\ell \phi(\boldsymbol{h}_\mu^\ell), \quad \boldsymbol{h}_\mu^1 = \frac{1}{\sqrt{D}} \boldsymbol{W}^0 \boldsymbol{x}_\mu, \tag{1}$$

where $\gamma$ is a scale factor that controls feature learning strength, with large $\gamma$ leading to rich feature learning dynamics and the limit of small $\gamma \to 0$ (or generally if $\gamma$ scales as $N^{-\alpha}$ for $\alpha > 0$ as $N \to \infty$, NTK parameterization corresponds to $\alpha = \frac{1}{2}$) gives lazy learning where no features are learned [4, 7, 9]. The parameters $\boldsymbol{\theta} = \{\boldsymbol{W}^0, \boldsymbol{W}^1, ..., \boldsymbol{w}^L\}$ are optimized with gradient descent or gradient flow $\frac{d}{dt} \boldsymbol{\theta} = -N\gamma^2 \nabla_{\boldsymbol{\theta}} \mathcal{L}$ where $\mathcal{L} = \mathbb{E}_{\boldsymbol{x}_\mu \in \mathcal{D}} \ell(f(\boldsymbol{x}_\mu, \boldsymbol{\theta}), y_\mu)$ is a loss computed over dataset $\mathcal{D} = \{(\boldsymbol{x}_1, y_1), (\boldsymbol{x}_2, y_2), \ldots (\boldsymbol{x}_P, y_P)\}$. This parameterization and learning rate scaling ensures that $\frac{d}{dt} f_\mu \sim \mathcal{O}_{N,\gamma}(1)$ and $\frac{d}{dt} \boldsymbol{h}_\mu^\ell = \mathcal{O}_{N,\gamma}(\gamma)$ at initialization. This is equivalent to maximal update parameterization ($\mu$P)[8], which can be easily extended to other architectures including neural networks with trainable bias parameters as well as convolutional, recurrent, and attention layers [8, 9].

## 3 Review of Dynamical Mean Field Theory

The infinite-width training dynamics of feature learning neural networks was described by a DMFT in [9, 46]. We first review the DMFT's key concepts, before extending it to get insight into finite-widths. To arrive at the DMFT, one first notes that the training dynamics of such networks can be rewritten in terms of a collection of dynamical variables (or *order parameters*) $\boldsymbol{q} = \text{Vec}\{f_\mu(t), \Phi_{\mu\nu}^\ell(t, s), G_{\mu\nu}^\ell(t, s), ...\}$ [9], which include feature and gradient kernels [9, 54]

$$\Phi_{\mu\nu}^\ell(t, s) \equiv \frac{1}{N} \phi(\boldsymbol{h}_\mu^\ell(t)) \cdot \phi(\boldsymbol{h}_\nu^\ell(s)), \quad G_{\mu\nu}^\ell(t, s) \equiv \frac{1}{N} \boldsymbol{g}_\mu^\ell(t) \cdot \boldsymbol{g}_\nu^\ell(s), \tag{2}$$

where $\boldsymbol{g}_\mu^\ell(t) = \gamma N \frac{\partial f_\mu(t)}{\partial \boldsymbol{h}_\mu^\ell(t)}$ are the back-propagated gradient signals. Further, for width-$N$ networks the distribution of these dynamical variables across weight initializations (from a Gaussian distribution $\boldsymbol{\theta} \sim \mathcal{N}(0, \mathbf{I})$) is given by $p(\boldsymbol{q}) \propto \exp(NS(\boldsymbol{q}))$, where the action $S(\boldsymbol{q})$ contains interactions between neuron activations and the kernels at each layer [9].

The DMFT introduced in [9] arises in the $N \to \infty$ limit when $p(\boldsymbol{q})$ is strongly peaked around the saddle point $\boldsymbol{q}_\infty$ where $\frac{\partial S}{\partial \boldsymbol{q}}|_{\boldsymbol{q}_\infty} = 0$. Analysis of the saddle point equations reveal that the training dynamics of the neural network can be alternatively described by a stochastic process. A key feature of this process is that it describes the training time evolution of the distribution of neuron pre-activations in each layer (informally the histogram of the elements of $\boldsymbol{h}_\mu^\ell(t)$) where each neuron's pre-activation behaves as an i.i.d. draw from this *single-site* stochastic process. We denote these random processes by $h_\mu^\ell(t)$. Kernels in (2) are now computed as *averages* over these infinite-width single site processes $\Phi_{\mu\nu}^\ell(t, s) = \langle \phi(h_\mu^\ell(t))\phi(h_\nu^\ell(s)) \rangle$, $G_{\mu\nu}^\ell(t, s) = \langle g_\mu^\ell(t)g_\nu^\ell(s) \rangle$, where averages arise from the $N \to \infty$ limit of the dot products in (2). DMFT also provides a set of self-consistent equations that describe the complete statistics of these random processes, which depend on the kernels, as well as other quantities. To make our notation and terminology clearer for a machine learning audience, we provide Table 1 for a definition of the physics terminology in machine learning language.

| Order params. $\boldsymbol{q}$ | Action $S(\boldsymbol{q})$ | Propagator $\boldsymbol{\Sigma}$ | Single Site Density |
|---|---|---|---|
| Concentrating variables | $\boldsymbol{q}$'s log-density | Asymptotic Covariance | Neuron Marginals |

Table 1: Relationship between the physics and ML terminology for the central objects in this paper. The $\boldsymbol{q}$ which concentrate at infinite width, but fluctuate at finite width $N$. This paper is primarily interested in $\boldsymbol{\Sigma}$, the asymptotic covariance of the order parameters.

## 4 Dynamical Fluctuations Around Mean Field Theory

We are interested in going beyond the infinite-width limit to study more realistic finite-width networks. In this regime, the order parameters $\boldsymbol{q}$ fluctuate in a $\mathcal{O}(1/\sqrt{N})$ neighborhood of $\boldsymbol{q}_\infty$ [55, 51, 53, 46].

Statistics of these fluctuations can be calculated from a general cumulant expansion (see App. D) [55, 56, 51]. We will focus on the leading-order corrections to the infinite-width limit in this expansion.

**Proposition 1** *The finite-width $N$ average of observable $O(\boldsymbol{q})$ across initializations, which we denote by $\langle O(\boldsymbol{q})\rangle_N$, admits an expansion of the form whose leading terms are*

$$\langle O(\boldsymbol{q})\rangle_N = \frac{\int d\boldsymbol{q}\exp\left(NS[\boldsymbol{q}]\right)O(\boldsymbol{q})}{\int d\boldsymbol{q}\exp\left(NS[\boldsymbol{q}]\right)} = \langle O(\boldsymbol{q})\rangle_\infty + N\left[\langle V(\boldsymbol{q})O(\boldsymbol{q})\rangle_\infty - \langle V(\boldsymbol{q})\rangle_\infty\langle O(\boldsymbol{q})\rangle_\infty\right] + ...,$$
(3)

*where $\langle\rangle_\infty$ denotes an average over the Gaussian distribution $\boldsymbol{q}\sim\mathcal{N}\left(\boldsymbol{q}_\infty, -\frac{1}{N}\left(\nabla_{\boldsymbol{q}}^2 S[\boldsymbol{q}_\infty]\right)^{-1}\right)$ and the function $V(\boldsymbol{q})\equiv S(\boldsymbol{q}) - S(\boldsymbol{q}_\infty) - \frac{1}{2}(\boldsymbol{q}-\boldsymbol{q}_\infty)^\top\nabla_{\boldsymbol{q}}^2 S(\boldsymbol{q}_\infty)(\boldsymbol{q}-\boldsymbol{q}_\infty)$ contains cubic and higher terms in the Taylor expansion of $S$ around $\boldsymbol{q}_\infty$. The terms shown include all the leading and sub-leading terms in the series in powers of $1/N$. The terms in ellipses are at least $\mathcal{O}(N^{-1})$ suppressed compared to the terms provided.*

The proof of this statement is given in App. D. The central object to characterize finite size effects is the unperturbed covariance (the *propagator*): $\boldsymbol{\Sigma} = -\left[\nabla^2 S(\boldsymbol{q}_\infty)\right]^{-1}$. This object can be shown to capture leading order fluctuation statistics $\left\langle(\boldsymbol{q}-\boldsymbol{q}_\infty)(\boldsymbol{q}-\boldsymbol{q}_\infty)^\top\right\rangle_N = \frac{1}{N}\boldsymbol{\Sigma} + \mathcal{O}(N^{-2})$ (App. D.1), which can be used to reason about, for example, expected square error over random initializations. Correction terms at finite width may give a possible explanation of the superior performance of wide networks at fixed $\gamma$ [7, 17, 18]. To calculate such corrections, in App. E, we provide a complete description of Hessian $\nabla_{\boldsymbol{q}}^2 S(\boldsymbol{q})$ and its inverse (the propagator) for a depth-$L$ network. This description constitutes one of our main results. The resulting expressions are lengthy and are left to App. E. Here, we discuss them at a high level. Conceptually there are two primary ingredients for obtaining the full propagator:

- Hessian sub-blocks $\kappa$ which describe the *uncoupled variances* of the kernels, such as

$$\kappa_{\mu\nu\alpha\beta}^{\Phi^\ell}(t,s,t',s') \equiv \left\langle\phi(h_\mu^\ell(t))\phi(h_\nu^\ell(s))\phi(h_\alpha^\ell(t'))\phi(h_\beta^\ell(s'))\right\rangle - \Phi_{\mu\nu}^\ell(t,s)\Phi_{\alpha\beta}^\ell(t',s') \quad (4)$$

Similar terms also appear in other studies on finite width Bayesian inference [13, 31, 32] and in studies on kernel variance at initialization [27, 14, 29, 53].

- Blocks which capture the *sensitivity* of field averages to pertubations of order parameters, such as

$$D_{\mu\nu\alpha\beta}^{\Phi^\ell\Phi^{\ell-1}}(t,s,t',s') \equiv \frac{\partial\left\langle\phi(h_\mu^\ell(t))\phi(h_\nu^\ell(s))\right\rangle}{\partial\Phi_{\alpha\beta}^{\ell-1}(t',s')}, \quad D_{\mu\nu\alpha}^{G^\ell\Delta}(t,s,t') \equiv \frac{\partial\left\langle g_\mu^\ell(t)g_\nu^\ell(s)\right\rangle}{\partial\Delta_\alpha(t')}, \quad (5)$$

where $\Delta_\mu(t) = -\frac{\partial\ell(f_\mu,y_\mu)}{\partial f_\mu}\big|_{f_\mu(t)}$ are error signal for each data point.

Abstractly, we can consider the uncoupled variances $\boldsymbol{\kappa}$ as "sources" of finite-width noise for each order parameter and the $\boldsymbol{D}$ blocks as summarizing a directed causal graph which captures how this noise propagates in the network (through layers and network predictions). In Figure 1, we illustrate this graph showing directed lines that represent causal influences of order parameters on fields and vice versa. For instance, if $\Phi^\ell$ were perturbed, $D^{\Phi^{\ell+1},\Phi^\ell}$ would quantify the resulting perturbation to $\Phi^{\ell+1}$ through the fields $h^{\ell+1}$.

In App. E, we calculate $\boldsymbol{\kappa}$ and $\boldsymbol{D}$ tensors, and show how to use them to calculate the propagator. As an example of our results:

**Proposition 2** *Partition $\boldsymbol{q}$ into primal $\boldsymbol{q}_1 = Vec\{f_\mu(t),\Phi_{\mu\nu}^\ell(t,s)...\}$ and conjugate variables $\boldsymbol{q}_2 = Vec\{\hat{f}_\mu(t),\hat{\Phi}_{\mu\nu}^\ell(t,s)...\}$. Let $\boldsymbol{\kappa} = \frac{\partial^2}{\partial\boldsymbol{q}_2\partial\boldsymbol{q}_2^\top}S[\boldsymbol{q}_1,\boldsymbol{q}_2]$ and $\boldsymbol{D} = \frac{\partial^2}{\partial\boldsymbol{q}_2\partial\boldsymbol{q}_1^\top}S[\boldsymbol{q}_1,\boldsymbol{q}_2]$, then the propagator for $\boldsymbol{q}_1$ has the form $\boldsymbol{\Sigma}_{\boldsymbol{q}_1} = \boldsymbol{D}^{-1}\boldsymbol{\kappa}\left[\boldsymbol{D}^{-1}\right]^\top$ (App E). The variables $\boldsymbol{q}_1$ are related to network observables, while conjugates $\boldsymbol{q}_2$ arise as Lagrange multipliers in the DMFT calculation. From the propagator $\boldsymbol{\Sigma}_{\boldsymbol{q}_1}$ we can read off the variance of network observables such as $N Var(f_\mu)\sim\Sigma_{f_\mu}$.*

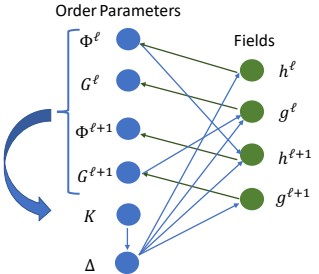

Figure 1: The directed causal graph between DMFT order parameters (blue) and fields (green) defines the $D$ tensors of our theory. Each arrow represents a causal dependence. $K$ denotes the NTK.

The necessary order parameters for calculating the fluctuations are obtained by solving the DMFT using numerical methods introduced in [9]. We provide a pseudocode for this procedure in App. F. We proceed to solve the equations defining $\boldsymbol{\Sigma}$ in special cases which are illuminating and numerically feasible including lazy training, two layer networks and deep linear NNs.

## 5  Lazy Training Limit

To gain some initial intuition about why kernel fluctuations alter learning dynamics, we first analyze the static kernel limit $\gamma \to 0$ where features are frozen. To prevent divergence of the network in this limit, we use a background subtracted function $\tilde{f}(\boldsymbol{x}, \boldsymbol{\theta}) = f(\boldsymbol{x}, \boldsymbol{\theta}) - f(\boldsymbol{x}, \boldsymbol{\theta}_0)$ which is identically zero at initialization [4]. For mean square error, the $N \to \infty$ and $\gamma \to 0$ limit is governed by $\frac{\partial \tilde{f}(\boldsymbol{x})}{\partial t} = \mathbb{E}_{\boldsymbol{x}' \sim \mathcal{D}} \Delta(\boldsymbol{x}') K(\boldsymbol{x}, \boldsymbol{x}')$ with $\Delta(\boldsymbol{x}) = y(\boldsymbol{x}) - \tilde{f}(\boldsymbol{x})$ (for MSE) and $K$ is the static (finite width and random) NTK. The finite-$N$ initial covariance of the NTK has been analyzed in prior works [27, 13, 14], which reveal a dependence on depth and nonlinearity. Since the NTK is static in the $\gamma \to 0$ limit, it has constant initialization variance through training. Further, all sensitivity blocks of the Hessian involving the kernels and the prediction errors $\boldsymbol{\Delta}$ (such as the $D^{\Phi^\ell, \Delta}$) vanish. We represent the covariance of the NTK as $\kappa(\boldsymbol{x}_1, \boldsymbol{x}_2, \boldsymbol{x}_2, \boldsymbol{x}_3) = N\text{Cov}(K(\boldsymbol{x}_1, \boldsymbol{x}_2), K(\boldsymbol{x}_3, \boldsymbol{x}_4))$. To identify the dynamics of the error $\boldsymbol{\Delta}$ covariance, we relate $K$, the finite width NTK, to $K_\infty$ which is the deterministic infinite width NTK $K_\infty$. We consider the eigendecomposition of the infinite-width NTK $K_\infty(\boldsymbol{x}, \boldsymbol{x}') = \sum_k \lambda_k \psi_k(\boldsymbol{x}) \psi_k(\boldsymbol{x}')$ with respect to the training distribution $\mathcal{D}$, and decompose $\kappa$ in this basis.

$$\kappa_{k\ell mn} = \langle \kappa(\boldsymbol{x}_1, \boldsymbol{x}_2, \boldsymbol{x}_3, \boldsymbol{x}_4) \psi_k(\boldsymbol{x}_1) \psi_\ell(\boldsymbol{x}_2) \psi_n(\boldsymbol{x}_3) \psi_m(\boldsymbol{x}_4) \rangle_{\boldsymbol{x}_1, \boldsymbol{x}_2, \boldsymbol{x}_3, \boldsymbol{x}_4 \sim \mathcal{D}}, \tag{6}$$

where averages are computed over the training distribution $\mathcal{D}$.

**Proposition 3** *For MSE loss, the prediction error covariance $\boldsymbol{\Sigma}^\Delta(t, s) = N\text{Cov}_0(\boldsymbol{\Delta}(t), \boldsymbol{\Delta}(s))$ satisfies a differential equation (App. H)*

$$\left( \frac{\partial}{\partial t} + \lambda_k \right) \left( \frac{\partial}{\partial s} + \lambda_\ell \right) \Sigma_{k\ell}^\Delta(t, s) = \sum_{nm} \kappa_{km\ell n} \Delta_m^\infty(t) \Delta_n^\infty(s), \tag{7}$$

*where $\Delta_k^\infty(t) \equiv \exp\left(-\lambda_k t\right) \langle \psi_k(\boldsymbol{x}) y(\boldsymbol{x}) \rangle_{\boldsymbol{x}}$ are the errors at infinite width for eigenmode k.*

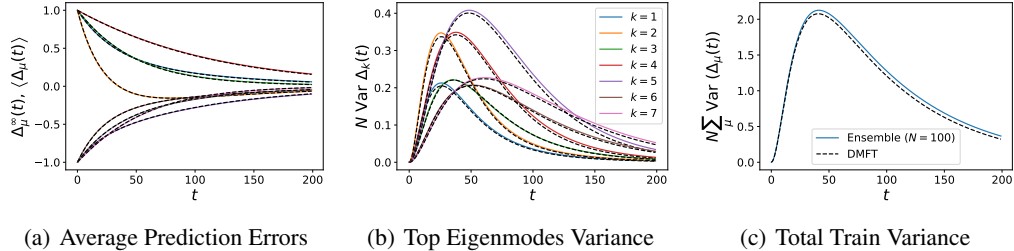

(a) Average Prediction Errors      (b) Top Eigenmodes Variance      (c) Total Train Variance

Figure 2: We show the accuracy of the lazy-limit ODE in equation (where) comapared to a two-layer finite width $N = 100$ ReLU network trained with $\gamma = 0.05$ on $P = 10$ random training data points. (a) The average dynamics over an ensemble of $E = 500$ networks (solid colors) compared to the infinite width predictions (dashed black). (b) The predicted finite size variance for each eigenmode of the error $\Delta_k(t) = \boldsymbol{\Delta}(t) \cdot \boldsymbol{\phi}_k$. These are not ordered simply by magnitude of eigenvalues or the target projections $y_k = \boldsymbol{y} \cdot \boldsymbol{\phi}_k$, but rather depend on all eigenvalue gaps $\lambda_k - \lambda_\ell$ for $k \neq \ell$ and also the $\kappa_{k\ell nm}$ tensor. (c) The total variance for all training points $N \sum_\mu \text{Var}\Delta_\mu(t) = N \sum_k \text{Var}\Delta_k(t)$ is also well predicted by the DMFT propagator equations.

An example verifying these dynamics is provided in Figure 2. In the case where the target is an eigenfunction $y = \psi_{k^*}$, the covariance has the form $\Sigma_{k\ell}^\Delta(t, s) = \kappa_{k\ell k^* k^*} \frac{\exp(-\lambda_{k^*}(t+s))}{(\lambda_k - \lambda_{k^*})(\lambda_\ell - \lambda_{k^*})}$. If the kernel is rank one with eigenvalue $\lambda$, then the dynamics have the simple form $\Sigma^\Delta(t, s) = \kappa y^2 \, t \, s \, e^{-\lambda(t+s)}$. We note that similar terms appear in the prediction dynamics obtained by truncating the Neural Tangent Hierarchy [10, 11], however those dynamics concerned small feature learning

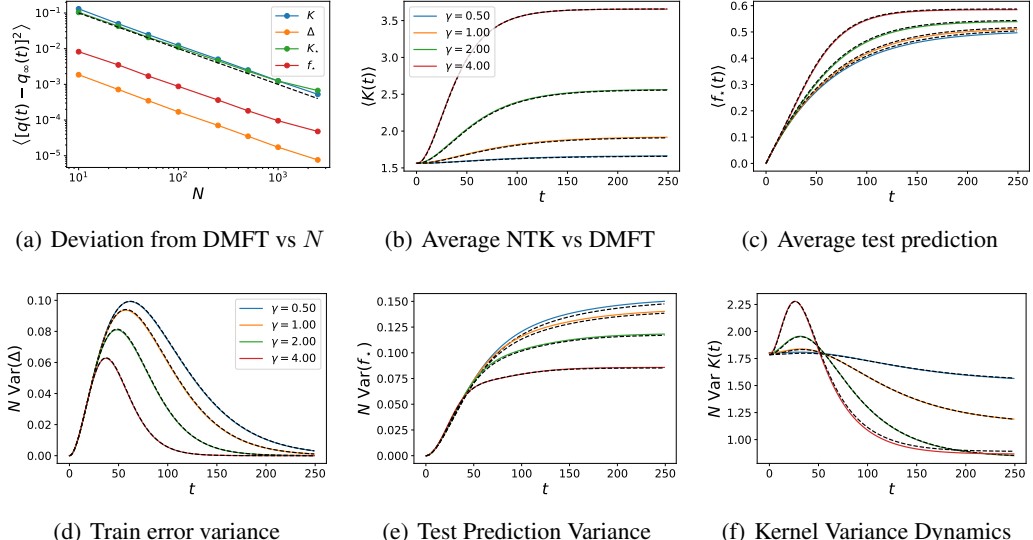

(a) Deviation from DMFT vs $N$      (b) Average NTK vs DMFT      (c) Average test prediction

(d) Train error variance      (e) Test Prediction Variance      (f) Kernel Variance Dynamics

Figure 3: An ensemble of $E = 1000$ two layer $N = 256$ tanh networks trained on a single training point. Dashed black lines are DMFT predictions. (a) The square deviation from the infinite width DMFT scales as $\mathcal{O}(1/N)$ for all order parameters. (b) The ensemble average NTK $\langle K(t) \rangle$ (solid colors) and (c) ensemble average test point predictions $f_\star(t)$ for a point with $\frac{\boldsymbol{x} \cdot \boldsymbol{x}_\star}{D} = 0.5$ closely follow the infinite width predictions (dashed black). (d) The variance (estimated over the ensemble) of the train error $\Delta(t) = y - f(t)$ initially increases and then decreases as the training point is fit. (e) The variance of $f_\star$ increases with time but decreases with $\gamma$. (f) The variance of the NTK during feature learning experiences a transient increase before decreasing to a lower value.

corrections rather than from initialization variance (App. H.1). Corrections to the mean $\langle \Delta \rangle$ are analyzed in App. H.2. We find that the variance and mean correction dynamics involves non-trivial coupling across eigendirections with a mixture of exponentials with timescales $\{\lambda_k^{-1}\}$.

## 6 Rich Regime in Two-Layer Networks

In this section, we analyze how feature learning alters the variance through training. We show a denoising effect where the signal to noise ratios of the order parameters improve with feature learning.

### 6.1 Kernel and Error Coupled Fluctuations on Single Training Example

In the rich regime, the kernel evolves over time but inherits fluctuations from the training errors $\boldsymbol{\Delta}$. To gain insight, we first study a simplified setting where the data distribution is a single training example $\boldsymbol{x}$ and single test point $\boldsymbol{x}_\star$ in a two layer network. We will track $\Delta(t) = y - f(\boldsymbol{x}, t)$ and the test prediction $f_\star(t) = f(\boldsymbol{x}_\star, t)$. To identify the dynamics of these predictions we need the NTK $K(t)$ on the train point, as well as the train-test NTK $K_\star(t)$. In this case, all order parameters can be viewed as scalar functions of a single time index (unlike the deep network case, see App. E).

**Proposition 4** *Computing the Hessian of the DMFT action and inverting (App. I), we obtain the following covariance for $\boldsymbol{q}_1 = Vec\{\Delta(t), f_\star(t), K(t), K_\star(t)\}_{t \in \mathbb{R}_+}$*

$$\boldsymbol{\Sigma}_{\boldsymbol{q}_1} = \begin{bmatrix} \mathbf{I} + \boldsymbol{\Theta}_K & 0 & \boldsymbol{\Theta}_\Delta & 0 \\ -\boldsymbol{\Theta}_{K_\star} & \mathbf{I} & 0 & -\boldsymbol{\Theta}_\Delta \\ -\boldsymbol{D} & 0 & \mathbf{I} & 0 \\ -\boldsymbol{D}_\star & 0 & 0 & \mathbf{I} \end{bmatrix}^{-1} \begin{bmatrix} 0 & 0 & 0 & 0 \\ 0 & 0 & 0 & 0 \\ 0 & 0 & \boldsymbol{\kappa} & \boldsymbol{\kappa}_\star^\top \\ 0 & 0 & \boldsymbol{\kappa}_\star & \boldsymbol{\kappa}_{\star\star} \end{bmatrix} \begin{bmatrix} \mathbf{I} + \boldsymbol{\Theta}_K & 0 & \boldsymbol{\Theta}_\Delta & 0 \\ -\boldsymbol{\Theta}_{K_\star} & \mathbf{I} & 0 & -\boldsymbol{\Theta}_\Delta \\ -\boldsymbol{D} & 0 & \mathbf{I} & 0 \\ -\boldsymbol{D}_\star & 0 & 0 & \mathbf{I} \end{bmatrix}^{-1\top},$$

(8)

*where $[\boldsymbol{\Theta}_K](t,s) = \Theta(t-s)K(s)$, $[\boldsymbol{\Theta}_\Delta](t,s) = \Theta(t-s)\Delta(s)$ are Heaviside step functions and $D(t,s) = \left\langle \frac{\partial}{\partial \Delta(s)} (\phi(h(t))^2 + g(t)^2) \right\rangle$ and $D_\star(t,s) = \left\langle \frac{\partial}{\partial \Delta(s)} (\phi(h(t))\phi(h_\star(t)) + g(t)g_\star(t)) \right\rangle$*

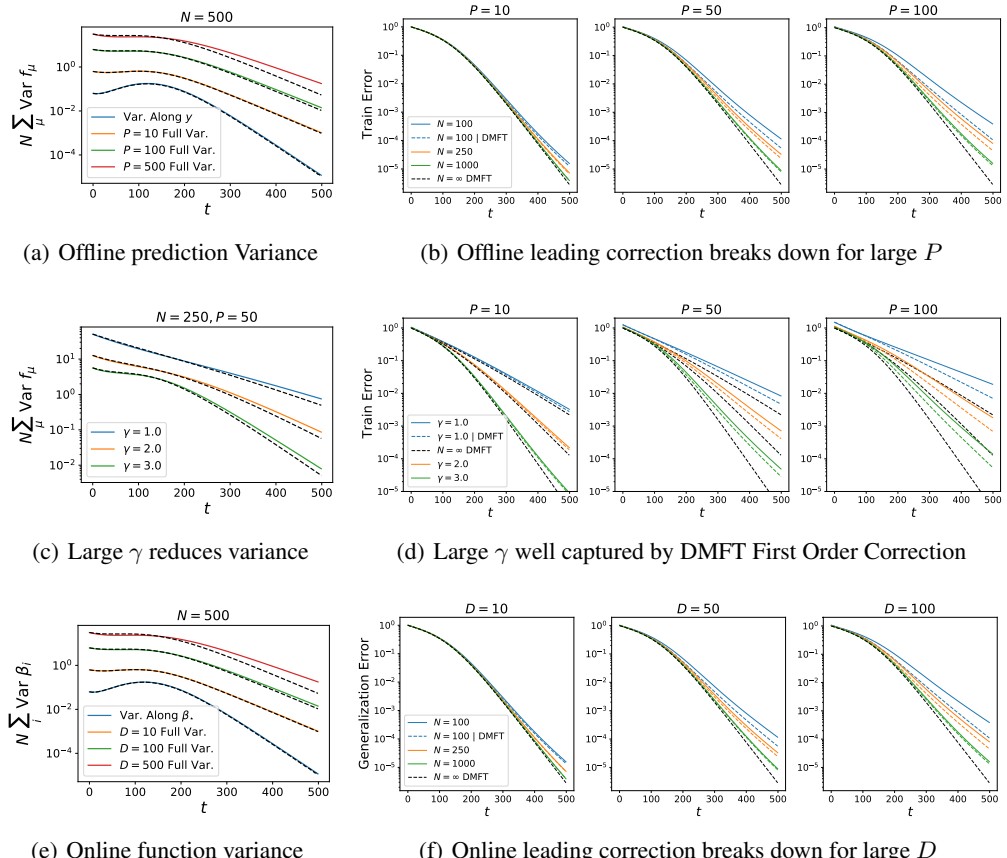

Figure 4: Large input dimension or multiple samples amplify finite size effects in a simple two layer model with unstructured data. Black dashed lines are theory. (a) The variance of offline learning with $P$ training examples in a two layer linear network. (b) The leading perturbative approximation to the train error breaks down when samples $P$ becomes comparable to $N$. (c)-(d) Richer training reduces variance. (e)-(f) Online learning in a depth 2 linear network has identical dynamics and finite width fluctuations, but with predictor variance $\sim D/N$ for input dimension $D$ (Appendix K).

*quantify sensitivity of the kernel to perturbations in the error signal $\Delta(s)$. Lastly $\kappa$ and $\kappa_\star$ are the uncoupled variances of $K(t)$ and $K_{\star\star}(t)$ and $\kappa_\star$ is the uncoupled covariance of $K(t), K_\star(t)$.*

In Fig. 3, we plot the resulting theory (diagonal blocks of $\Sigma_{q_1}$ from Equation 8) for two layer neural networks. As predicted by theory, all average squared deviations from the infinite width DMFT scale as $\mathcal{O}(N^{-1})$. Similarly, the average kernels $\langle K \rangle$ and test predictions $\langle f_\star \rangle$ change by a larger amount for larger $\gamma$ (equation (79)). The experimental variances also match the theory quite accurately. The variance of the train error $\Delta(t)$ peaks earlier and at a lower value for richer training, but all variances go to zero at late time as the model approaches the interpolation condition $\Delta = 0$. As $\gamma \to 0$ the curve approaches $N \, \mathrm{Var}(\Delta(t)) \sim \kappa \, y^2 \, t^2 \, e^{-2t}$, where $\kappa$ is the initial NTK variance (see Section 5). While the train prediction variance goes to zero, the test point prediction does not, with richer networks reaching a lower asymptotic variance. We suspect this dynamical effect could explain lower variance observed in feature learning networks compared to lazy networks [7, 18]. In Fig. A.1, we show that the reduction in variance is not due to a reduction in the uncoupled variance $\kappa(t, s)$, which increases in $\gamma$. Rather the reduction in variance is driven by the coupling of perturbations across time.

## 6.2 Offline Training with Multiple Samples or Online Training in High Dimension

In this section we go beyond the single sample equations of the prior section and explore training with multiple $P$ examples. In this case, we have training errors $\{\Delta_\mu(t)\}_{\mu=1}^P$ and multiple kernel entries $K_{\mu\nu}(t)$ (App. E). Each of the errors $\Delta_\mu(t)$ receives a $\mathcal{O}(N^{-1/2})$ fluctuation, the training error $\sum_\mu \langle \Delta_\mu^2 \rangle$ has an additional variance on the order of $\mathcal{O}(\frac{P}{N})$. In the case of two-layer linear

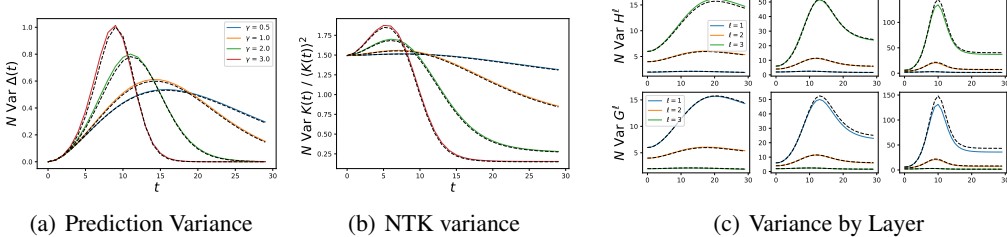

(a) Prediction Variance  (b) NTK variance  (c) Variance by Layer

Figure 5: Depth 4 linear network with single training point. Black dashed lines are theory. (a) The variance of the training error along the task relevant subspace. We see that unlike the two layer model, more feature learning can lead to larger peaks in the finite size variance. (b) The variance of the NTK in the task relevant subspace. When properly normalized against the square of the mean $\langle K(t) \rangle^2$, the final NTK variance decreases with feature learning. (c) The gap in feature kernel variance across different layers of the network is amplified by feature learning strength $\gamma$.

networks trained on whitened data ($\frac{1}{D}\boldsymbol{x}_\mu \cdot \boldsymbol{x}_\nu = \delta_{\mu\nu}$), the equations for the propagator simplify and one can separately solve for the variance of $\boldsymbol{\Delta}(t) \in \mathbb{R}^P$ along signal direction $\boldsymbol{y} \in \mathbb{R}^P$ and along each of the $P-1$ orthogonal directions (App. J). At infinite width, the task-orthogonal component $\boldsymbol{\Delta}_\perp$ vanishes and only the signal dimension $\Delta_y(t)$ evolves in time with differential equation [9, 46]

$$\frac{d}{dt}\Delta_y(t) = 2\sqrt{1 + \gamma^2(y - \Delta_y(t))^2}\, \Delta_y(t)\, , \ \boldsymbol{\Delta}_\perp(t) = 0. \tag{9}$$

However, at finite width, both the $\Delta_y(t)$ and the $P-1$ orthogonal variables $\boldsymbol{\Delta}_\perp$ inherit initialization variance, which we represent as $\Sigma_{\Delta_y}(t,s)$ and $\Sigma_\perp(t,s)$. In Fig. 4 (a)-(b) we show this approximate solution $\langle |\boldsymbol{\Delta}(t)|^2 \rangle \sim \Delta_y(t)^2 + \frac{2}{N}\Delta_y^1(t)\Delta_y(t) + \frac{1}{N}\Sigma_{\Delta_y}(t,t) + \frac{(P-1)}{N}\Sigma_\perp(t,t) + \mathcal{O}(N^{-2})$ across varying $\gamma$ and varying $P$ (see Appendix J for $\Sigma_{\Delta_y}$ and $\Sigma_\perp$ formulas). We see that variance of train point predictions $f_\mu(t)$ increases with the total number of points despite the signal of the target vector $\sum_\mu y_\mu^2$ being fixed. In this model, the bias correction $\frac{2}{N}\Delta_y^1(t)\Delta_y(t)$ is always $\mathcal{O}(1/N)$ but the variance correction is $\mathcal{O}(P/N)$. The fluctuations along the $P-1$ orthogonal directions begin to dominate the variance at large $P$. Fig. 4 (b) shows that as $P$ increases, the leading order approximation breaks down as higher order terms become relevant. Analysis for online training reveals identical fluctuation statistics, but with variance that scales as $\sim D/N$ (Appendix K) as we verify in Figure 4 (e)-(f).

## 7 Deep Networks

In networks deeper than two layers, the DMFT propagator has complicated dependence on non-diagonal (in time) entries of the feature kernels (see App. E). This leads to Hessian blocks with four time and four sample indices such as $D^{\Phi^\ell}_{\mu\nu\alpha\beta}(t,s,t',s') = \frac{\partial}{\partial\Phi^{\ell-1}_{\alpha\beta}(t',s')}\langle\phi(h^\ell_\mu(t))\phi(h^\ell_\nu(s))\rangle$, rendering any numerical calculation challenging. However, in deep linear networks trained on whitened data, we can exploit the symmetry in sample space and the Gaussianity of preactivation features to exactly compute derivatives without Monte Carlo sampling as we discuss in App. L. An example set of results for a depth 4 network is provided in Fig. 5. The variance for the feature kernels $H^\ell$ accumulate finite size variance by layer along the forward pass and the gradient kernels $G^\ell$ accumulate variance on the backward pass. The SNR of the kernels $\frac{\langle H \rangle^2}{N\,\mathrm{Var}(H)}$ improves with feature learning, suggesting that richer networks will be better modeled by their mean field limits. Examples of the off-diagonal correlations obtained from the propagator are provided in App. Fig. A.3.

## 8 Variance can be Small Near Edge of Stability

In this section, we move beyond the gradient flow formalism and ask what large step sizes do to finite size effects. Recent studies have identified that networks trained at large learning rates can be qualitatively different than networks in the gradient flow regime, including the catapult [57] and edge of stability (EOS) phenomena [19–21]. In these settings, the kernel undergoes an initial scale

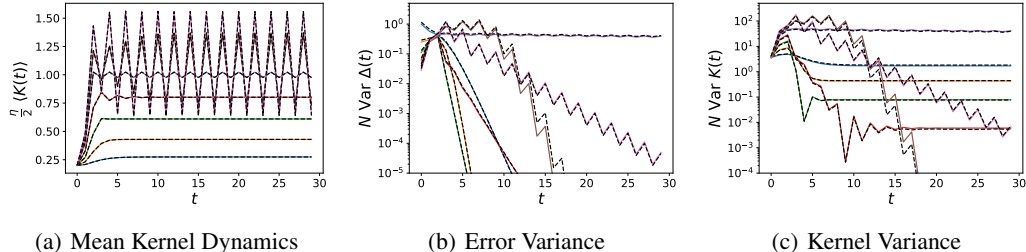

(a) Mean Kernel Dynamics        (b) Error Variance        (c) Kernel Variance

Figure 6: Edge of stability effects do not imply deviations from infinite width behavior. Black dashed lines are theory. (a) The average kernel over an ensemble of several $N = 500$ NNs (solid color). For small $\gamma$, the kernel reaches its asymptote before hitting the edge of stability. For large $\gamma$, the kernel increases and then oscillates around $2/\eta$. (b)-(c) Remarkably variance due to finite size can *reduce* during training (for $\gamma$ smaller and larger than the critical value $\sim 1/\eta$), showing that infinite width DMFT can be predictive of finite NNs trained with large learning rate.

growth before exhibiting either a recovery or a clipping effect. In this section, we explore whether these dynamics are highly sensitive to initialization variance or if finite networks are well captured by mean field theory. Following [57], we consider two layer networks trained on a single example $|\boldsymbol{x}|^2 = D$ and $y = 1$. We use learning rate $\eta$ and feature learning strength $\gamma$. The infinite width mean field equations for the prediction $f_t$ and the kernel $K_t$ are (App. M)

$$f_{t+1} = f_t + \eta K_t \Delta_t + \eta^2 \gamma^2 f_t \Delta_t^2 \, , \;\; K_{t+1} = K_t + 4\eta\gamma^2 f_t \Delta_t + \eta^2 \gamma^2 \Delta_t^2 K_t. \quad (10)$$

For small $\eta$, the equations are well approximated by the gradient flow limit and for small $\gamma$ corresponds to a discrete time linear model. For large $\eta\gamma > 1$, the kernel $K$ progressively sharpens (increases in scale) until it reaches $2/\eta$ and then oscillates around this value. It may be expected that near the EOS, the large oscillations in the kernels and predictions could lead to amplified finite size effects, however, we show in Fig. 6 that the leading order propagator elements decrease even after reaching the EOS threshold, indicating *reduced* disagreement between finite and infinite width dynamics.

## 9   Finite Width Alters Bias, Training Rate, and Variance in Realistic Tasks

To analyze the effect of finite width on neural network dynamics during realistic learning tasks, we studied a vanilla depth-6 ReLU CNN trained on CIFAR-10 (experimental details in App. B, G.2) In Fig. 7, we train an ensemble of $E = 8$ independently initialized CNNs of each width $N$. Wider networks not only have better performance for a single model (solid), but also have lower bias (dashed), measured with ensemble averaging of the logits. Because of faster convergence of wide networks, we observe wider networks have higher variance, but if we plot variance at fixed ensembled training accuracy, wider networks have consistently lower variance (Fig. 7(d)).

We next seek an explanation for why wider networks after ensembling trains at a faster *rate*. Theoretically, this can be rationalized by a finite-width alteration to the ensemble averaged NTK, which governs the convergence timescale of the ensembled predictions (App. G.1). Our analysis in App. G.1 suggests that the rate of convergence receives a finite size correction with leading correction $\mathcal{O}(N^{-1})$ G.2. To test this hypothesis, we fit the ensemble training loss curve to exponential function $\mathcal{L} \approx C \exp\left(-R_N t\right)$ where $C$ is a constant. We plot the fit $R_N$ as a function of $N^{-1}$ result in Fig. 7(e). For large $N$, we see the leading behavior is linear in $N^{-1}$, but begins to deviate at small $N$ as a quadratic function of $N^{-1}$, suggesting that second order effects become relevant around $N \lesssim 100$.

In App. Fig. A.4, we train a smaller subset of CIFAR-10 where we find that $R_N$ is well approximated by a $\mathcal{O}(N^{-1})$ correction, consistent with the idea that higher sample size drives the dynamics out of the leading order picture. We also analyze the effect of $\gamma$ on variance in this task. In App. Fig. A.5, we train $N = 64$ models with varying $\gamma$. Increased $\gamma$ reduces variance of the logits and alters the representation (measured with kernel-task alignment), the training and test accuracy are roughly insensitive to the richness $\gamma$ in the range we considered.

## 10   Discussion

We studied the leading order fluctuations of kernels and predictions in mean field neural networks. Feature learning dynamics can reduce undesirable finite size variance, making finite networks order

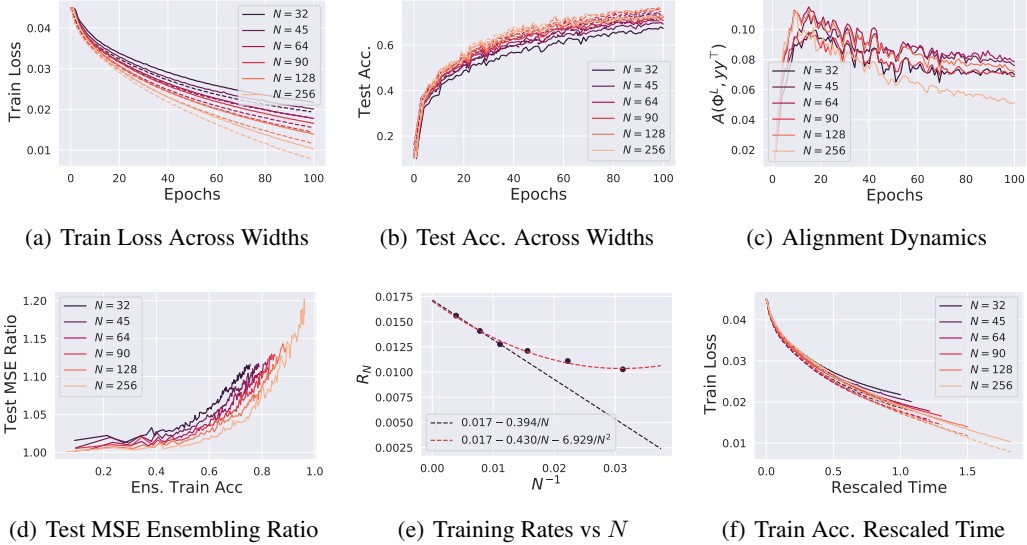

Figure 7: Depth 6 CNN trained on CIFAR-10 for different widths $N$ with richness $\gamma = 0.2$, $E = 8$ ensembles. (a)-(b) For this range of widths, we find that smaller networks perform worse in train and test error, not only in terms of the single models (solid) but also in terms of bias (dashed). The delayed training of ensembled finite width models indicates that the correction to the mean order parameters (App. G) is non-negligible. (c) Alignment of the average kernel to test labels is also not conserved across width. (d) The ratio of the test MSE for a single model to the ensembled logit MSE. (e) The fitted rate $R_N$ of training width $N$ models as a function of $N^{-1}$. We rescale the time axis by $R_N$ to allow for a fair comparison of prediction variance for networks at comparable performance levels. (f) In rescaled time, ensembled network training losses (dashed) are coincident.

parameters closer to the infinite width limit. In several toy models, we revealed some interesting connections between the influence of feature learning, depth, sample size, and large learning rate and the variance of various DMFT order parameters. Lastly, in realistic tasks, we illustrated that bias corrections can be significant as rates of learning can be modified by width. Though our full set of equations for the leading finite size fluctuations are quite general in terms of network architecture and data structure, they are only derived at the level of rigor of physics rather than a formally rigorous proof which would need several additional assumptions to make the perturbation expansion properly defined. Further, the leading terms in our perturbation series involving only $\Sigma$ does not capture the complete finite size distribution defined in Eq. (3), especially as the sample size becomes comparable to the width. It would be interesting to see if proportional limits of the rich training regime where samples and width scale linearly can be examined dynamically [58]. Future work could explore in greater detail the higher order contributions from averages involving powers of $V(\boldsymbol{q})$ by examining cubic and higher derivatives of $S$ in Eq. (3). It could also be worth examining in future work how finite size impacts other biologically plausible learning rules, where the effective NTK can have asymmetric (over sample index) fluctuations [46]. Also of interest would be computing the finite width effects in other types of architectures, including residual networks with various branch scalings [59, 60]. Further, even though we expect our perturbative expressions to give a precise asymptotic description of finite networks in mean field/$\mu$P, the resulting expressions are not realistically computable in deep networks trained on large dataset size $P$ for long times $T$ since the number of Hessian entries scales as $\mathcal{O}(T^4 P^4)$ and a matrix of this size must be stored in memory and inverted in the general case. Future work could explore solveable special cases such as high dimensional limits.

## Code Availability

Code to reproduce the experiments in this paper is provided at `https://github.com/Pehlevan-Group/dmft_fluctuations`. Details about numerical methods and computational implementation can be found in Appendices F and N.

**Acknowledgements**

CP is supported by NSF Award DMS-2134157, NSF CAREER Award IIS-2239780, and a Sloan Research Fellowship. BB is supported by a Google PhD research fellowship and NSF Award DMS-2134157. This work has been made possible in part by a gift from the Chan Zuckerberg Initiative Foundation to establish the Kempner Institute for the Study of Natural and Artificial Intelligence. The computations in this paper were run on the FASRC cluster supported by the FAS Division of Science Research Computing Group at Harvard University. BB thanks Alex Atanasov, Jacob Zavatone-Veth for their comments on this manuscript and Boris Hanin, Greg Yang, Mufan Bill Li and Jeremy Cohen for helpful discussions.

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

# Appendix

## A Additional Figures

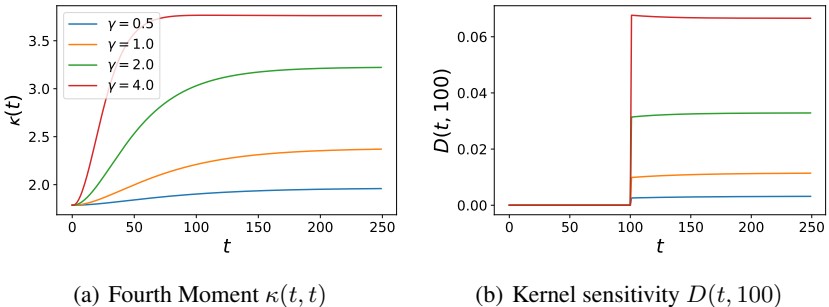

(a) Fourth Moment $\kappa(t, t)$

(b) Kernel sensitivity $D(t, 100)$

Figure A.1: The $\kappa$ and $D$ functions for varying $\gamma$ in Figure 3. (a) The uncoupled kernel variance $\kappa(t, t)$ increases monotonically with $\gamma$. This reveals that the dynamical filtering of $\kappa$ is what is responsible for the late time decrease in variance during feature learning. (b) The tensor $D(t, s)$ measures sensitivity of kernel at time $t$ to perturbation in $\Delta$ at time $s$. The $D(t, s)$ entries also increase with $\gamma$. This suggests that the reduction in variance of the training error and the kernel are not due to reduction in $\kappa$, but rather a dynamical filtering effect due to scale growth in $K_\infty$ and rapid reduction in $\Delta_\infty$.

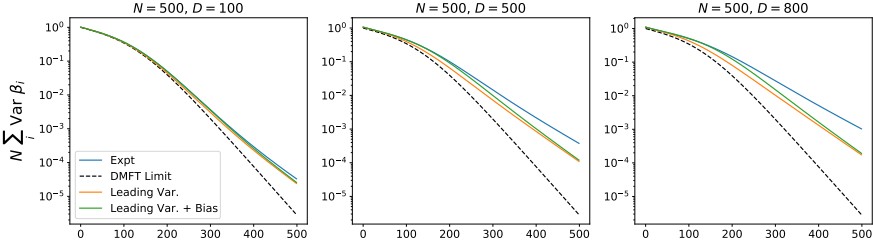

Figure A.2: A comparison of the bias and variance corrections in the toy model of Figure 4. At small $D/N$ (or $P/N$ for offline training) the leading variance and the leading variance and leading bias both track the experiment. Both the bias and the variance contribute positively towards the total generalization error since the variance correction alone (orange) exceeds the DMFT limiting error (dashed) and the variance and bias correction together (green) exceed variance alone (orange). However, for large $D/N$ (or $P/N$) the leading order picture fails to describe the finite width experiment, indicating significant variance possibly at higher order scales (like $D^2/N^2, D^3/N^3, ...$).

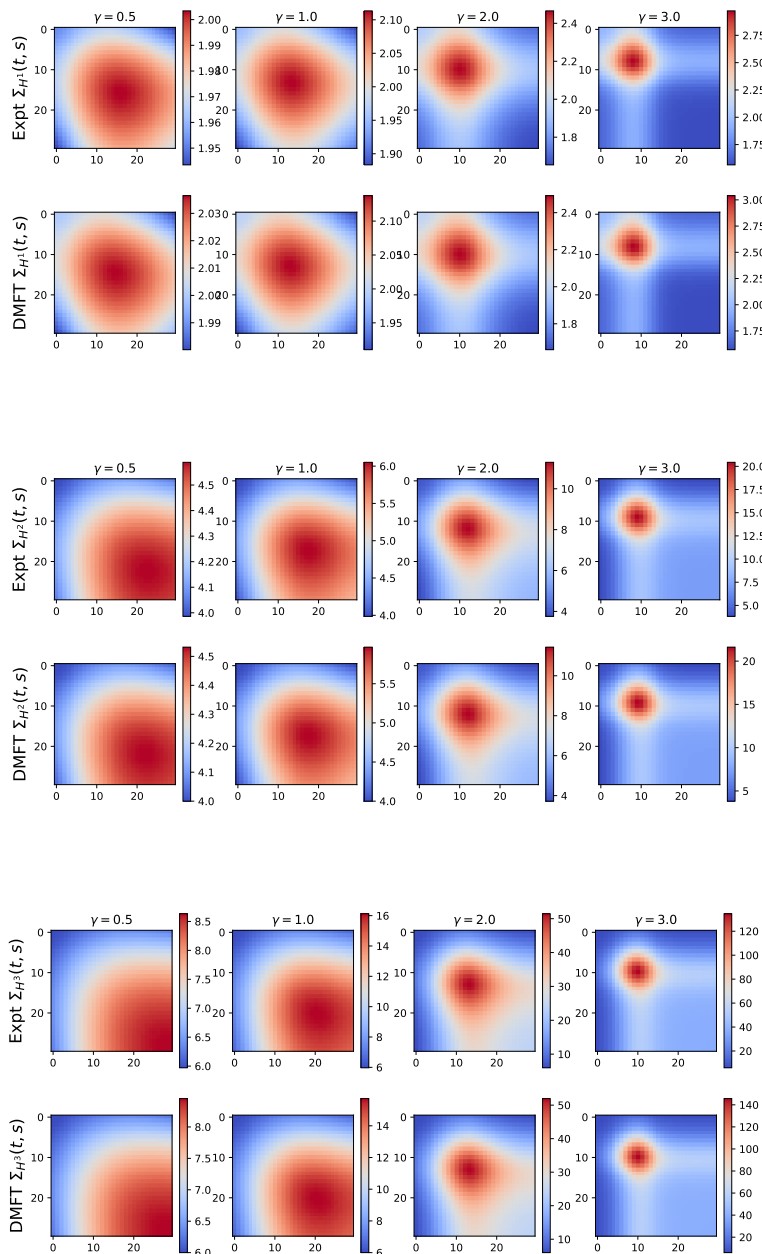

Figure A.3: The covariance of kernel entries across pairs of time points $\Sigma_{H^\ell}(t,s) = N\,\mathrm{Cov}(H^\ell(t,t), H^\ell(s,s))$ for depth 4 linear network trained on whitened data. The variance becomes increasingly localized in time as feature learning $\gamma$ increases.

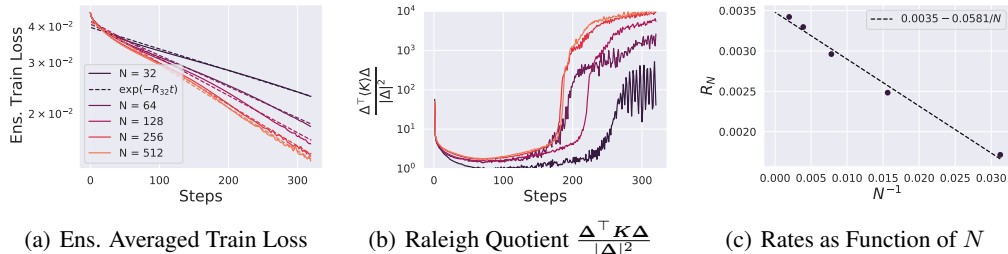

(a) Ens. Averaged Train Loss     (b) Raleigh Quotient $\frac{\Delta^\top K \Delta}{|\Delta|^2}$     (c) Rates as Function of $N$

Figure A.4: The ensemble averaged train loss for the same depth 6 CNN trained on a random subsample of $P = 64$ CIFAR-10 points. Training is full batch gradient descent with $\gamma = 0.05$. (a) The ensemble train accuracy for this subset of CIFAR-10 is well modeled as an exponential in time $\mathcal{L}(t) \propto \exp(-R_N t)$ with a rate $R_N$ that depends on width. (b) The projection of the errors $\Delta$ on the average NTK $\langle K \rangle$ (which is related to the rate of decay of the training loss, see Appendix G) reveals that wider networks are more aligned with their instantaneous error signals. (c) The rates $R_N$ are indeed a linear functions of $N^{-1}$, with $R_N = R_\infty + \frac{R^1}{N}$, consistent with the average NTK $\langle K \rangle$ receiving a $N^{-1}$ correction. Using ensembling to find a scaling law like that above can thus allow one extrapolate the training rate of infinite width mean field models.

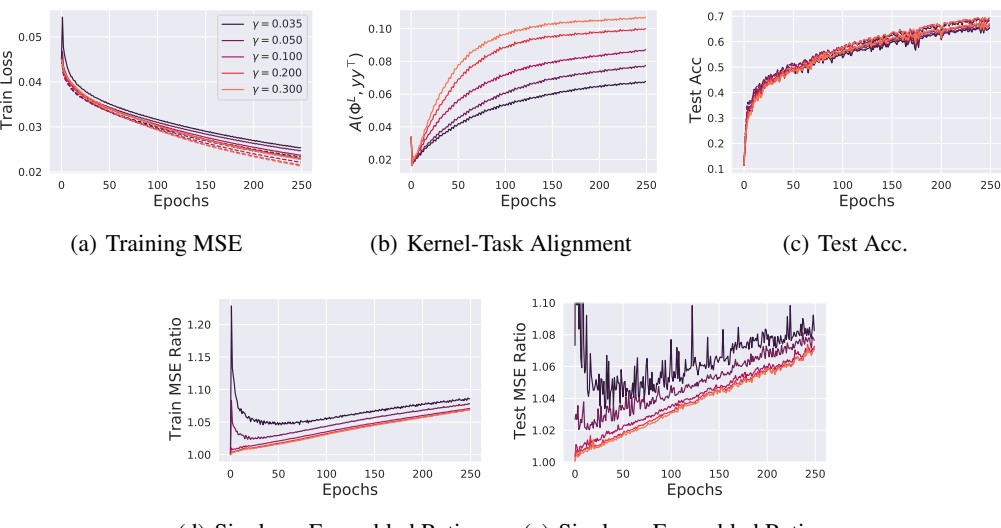

(a) Training MSE     (b) Kernel-Task Alignment     (c) Test Acc.

(d) Single vs Ensembled Ratio     (e) Single vs Ensembled Ratio

Figure A.5: Width $N = 64$ depth 6 CNNs trained on the full CIFAR-10 with MSE. An ensemble of size $E = 10$ randomly initialized networks are trained. (a) Training MSE for varying $\gamma$. (b) Final layer kernel-task alignment does strongly depend on $\gamma$, despite similar train dynamics. (c) Top-1 classification test accuracy is only slightly different across $\gamma$. A small benefit from ensembling is visible late in training. (c) Initialization variance (measured by the ratio of single model to ensembled MSE) for training and test losses. Richer networks have lower variance throughout training. (b) Networks have distinct kernel dynamics when trained with different $\gamma$ as evidenced by the alignment (cosine similarity) between the final layer feature kernel $\Phi^L$ and the target test labels $\boldsymbol{y}$.

## B    CIFAR-10 Experimental Details

We trained the following depth 6 CNN architecture in the mean field parameterization using FLAX [61] on a single GPU. The bias parameters were zero in each hidden Conv layer, but were used for the readout weights. The networks were trained with MSE loss on centered $10$ dimensional targets $\boldsymbol{y}_\mu \in \mathbb{R}^{10}$ for $\mu \in [P]$. Each convolution was followed by an average pooling operation. To obtain mean field behavior, NTK parameterization with a modified final layer is used [7, 9].

```python
from flax import linen as nn
import jax.numpy as jnp

class CNN(nn.Module):

  width: int

  def setup(self):
    kif = nn.initializers.normal(stddev = 1.0) # O_N(1) entries
    self.conv1 = nn.Conv(features = self.width, kernel_init = kif,
    use_bias = False, kernel_size = (3,3))
    self.conv2 = nn.Conv(features = self.width, kernel_init = kif,
    use_bias = False, kernel_size = (3,3))
    self.conv3 = nn.Conv(features = self.width, kernel_init = kif,
    use_bias = False, kernel_size = (3,3))
    self.conv4 = nn.Conv(features = self.width, kernel_init  = kif,
    use_bias = False, kernel_size = (3,3))
    self.conv5 = nn.Conv(features = self.width, kernel_init  = kif,
    use_bias = False, kernel_size = (3,3))
    self.readout = nn.Dense(features = 10, use_bias = True,
    kernel_init = kif)
    return

  def __call__(self, x, train = True):
    N = self.width
    D = 3
    x = self.conv1(x) / jnp.sqrt(D * 9)
    x = jnp.sqrt(2.0) * nn.relu(x)
    x = nn.avg_pool(x, window_shape=(2,2), strides = (2,2)) # 32 x 32
    -> 16 x 16
    x = self.conv2(x) / jnp.sqrt(N*9) # explicit N^{-1/2}
    x = jnp.sqrt(2.0) * nn.relu(x)
    x = nn.avg_pool(x, window_shape=(2,2), strides = (2,2)) # 16 x 16
    -> 8 x 8
    x = self.conv3(x)/jnp.sqrt(N*9)
    x = jnp.sqrt(2.0) * nn.relu(x)
    x = nn.avg_pool(x, window_shape=(2,2), strides = (2,2)) # 8 x 8 ->
     4 x 4
    x = self.conv4(x) / jnp.sqrt(N*9)
    x = jnp.sqrt(2.0) * nn.relu(x)
    x = nn.avg_pool(x, window_shape=(2,2), strides = (2,2)) #  4 x 4
    -> 2 x 2
    x = self.conv5(x) / jnp.sqrt(N*9)
    x = jnp.sqrt(2.0) * nn.relu(x)
    x = nn.avg_pool(x, window_shape=(2,2), strides = (2,2)) #  2 x 2
    -> 1 x 1
    x = x.reshape((x.shape[0], -1))  # flatten
    x = self.readout(x) / N # for mean field scaling
    return x
```

All models were trained with standard SGD with a batch size of 256. Each element in the ensemble of $E$ networks is trained on identical batches presented in identical order. For the Figure 7 experiments, the raw learning rate is scaled as $\eta = 10N\sqrt{\gamma}$ with $\gamma = 0.2$ (note that mean field theory requires scaling the raw learning rate linearly with $N$ since the raw NTK is $\mathcal{O}(N^{-1})$ [9]). For Figure A.5, the learning rate is $\eta = 5N\sqrt{\gamma}$. We find that choosing $\eta \propto \sqrt{\gamma}$ gives approximately conserved training

times across $\gamma$ (though distinct representation dynamics). The Figure A.4 shows the dynamics of fitting $P = 64$ training points with full batch gradient descent and $\gamma = 0.1$.

## C Review of DMFT: Deriving the Action

In this section we derive the DMFT action which contains all of the necessary statistical information about randomly initialized finite width $N$ networks. From the action $S$ the DMFT saddle point and the propagator can be computed. This derivation follows closely the original derivation by Bordelon & Pehlevan [9]. We start by writing the gradient flow dynamics on weight matrices

$$\frac{d}{dt}\boldsymbol{W}^\ell(t) = \frac{\gamma}{\sqrt{N}} \sum_{\mu=1}^P \Delta_\mu(t) \boldsymbol{g}_\mu^{\ell+1}(t) \phi(\boldsymbol{h}_\mu^\ell(t))^\top \tag{11}$$

where $\Delta_\mu(t) = -\frac{\partial \mathcal{L}}{\partial f_\mu(t)}$ are the error signals and $\boldsymbol{g}_\mu^\ell(t) = N\gamma \frac{\partial f_\mu(t)}{\partial \boldsymbol{h}_\mu^\ell(t)}$ are the back-propagation signals. The prediction dynamics satisfy

$$\frac{d}{dt}f_\mu(t) = \sum_\nu K_{\mu\nu}(t)\Delta_\nu(t) \tag{12}$$

where $K_{\mu\nu}(t)$ is the instantaneous neural tangent kernel (NTK). At finite width $N$ all of the above quantities depend on the precise initialization of the network. We transform the weight dynamics into an integral equation and use the recurrence for $\boldsymbol{h}^\ell$ to obtain the following

$$\boldsymbol{h}^{\ell+1}(t) = \frac{1}{\sqrt{N}}\boldsymbol{W}^\ell(0)\phi(\boldsymbol{h}_\mu^\ell(t)) + \gamma \int_0^t ds \sum_\nu \Phi_{\mu\nu}^\ell(t,s)\boldsymbol{g}_\nu^{\ell+1}(s)$$

$$\boldsymbol{g}_\mu^\ell(t) = \dot{\phi}(\boldsymbol{h}_\mu^\ell(t)) \odot \boldsymbol{z}_\mu^\ell(t)$$

$$\boldsymbol{z}_\mu^\ell(t) = \frac{1}{\sqrt{N}}\boldsymbol{W}^\ell(0)^\top \boldsymbol{g}_\mu^{\ell+1}(t) + \gamma \int_0^t ds \sum_\nu G_{\mu\nu}^{\ell+1}(t,s)\phi(\boldsymbol{h}_\nu^\ell(s)) \tag{13}$$

where we introduced the feature and gradient kernels

$$\Phi_{\mu\nu}^\ell(t,s) = \frac{1}{N}\phi(\boldsymbol{h}_\mu^\ell(t)) \cdot \phi(\boldsymbol{h}_\nu^\ell(s)) \ , \ G_{\mu\nu}^\ell(t,s) = \frac{1}{N}\boldsymbol{g}_\mu^\ell(t) \cdot \boldsymbol{g}_\nu^\ell(s). \tag{14}$$

Written this way, we see that the source of the disorder which depends on the initial random weights $\boldsymbol{W}^\ell(0)$ comes through the fields

$$\boldsymbol{\chi}_\mu^{\ell+1}(t) = \frac{1}{\sqrt{N}}\boldsymbol{W}^\ell(0)\phi(\boldsymbol{h}_\mu^\ell(t)) \ , \ \boldsymbol{\xi}_\mu^{\ell+1}(t) = \frac{1}{\sqrt{N}}\boldsymbol{W}^\ell(0)^\top \boldsymbol{g}_\mu^\ell(t). \tag{15}$$

If we can characterize the distribution of the fields $\boldsymbol{\chi}_\mu^\ell(t)$ and $\boldsymbol{\xi}_\mu^\ell(t)$, then we can consequently characterize the distribution of $\boldsymbol{h}_\mu^\ell(t), \boldsymbol{g}_\mu^\ell(t)$. We therefore choose to study the moment generating functional

$$Z[\{\boldsymbol{j}^\ell, \boldsymbol{v}^\ell\}] = \left\langle \exp\left( \sum_{\ell\mu} \int dt \ [\boldsymbol{j}_\mu^\ell(t) \cdot \boldsymbol{\chi}_\mu^\ell(t) + \boldsymbol{v}_\mu^\ell(t) \cdot \boldsymbol{\chi}_\mu^\ell(t)] \right) \right\rangle_{\boldsymbol{\theta}_0} \tag{16}$$

Moments of these fields can be computed through differentiation with respect to the sources $\boldsymbol{j}, \boldsymbol{v}$ near zero-source ($\boldsymbol{j} = \boldsymbol{v} = 0$)

$$\left\langle \chi_{\mu_1}^{\ell_1}(t_1)...\chi_{\mu_n}^{\ell_n}(t_n)\xi_{\mu_1}^{\ell_1}(t_1)...\xi_{\mu_m}^{\ell_m}(t_m) \right\rangle$$

$$= \frac{\delta}{\delta j_{\mu_1}^{\ell_1}(t_1)} \cdots \frac{\delta}{\delta j_{\mu_n}^{\ell_n}(t_n)} \frac{\delta}{\delta v_{\mu_1}^{\ell_1}(t_1)} \cdots \frac{\delta}{\delta v_{\mu_m}^{\ell_m}(t_m)} Z[\{\boldsymbol{j}^\ell, \boldsymbol{v}^\ell\}]|_{\boldsymbol{j}=\boldsymbol{v}=0}. \tag{17}$$

To average over the initial weights, we introduce a Fourier representation of the Dirac-Delta function $1 = \int dz \delta(z) = \int \frac{dz d\hat{z}}{2\pi} \exp(i\hat{z}z)$. We perform this transformation for each of the fields to enforce

their definition

$$\delta\left(\boldsymbol{\chi}_\mu^\ell(t) - \frac{1}{\sqrt{N}}\boldsymbol{W}^\ell(0)\phi(\boldsymbol{h}_\mu^\ell(t))\right) = \int \frac{d\hat{\boldsymbol{\chi}}_\mu^\ell(t)}{(2\pi)^N}\exp\left(\hat{\boldsymbol{\chi}}_\mu^\ell(t)\cdot\left[\boldsymbol{\chi}_\mu^\ell(t) - \frac{1}{\sqrt{N}}\boldsymbol{W}^\ell(0)\phi(\boldsymbol{h}_\mu^\ell(t))\right]\right)$$

$$\delta\left(\boldsymbol{\xi}_\mu^\ell(t) - \frac{1}{\sqrt{N}}\boldsymbol{W}^\ell(0)^\top\boldsymbol{g}_\mu^\ell(t)\right) = \int \frac{d\hat{\boldsymbol{\xi}}_\mu^\ell(t)}{(2\pi)^N}\exp\left(\hat{\boldsymbol{\xi}}_\mu^\ell(t)\cdot\left[\boldsymbol{\xi}_\mu^\ell(t) - \frac{1}{\sqrt{N}}\boldsymbol{W}^\ell(0)^\top\boldsymbol{g}_\mu^{\ell+1}(t)\right]\right)$$

$$(18)$$

We insert these Dirac delta functions so that we can directly average over the weights

$$\ln\mathbb{E}_{\boldsymbol{W}^\ell(0)}\exp\left(-\frac{i}{\sqrt{N}}\mathrm{Tr}\boldsymbol{W}^\ell(0)^\top\int dt\sum_\mu\left[\hat{\boldsymbol{\chi}}_\mu^{\ell+1}(t)\phi(\boldsymbol{h}_\mu^\ell(t))^\top + \boldsymbol{g}_\mu^{\ell+1}(t)\hat{\boldsymbol{\xi}}_\mu^\ell(t)^\top\right]\right)$$

$$= -\frac{1}{2}\sum_{\mu\nu}\int dtds\left[\hat{\boldsymbol{\chi}}_\mu^{\ell+1}(t)\cdot\hat{\boldsymbol{\chi}}_\nu^{\ell+1}(s)\Phi_{\mu\nu}^\ell(t,s) + \hat{\boldsymbol{\xi}}_\mu^\ell(t)\cdot\hat{\boldsymbol{\xi}}_\nu^\ell(s)G_{\mu\nu}^{\ell+1}(t,s)\right]$$

$$-\frac{1}{N}\sum_{\mu\nu}\int dtds(\hat{\boldsymbol{\chi}}_\mu^{\ell+1}(t)\cdot\boldsymbol{g}_\nu^{\ell+1}(s))(\phi(\boldsymbol{h}_\mu^\ell(t))\cdot\hat{\boldsymbol{\xi}}_\nu^\ell(s)) \tag{19}$$

where we introduced the kernels $\Phi^\ell, G^\ell$. We next introduce the order parameter

$$A_{\mu\nu}^\ell(t,s) = -\frac{i}{N}\phi(\boldsymbol{h}_\mu^\ell(t))\cdot\hat{\boldsymbol{\xi}}_\nu^\ell(s) \tag{20}$$

To enforce the definitions of the new order parameters $\{\Phi, G, A\}$ we again introduce Dirac-delta functions

$$\delta\left(N\Phi_{\mu\nu}^\ell(t,s) - \phi(\boldsymbol{h}_\mu^\ell(t))\cdot\phi(\boldsymbol{h}_\nu^\ell(s))\right)$$

$$= \int \frac{d\hat{\Phi}_{\mu\nu}^\ell(t,s)}{2\pi i}\exp\left(\hat{\Phi}_{\mu\nu}^\ell(t,s)\left[N\Phi_{\mu\nu}^\ell(t,s) - \phi(\boldsymbol{h}_\mu^\ell(t))\cdot\phi(\boldsymbol{h}_\nu^\ell(s))\right]\right) \tag{21}$$

Analogous constraints for $G$ and $A$ are enforced with conjugate variables $\hat{G}, B$. After introducing these variables, we find that the moment generating functional has the form

$$Z = \int\prod_{\ell\mu\nu ts}\frac{d\hat{\Phi}_{\mu\nu}^\ell(t,s)d\Phi_{\mu\nu}^\ell(t,s)}{2\pi i}\frac{d\hat{G}_{\mu\nu}^\ell(t,s)dG_{\mu\nu}^\ell(t,s)}{2\pi i}\frac{d\hat{G}_{\mu\nu}^\ell(t,s)dG_{\mu\nu}^\ell(t,s)}{2\pi i}\frac{dA_{\mu\nu}^\ell(t,s)dB_{\mu\nu}^\ell(t,s)}{2\pi i}$$

$$\exp\left(NS[\{\Phi^\ell, \hat{\Phi}^\ell, G^\ell, \hat{G}^\ell, A^\ell, B^\ell\}]\right) \tag{22}$$

where $S$ is the $\mathcal{O}(1)$ DMFT *action* which defines the statistical distribution over the dynamics. The action takes the form

$$S = \sum_{\ell\mu\nu}\int dt\,ds\left[\hat{\Phi}_{\mu\nu}^\ell(t,s)\Phi_{\mu\nu}^\ell(t,s) + \hat{G}_{\mu\nu}^\ell(t,s) - A_{\mu\nu}^\ell(t,s)B_{\nu\mu}^\ell(s,t)\right]$$

$$+\frac{1}{N}\sum_{\ell=1}^L\sum_{i=1}^N\ln\mathcal{Z}_\ell[\{j_i^\ell, v_i^\ell\}] \tag{23}$$

where $\mathcal{Z}_\ell$ is the single site stochastic process for layer $\ell$ which defines the marginal distribution of $\chi, \xi$, with the following form

$$\mathcal{Z}_\ell[\{j^\ell(t), v^\ell(t)\}] = \int\prod_{\mu t}\frac{d\hat{\chi}_\mu^\ell(t)d\chi_\mu^\ell(t)}{2\pi}\frac{d\hat{\xi}_\mu^\ell(t)d\xi_\mu^\ell(t)}{2\pi}\exp\left(\int dt\sum_\mu[j_\mu^\ell(t)\chi_\mu^\ell(t) + v_\mu^\ell(t)\xi_\mu^\ell(t)]\right)$$

$$\exp\left(-\frac{1}{2}\sum_{\mu\nu}\int dtds\left[\hat{\Phi}_{\mu\nu}^\ell(t,s)\hat{\chi}_\mu^\ell(t)\hat{\chi}_\nu^\ell(s) + \hat{G}_{\mu\nu}^\ell(t,s)\hat{\xi}_\mu^\ell(t)\hat{\xi}_\nu^\ell(s)\right]\right)$$

$$\exp\left(-i\sum_{\mu\nu}\int dtds\left[B_{\mu\nu}^\ell(t,s)\hat{\xi}_\mu^\ell(t)\phi(h_\nu^\ell(s)) + A_{\mu\nu}^{\ell-1}(t,s)\hat{\chi}_\mu^\ell(t)g_\nu^\ell(s)\right]\right)$$

$$\exp\left(i\sum_\mu\int dt[\hat{\chi}_\mu^\ell(t)\chi_\mu^\ell(t) + \hat{\xi}_\mu^\ell(t)\xi_\mu^\ell(t)]\right) \tag{24}$$

where in the above, the $\{h, g\}$ fields should be regarded as functionals of $\{\chi, \xi\}$. At zero source $j^\ell, v^\ell \to 0$ this function $S$ can be regarded as the log density for the complete collection of order parameters $q = \{\hat{\Phi}, \Phi, \hat{G}, G, A, B\}$ which collectively control the dynamics. Concretely, we have that $p(q) \propto \exp(NS(q))$. In the next section we explore an approximation scheme for averages over this distribution at large $N$.

## D  Cumulant Expansion of Observables

We are interested in a principled power series expansion (in $1/N$) of any observable average $\langle O(q) \rangle$ that depends on DMFT order parameters $q$. At any width $N$ the observable average takes the form

$$\langle O(q) \rangle_N = \frac{\int dq \exp(NS(q)) O(q)}{\int dq \exp(NS(q))} \tag{25}$$

As discussed in the main text, the $N \to \infty$ limit gives $\langle O(q) \rangle_N \sim O(q_\infty)$ where $\frac{\partial S}{\partial q}|_{q_\infty} = 0$ by a steepest descent argument [55]. We assume that $S$'s Hessian is negative semidefinite so that $\boldsymbol{\Sigma} \equiv -\left[\nabla^2 S(q)|_{q_\infty}\right]^{-1} \succeq 0$ and Taylor expand $S(q)$ around the saddle point $q_\infty$ giving $S(q) = S(q_\infty) + \frac{1}{2}(q - q_\infty)^\top \nabla^2 S(q)(q - q_\infty) + V(q - q_\infty)$. We note that the remainder function $V$ contains only cubic and higher powers of $q - q_\infty \equiv \boldsymbol{\delta}/\sqrt{N}$. The variable $\boldsymbol{\delta}$ will be order $\mathcal{O}(1)$. This will allow us to verify that additional terms are suppressed in powers of $1/N$. Expanding both the numerator and denominator's integrands in powers of $V$, we find

$$
\begin{aligned}
\langle O(q) \rangle_N &= \frac{\int dq \exp\left(-\frac{N}{2}(q - q_\infty)^\top \boldsymbol{\Sigma}^{-1}(q - q_\infty) + NV(q - q_\infty)\right) O(q)}{\int dq \exp\left(-\frac{N}{2}(q - q_\infty)^\top \boldsymbol{\Sigma}^{-1}(q - q_\infty) + NV(q - q_\infty)\right)} \\
&= \frac{\int d\boldsymbol{\delta} \exp\left(-\frac{1}{2}\boldsymbol{\delta}^\top \boldsymbol{\Sigma}^{-1}\boldsymbol{\delta}\right)\left(1 + NV + \frac{N^2}{2}V^2 + ...\right) O(q_\infty + N^{-1/2}\boldsymbol{\delta})}{\int d\boldsymbol{\delta} \exp\left(-\frac{1}{2}\boldsymbol{\delta}^\top \boldsymbol{\Sigma}^{-1}\boldsymbol{\delta}\right)\left(1 + NV + \frac{N^2}{2}V^2 + ...\right)} \\
&= \frac{\langle O \rangle_\infty + N \langle VO \rangle_\infty + \frac{N^2}{2!}\langle V^2 O \rangle_\infty + \frac{N^3}{3!}\langle V^3 O \rangle_\infty + ...}{1 + N \langle V \rangle_\infty + \frac{N^2}{2!}\langle V^2 \rangle_\infty + \frac{N^3}{3!}\langle V^3 \rangle_\infty + ...} \\
&= \langle O \rangle_\infty \frac{1 + N \langle VO \rangle_\infty / \langle O \rangle_\infty + \frac{N^2}{2!}\langle V^2 O \rangle_\infty / \langle O \rangle_\infty + \frac{N^3}{3!}\langle V^3 O \rangle_\infty / \langle O \rangle_\infty + ...}{1 + N \langle V \rangle_\infty + \frac{N^2}{2!}\langle V^2 \rangle_\infty + \frac{N^3}{3!}\langle V^3 \rangle_\infty + ...}
\end{aligned} \tag{26}
$$

where $\langle \rangle_\infty$ represents an average over the Gaussian fluctuation $\mathcal{N}\left(q_\infty, -\frac{1}{N}\left[\nabla_q^2 S(q_\infty)\right]^{-1}\right)$. We see that the series in the denominator contains terms of the form $\frac{N^k}{k!}\langle V^k \rangle_\infty$ while the numerator depends on terms of the form $\frac{N^k}{k!}\langle V^k O \rangle_\infty / \langle O \rangle_\infty$. In either of these power series, the $k$-th term can contribute at most

$$\frac{N^k \langle V^k O \rangle_\infty}{\langle O \rangle_\infty} , \ N^k \langle V^k \rangle_\infty \sim \begin{cases} \mathcal{O}(N^{-(k+1)/2}) & k \text{ odd} \\ \mathcal{O}(N^{-k/2}) & k \text{ even} \end{cases} \tag{27}$$

since $V$ contributes only cubic and higher terms. Thus each term in the numerator and denominator's series contains increasing powers of $1/N$. Concretely, each of the two series have terms of order $\{N^0, N^{-1}, N^{-1}, N^{-2}, N^{-2}, ...\}$. Thus any quantity of the form $\frac{\langle O \rangle}{\langle O \rangle_\infty}$ admits a ratio of power series in powers of $1/N$. One could truncate each of the series in the numerator and denominator to a desired order in $N$. Alternatively, the denominator could be expanded giving a single series (the cumulant expansion [56]). The first few terms in the cumulant expansion have the form

$$
\begin{aligned}
\langle O \rangle_N &= \langle O \rangle_\infty + N \left[\langle OV \rangle_\infty - \langle O \rangle_\infty \langle V \rangle_\infty\right] \\
&\quad + \frac{N^2}{2}\left[\langle V^2 O \rangle_\infty - 2\langle VO \rangle_\infty \langle V \rangle_\infty + 2 \langle V \rangle_\infty^2 \langle O \rangle_\infty - \langle V^2 \rangle_\infty \langle O \rangle_\infty\right] + ...
\end{aligned} \tag{28}
$$

In this work, we mainly are interested in the leading order correction to $\langle O \rangle$ which can always be obtained with the truncation after the terms linear in $V$ for any observable $O$.

### D.1 Square Deviation from DMFT

We will now analyze the fluctuation statistics of our order parameters around the saddle point $\left\langle (\boldsymbol{q} - \boldsymbol{q}_\infty)(\boldsymbol{q} - \boldsymbol{q}_\infty)^\top \right\rangle_N$ which has the form

$$
\begin{aligned}
\left\langle (\boldsymbol{q} - \boldsymbol{q}_\infty)(\boldsymbol{q} - \boldsymbol{q}_\infty)^\top \right\rangle_N &= \frac{\left\langle (\boldsymbol{q} - \boldsymbol{q}_\infty)(\boldsymbol{q} - \boldsymbol{q}_\infty)^\top \right\rangle_\infty + N \left\langle V (\boldsymbol{q} - \boldsymbol{q}_\infty)(\boldsymbol{q} - \boldsymbol{q}_\infty)^\top \right\rangle_\infty + \dots}{1 + N \left\langle V \right\rangle_\infty + \dots} \\
&= \left[ \frac{\frac{1}{N}\boldsymbol{\Sigma} + \mathcal{O}(N^{-2})}{1 + \mathcal{O}(N^{-1})} \right] \sim \frac{1}{N}\boldsymbol{\Sigma} + \mathcal{O}(N^{-2}),
\end{aligned}
\tag{29}
$$

as stated in the main text and verified empirically in Figure 3 (a). The reason that the terms in the numerator involving $V$ can be no larger than $\mathcal{O}(N^{-2})$ comes from vanishing of odd moments for $\boldsymbol{q} - \boldsymbol{q}_\infty$ in the unperturbed distribution. Thus the leading expression for $\left\langle (\boldsymbol{q} - \boldsymbol{q}_\infty)(\boldsymbol{q} - \boldsymbol{q}_\infty)^\top \right\rangle$ only depends on $\boldsymbol{\Sigma}$ and not on $V$.

### D.2 Mean Deviation from DMFT

Although the square displacement from DMFT only depended on $\boldsymbol{\Sigma}$ and not on $V$, we note that the *average order parameter displacement* $\left\langle \boldsymbol{q} - \boldsymbol{q}_\infty \right\rangle$ does receive a $\mathcal{O}(1/N)$ correction that depends on the perturbed potential $V$

$$
\begin{aligned}
\left\langle \boldsymbol{q} - \boldsymbol{q}_\infty \right\rangle_N &= \frac{\left\langle \boldsymbol{q} - \boldsymbol{q}_\infty \right\rangle_\infty + N \left\langle (\boldsymbol{q} - \boldsymbol{q}_\infty)V \right\rangle_\infty + \frac{N^2}{2}\left\langle (\boldsymbol{q} - \boldsymbol{q}_\infty)V^2 \right\rangle_\infty + \dots}{1 + N \left\langle V \right\rangle_\infty + \frac{N^2}{2}\left\langle V^2 \right\rangle_\infty + \dots} \\
&\sim \frac{\boldsymbol{\Sigma} \left\langle \frac{\partial V}{\partial \boldsymbol{q}} \right\rangle_\infty + \mathcal{O}(N^{-2})}{1 + \mathcal{O}(N^{-1})} \sim \boldsymbol{\Sigma} \left\langle \frac{\partial V}{\partial \boldsymbol{q}} \right\rangle_\infty + \mathcal{O}(N^{-2}).
\end{aligned}
\tag{30}
$$

where in the last line we used Stein's lemma (Gaussian integration by parts) for the Gaussian distribution over $\boldsymbol{q}$. Note that $\left\langle \frac{\partial V}{\partial \boldsymbol{q}} \right\rangle_\infty \sim \mathcal{O}\left(\frac{1}{N}\right)$ since the derivative of the cubic term in $V$ gives a quadratic function of $\boldsymbol{q} - \boldsymbol{q}_\infty$, whose average must be $\mathcal{O}(N^{-1})$. In this work, we focus primarily on the structure of the propagator, but outline a general recipe for getting the leading mean correction in Appendix G and H.2.

### D.3 Covariance of Order Parameters

Lastly, we combine the previous two observations to reason about the scaling of the order parameter covariance over initializations. We note that the leading covariance of the order parameters over random initializations is also given by the propagator: $\mathrm{Cov}(\boldsymbol{q}) \sim \frac{1}{N}\boldsymbol{\Sigma} + \mathcal{O}(N^{-2})$, since

$$
\begin{aligned}
\mathrm{Cov}(\boldsymbol{q}) &= \left\langle \left(\boldsymbol{q} - \langle \boldsymbol{q} \rangle_N\right)\left(\boldsymbol{q} - \langle \boldsymbol{q} \rangle_N\right)^\top \right\rangle_N \\
&= \left\langle \left(\boldsymbol{q} - \boldsymbol{q}_\infty\right)\left(\boldsymbol{q} - \boldsymbol{q}_\infty\right)^\top \right\rangle_N - \left\langle \left(\boldsymbol{q}_\infty - \langle \boldsymbol{q} \rangle_N\right)\left(\boldsymbol{q}_\infty - \langle \boldsymbol{q} \rangle_N\right)^\top \right\rangle_N \\
&\sim \frac{1}{N}\boldsymbol{\Sigma} + \mathcal{O}(N^{-2})
\end{aligned}
\tag{31}
$$

due to the arguments above which showed that $\left\langle (\boldsymbol{q} - \boldsymbol{q}_\infty)(\boldsymbol{q} - \boldsymbol{q}_\infty)^\top \right\rangle \sim \frac{1}{N}\boldsymbol{\Sigma} + \mathcal{O}(N^{-2})$ and that $\boldsymbol{q}_\infty - \langle \boldsymbol{q} \rangle_N \sim \mathcal{O}(N^{-1})$. Therefore, in the leading order picture, it is safe to associate $\boldsymbol{\Sigma}$ with the covariance of order parameters over random initializations of the network weights.

## E  Propagator Structure for the full DMFT Action

In this section, we examine the propagator structure for the full DMFT action. This action is modified from other prior works [9, 46] to include the evolution of the network prediction errors $\Delta(t)$. Those prior works noted that $\Delta$ and the NTK $K$ are deterministic functions of deterministic order parameters $\{\Phi^\ell, G^\ell\}$ in the $N \to \infty$ limit so those authors did not explicitly include $\Delta$ or $K$ in the action. At finite width $N$, including $\Delta, K$ in the action is crucial as the fluctuation in prediction errors $\Delta$ has significant consequences for dynamical fluctuations of kernels through the preactivation and

pre-gradient fields. In this section, we will mainly focus on gradient flow, but we describe large step size in Appendix M.

$$S = \sum_{\ell\mu\nu} \int dtds \left[ \hat{\Phi}^\ell_{\mu\nu}(t,s)\Phi^\ell_{\mu\nu}(t,s) + \hat{G}^\ell_{\mu\nu}(t,s)G^\ell_{\mu\nu}(t,s) - \gamma^2 A^\ell_{\nu\mu}(s,t)B^\ell_{\mu\nu}(t,s) \right]$$

$$+ \sum_\mu \int dt \hat{\Delta}_\mu(t) \left[ \Delta_\mu(t) - y_\mu + \sum_\nu \int ds \Theta(t-s)K_{\mu\nu}(s)\Delta_\nu(s) \right]$$

$$+ \sum_{\mu\nu} \int dt \hat{K}_{\mu\nu}(t) \left[ K_{\mu\nu}(t) - \sum_\ell G^{\ell+1}_{\mu\nu}(t)\Phi^\ell_{\mu\nu}(t) \right]$$

$$+ \sum_\ell \ln \mathcal{Z}_\ell[\boldsymbol{\Delta}, \hat{\boldsymbol{\Phi}}^\ell, \hat{\boldsymbol{G}}^\ell, \boldsymbol{\Phi}^{\ell-1}, \boldsymbol{G}^{\ell+1}, \boldsymbol{A}^{\ell-1}, \boldsymbol{B}^\ell] \tag{32}$$

where the single site moment generating functionals $\mathcal{Z}_\ell$ have the form

$$\mathcal{Z}_\ell = \mathbb{E}_{\{h^\ell_\mu(t), z^\ell_\mu(t)\}} \exp\left( -\sum_{\mu\nu} \int dtds \left[ \phi(h^\ell_\mu(t))\phi(h^\ell_\nu(s))\hat{\Phi}^\ell_{\mu\nu}(t,s) + g^\ell_\mu(t)g^\ell_\nu(s)\hat{G}^\ell_{\mu\nu}(t,s) \right] \right)$$

$$h^\ell_\mu(t) = u^\ell_\mu(t) + \gamma \int_0^t ds \sum_\nu \left[ \Phi^{\ell-1}_{\mu\nu}(t,s)\Delta_\nu(s) + A^{\ell-1}_{\mu\nu}(t,s) \right] g^\ell_\nu(s) , \; \{u^\ell_\mu(t)\} \sim \mathcal{GP}(0, \boldsymbol{\Phi}^{\ell-1})$$

$$z^\ell_\mu(t) = r^\ell_\mu(t) + \gamma \int_0^t ds \sum_\nu \left[ G^{\ell+1}_{\mu\nu}(t,s)\Delta_\nu(s) + B^\ell_{\mu\nu}(t,s) \right] \phi(h^\ell_\nu(s)) , \; \{r^\ell_\mu(t)\} \sim \mathcal{GP}(0, \boldsymbol{G}^{\ell+1})$$

$$\tag{33}$$

with $g^\ell_\mu(t) = \dot{\phi}(h^\ell_\mu(t))z^\ell_\mu(t)$. The saddle point equations give the infinite width evolution of our order parameters.

$$\frac{\partial S}{\partial \hat{\Phi}^\ell_{\mu\nu}(t,s)} = \Phi^\ell_{\mu\nu}(t,s) - \left\langle \phi(h^\ell_\mu(t))\phi(h^\ell_\nu(s)) \right\rangle = 0$$

$$\frac{\partial S}{\partial \hat{G}^\ell_{\mu\nu}(t,s)} = G^\ell_{\mu\nu}(t,s) - \left\langle g^\ell_\mu(t)g^\ell_\nu(s) \right\rangle = 0$$

$$\frac{\partial S}{\partial A^\ell_{\nu\mu}(s,t)} = -\gamma^2 B^\ell_{\mu\nu}(t,s) + \gamma \left\langle \frac{\partial \phi(h^\ell_\mu(t))}{\partial r^\ell_\nu(s)} \right\rangle = 0$$

$$\frac{\partial S}{\partial B^\ell_{\nu\mu}(s,t)} = -\gamma^2 A^\ell_{\mu\nu}(t,s) + \gamma \left\langle \frac{\partial g^\ell_\mu(t)}{\partial u^\ell_\nu(s)} \right\rangle = 0$$

$$\frac{\partial S}{\partial \hat{K}_{\mu\nu}(t)} = K_{\mu\nu}(t) - \sum_\ell G^{\ell+1}_{\mu\nu}(t,t)\Phi^\ell_{\mu\nu}(t,t) = 0$$

$$\frac{\partial S}{\partial \hat{\Delta}_\mu(t)} = \Delta_\mu(t) - y_\mu + \int_0^t ds \sum_\nu K_{\mu\nu}(s)\Delta_\nu(s) = 0 \tag{34}$$

These equations exactly recover the mean field description obtained [9]. Note that $\langle \rangle$ for field averages is an average defined by $\mathcal{Z}_\ell$ and is distinct from the types averages $\langle \rangle$, $\langle \rangle_\infty$ we have been considering over the order parameters $\boldsymbol{q}$. The complementary set of equations for the primal variables, such as $\frac{\partial S}{\partial \Phi^\ell_{\mu\nu}(t,s)} = 0$, give that $\hat{K} = \hat{\Delta} = \hat{\Phi} = \hat{G} = 0$ at the saddle point. We now set out to compute the Hessan $\nabla^2_{\boldsymbol{q}} S$. To simplify the set of expressions, we will only explicitly write out the nonvanishing

blocks. We will start with second derivatives involving only pairs of dual variables $\{\hat{\Phi}, \hat{G}, A, B\}$

$$\frac{\partial^2 S}{\partial \hat{\Phi}^\ell_{\mu\nu}(t,s)\partial \hat{\Phi}^\ell_{\alpha\beta}(t',s')} = \left\langle \phi(h^\ell_\mu(t))\phi(h^\ell_\nu(s))\phi(h^\ell_\alpha(t'))\phi(h^\ell_\beta(s')) \right\rangle - \Phi^\ell_{\mu\nu}(t,s)\Phi^\ell_{\alpha\beta}(t',s')$$

$$\equiv \kappa^{\Phi^\ell}_{\mu\nu\alpha\beta}(t,s,t',s')$$

$$\frac{\partial^2 S}{\partial \hat{G}^\ell_{\mu\nu}(t,s)\partial \hat{G}^\ell_{\alpha\beta}(t',s')} = \left\langle g^\ell_\mu(t)g^\ell_\nu(s)g^\ell_\alpha(t')g^\ell_\beta(s') \right\rangle - G^\ell_{\mu\nu}(t,s)G^\ell_{\alpha\beta}(t',s')$$

$$\equiv \kappa^{G^\ell}_{\mu\nu\alpha\beta}(t,s,t',s')$$

$$\frac{\partial^2 S}{\partial \hat{\Phi}^\ell_{\mu\nu}(t,s)\partial \hat{G}^\ell_{\alpha\beta}(t',s')} = \left\langle \phi(h^\ell_\mu(t))\phi(h^\ell_\nu(s))g^\ell_\alpha(t')g^\ell_\beta(s') \right\rangle - \Phi^\ell_{\mu\nu}(t,s)G^\ell_{\alpha\beta}(t',s')$$

$$\equiv \kappa^{\Phi^\ell G^\ell}_{\mu\nu\alpha\beta}(t,s,t',s')$$

$$\frac{\partial^2 S}{\partial \hat{\Phi}^\ell_{\mu\nu}(t,s)\partial A^{\ell-1}_{\beta\alpha}(s',t')} = -\gamma \left\langle \frac{\partial \phi(h^\ell_\mu(t))}{\partial u^\ell_\beta(s')}\phi(h^\ell_\nu(s))g^\ell_\alpha(t') \right\rangle$$

$$-\gamma \left\langle \phi(h^\ell_\mu(t))\frac{\partial \phi(h^\ell_\nu(t))}{\partial u^\ell_\beta(s')}g^\ell_\alpha(t') \right\rangle$$

$$-\gamma \left\langle \phi(h^\ell_\mu(t))\phi(h^\ell_\nu(s))\frac{\partial g^\ell_\alpha(t')}{\partial u^\ell_\beta(s')} \right\rangle - \gamma^2 \Phi^\ell_{\mu\nu}(t,s)B^{\ell-1}_{\alpha\beta}(t',s')$$

$$\equiv -\gamma \kappa^{\Phi^\ell B^{\ell-1}}_{\mu\nu\alpha\beta}(t,s)$$

$$\frac{\partial^2 S}{\partial \hat{\Phi}^\ell_{\mu\nu}(t,s)\partial B^\ell_{\beta\alpha}(s',t')} = -\gamma \left\langle \frac{\partial \phi(h^\ell_\mu(t))}{\partial r^\ell_\beta(s')}\phi(h^\ell_\nu(s))\phi(h^\ell_\alpha(t')) \right\rangle$$

$$-\gamma \left\langle \phi(h^\ell_\mu(t))\frac{\partial \phi(h^\ell_\nu(t))}{\partial r^\ell_\beta(s')}\phi(h^\ell_\alpha(t')) \right\rangle$$

$$-\gamma \left\langle \phi(h^\ell_\mu(t))\phi(h^\ell_\nu(s))\frac{\partial \phi(h^\ell_\alpha(t'))}{\partial r^\ell_\beta(s')} \right\rangle - \gamma^2 \Phi^\ell_{\mu\nu}(t,s)A^\ell_{\alpha\beta}(t',s')$$

$$\equiv -\gamma \kappa^{\Phi^\ell A^\ell}_{\mu\nu\alpha\beta}(t,s)$$

$$\frac{\partial^2 S}{\partial \hat{G}^\ell_{\mu\nu}(t,s)\partial A^{\ell-1}_{\beta\alpha}(s',t')} = -\gamma \left\langle \frac{\partial g^\ell_\mu(t)}{\partial u^\ell_\beta(s')}g^\ell_\nu(s)g^\ell_\alpha(t') \right\rangle - \gamma \left\langle g^\ell_\mu(t)\frac{\partial g^\ell_\nu(t)}{\partial u^\ell_\beta(s')}g^\ell_\alpha(t') \right\rangle$$

$$-\gamma \left\langle g^\ell_\mu(t)g^\ell_\nu(s)\frac{\partial g^\ell_\alpha(t')}{\partial u^\ell_\beta(s')} \right\rangle - \gamma^2 G^\ell_{\mu\nu}(t,s)B^{\ell-1}_{\alpha\beta}(t',s')$$

$$\equiv -\gamma \kappa^{G^\ell B^{\ell-1}}_{\mu\nu\alpha\beta}(t,s)$$

$$\frac{\partial^2 S}{\partial \hat{G}^\ell_{\mu\nu}(t,s)\partial B^\ell_{\beta\alpha}(s',t')} = -\gamma \left\langle \frac{\partial g^\ell_\mu(t)}{\partial r^\ell_\beta(s')}g^\ell_\nu(s)\phi(h^\ell_\alpha(t')) \right\rangle - \gamma \left\langle g^\ell_\mu(t)\frac{\partial g^\ell_\nu(t)}{\partial r^\ell_\beta(s')}\phi(h^\ell_\alpha(t')) \right\rangle$$

$$= -\gamma \left\langle g^\ell_\mu(t)g^\ell_\nu(s)\frac{\partial \phi(h^\ell_\alpha(t'))}{\partial r^\ell_\beta(s')} \right\rangle - \gamma^2 G^\ell_{\mu\nu}(t,s)A^\ell_{\alpha\beta}(t',s')$$

$$\equiv -\gamma \kappa^{G^\ell A^\ell}_{\mu\nu\alpha\beta}(t,s)$$

$$\frac{\partial^2 S}{\partial A^\ell_{\mu\nu}(t,s)\partial B^\ell_{\beta\alpha}(s',t')} = -\gamma^2 \delta_{\mu\alpha}\delta_{\nu\beta}\delta(t-t')\delta(s-s')$$

$$\frac{\partial^2 S}{\partial A^{\ell-1}_{\nu\mu}(s,t)\partial B^\ell_{\beta\alpha}(s',t')} = \gamma^2 \left\langle \frac{\partial^2}{\partial u^\ell_\nu(s)\partial r^\ell_\beta(s')}\left[g^\ell_\mu(t)\phi(h^\ell_\alpha(t'))\right] \right\rangle - \gamma^4 B^{\ell-1}_{\mu\nu}(t,s)A^\ell_{\alpha\beta}(t',s')$$

$$\equiv \kappa^{B^{\ell-1}A^\ell}_{\mu\nu\alpha\beta}(t,s,t',s') \tag{35}$$

Next, we consider the second derivatives involving only primal variables $\{\Phi^\ell, G^\ell, K, \Delta\}$ which all vanish

$$\frac{\partial^2 S}{\partial \Phi^\ell_{\mu\nu}(t,s)\partial \Phi^{\ell'}_{\alpha\beta}(t',s')} = 0$$

$$\frac{\partial^2 S}{\partial G^\ell_{\mu\nu}(t,s)\partial G^{\ell'}_{\alpha\beta}(t',s')} = 0$$

$$\frac{\partial^2 S}{\partial \Phi^\ell_{\mu\nu}(t,s)\partial G^{\ell'}_{\alpha\beta}(t',s')} = 0$$

$$\frac{\partial^2 S}{\partial \Phi^\ell_{\mu\nu}(t,s)\partial K_{\alpha\beta}(s')} = 0$$

$$\frac{\partial^2 S}{\partial G^\ell_{\mu\nu}(t,s)\partial K_{\alpha\beta}(s')} = 0$$

$$\frac{\partial^2 S}{\partial \Phi^\ell_{\mu\nu}(t,s)\partial \Delta_\alpha(s')} = 0$$

$$\frac{\partial^2 S}{\partial G^\ell_{\mu\nu}(t,s)\partial \Delta_\alpha(s')} = 0$$

$$\frac{\partial^2 S}{\partial K_{\mu\nu}(t)\partial K_{\alpha\beta}(s)} = 0$$

$$\frac{\partial^2 S}{\partial K_{\mu\nu}(t)\partial \Delta_\alpha(s)} = 0$$

$$\frac{\partial^2 S}{\partial \Delta_\mu(t)\partial \Delta_\alpha(s)} = 0 \tag{36}$$

Now we consider all derivatives which involve one of the dual variables $\{\hat{\Phi}^\ell, \hat{G}^\ell, A^\ell, B^\ell\}$ and the primal variable $\Delta$

$$\frac{\partial^2 S}{\partial \hat{\Phi}^\ell_{\mu\nu}(t,s)\partial \Delta_\alpha(t')} = -\left\langle \frac{\partial}{\partial \Delta_\alpha(t')}[\phi(h^\ell_\mu(t))\phi(h^\ell_\nu(s))]\right\rangle \equiv -D^{\Phi^\ell \Delta}_{\mu\nu\alpha}(t,s,t')$$

$$\frac{\partial^2 S}{\partial \hat{G}^\ell_{\mu\nu}(t,s)\partial \Delta_\alpha(t')} = -\left\langle \frac{\partial}{\partial \Delta_\alpha(t')}[g^\ell_\mu(t)g^\ell_\nu(s)]\right\rangle \equiv -D^{G^\ell \Delta}_{\mu\nu\alpha}(t,s,t')$$

$$\frac{\partial^2 S}{\partial A^{\ell-1}_{\nu\mu}(s,t)\partial \Delta_\alpha(t')} = \gamma\left\langle \frac{\partial}{\partial \Delta_\alpha(t')\partial u^\ell_\nu(s)}g^\ell_\mu(t)\right\rangle \equiv \gamma D^{B^{\ell-1},\Delta}_{\mu\nu\alpha}(t,s,t')$$

$$\frac{\partial^2 S}{\partial B^\ell_{\nu\mu}(s,t)\partial \Delta_\alpha(t')} = \gamma\left\langle \frac{\partial}{\partial \Delta_\alpha(t')\partial r^\ell_\nu(s)}\phi(h^\ell_\mu(t))\right\rangle \equiv \gamma D^{A^\ell \Delta}_{\mu\nu\alpha}(t,s,t')$$

Now, we consider the second derivatives involving one derivative on a dual variable $\{\hat{\Phi}^\ell, \hat{G}^\ell, A, B\}$ and one of the primal variables $\{\Phi^\ell, G^\ell\}$.

$$\frac{\partial^2 S}{\partial \hat{\Phi}^\ell_{\mu\nu}(t,s)\partial \Phi^{\ell'}_{\alpha\beta}(t',s')} = \delta_{\ell,\ell'}\delta_{\mu\nu}\delta(t-t')\delta(s-s')$$

$$- \delta_{\ell-1,\ell'}\frac{\partial}{\partial \Phi^{\ell-1}_{\alpha\beta}(t',s')}\left\langle \phi(h^\ell_\mu(t))\phi(h^\ell_\nu(s))\right\rangle$$

$$\equiv \delta_{\ell,\ell'}\delta_{\mu\nu}\delta(t-t')\delta(s-s') - \delta_{\ell-1,\ell'}D^{\Phi^\ell\Phi^{\ell-1}}_{\mu\nu\alpha\beta}(t,s,t',s')$$

$$\frac{\partial^2 S}{\partial \hat{G}^\ell_{\mu\nu}(t,s)\partial G^{\ell'}_{\alpha\beta}(t',s')} = \delta_{\ell,\ell'}\delta_{\mu\nu}\delta(t-t')\delta(s-s') - \delta_{\ell+1,\ell'}\frac{\partial}{\partial G^{\ell+1}_{\alpha\beta}(t',s')}\left\langle g^\ell_\mu(t)g^\ell_\nu(s)\right\rangle$$

$$\equiv \delta_{\ell,\ell'}\delta_{\mu\nu}\delta(t-t')\delta(s-s') - \delta_{\ell-1,\ell'}D^{G^\ell G^{\ell+1}}_{\mu\nu\alpha\beta}(t,s,t',s')$$

$$\frac{\partial^2 S}{\partial \hat{\Phi}^\ell_{\mu\nu}(t,s)\partial G^{\ell+1}_{\alpha\beta}(t',s')} = -\frac{\partial}{\partial G^{\ell+1}_{\alpha\beta}(t',s')}\left\langle \phi(h^\ell_\mu(t))\phi(h^\ell_\nu(s))\right\rangle \equiv -D^{\Phi^\ell,G^{\ell+1}}_{\mu\nu\alpha\beta}(t,s,t',s')$$

$$\frac{\partial^2 S}{\partial \hat{G}^\ell_{\mu\nu}(t,s)\partial \Phi^{\ell-1}_{\alpha\beta}(t',s')} = -\frac{\partial}{\partial \Phi^{\ell-1}_{\alpha\beta}(t',s')}\left\langle g^\ell_\mu(t)g^\ell_\nu(s)\right\rangle \equiv -D^{G^\ell,\Phi^{\ell-1}}_{\mu\nu\alpha\beta}(t,s,t',s')$$

$$\frac{\partial^2 S}{\partial A^{\ell-1}_{\nu\mu}(s,t)\partial \Phi^{\ell-1}_{\alpha\beta}(t',s')} = \gamma\frac{\partial}{\partial \Phi^{\ell-1}_{\alpha\beta}(t',s')}\left\langle \frac{\partial g^\ell_\mu(t)}{\partial r^\ell_\nu(s)}\right\rangle \equiv \gamma D^{B^{\ell-1},\Phi^{\ell-1}}_{\mu\nu\alpha\beta}(t,s,t',s')$$

$$\frac{\partial^2 S}{\partial B^\ell_{\nu\mu}(s,t)\partial \Phi^{\ell-1}_{\alpha\beta}(t',s')} = \gamma\frac{\partial}{\partial \Phi^{\ell-1}_{\alpha\beta}(t',s')}\left\langle \frac{\partial \phi(h^\ell_\mu(t))}{\partial u^\ell_\nu(s)}\right\rangle \equiv \gamma D^{A^\ell,\Phi^{\ell-1}}_{\mu\nu\alpha\beta}(t,s,t',s')$$

$$\frac{\partial^2 S}{\partial A^{\ell-1}_{\nu\mu}(s,t)\partial G^{\ell+1}_{\alpha\beta}(t',s')} = \gamma\frac{\partial}{\partial G^{\ell+1}_{\alpha\beta}(t',s')}\left\langle \frac{\partial g^\ell_\mu(t)}{\partial r^\ell_\nu(s)}\right\rangle \equiv \gamma D^{B^{\ell-1},G^{\ell+1}}_{\mu\nu\alpha\beta}(t,s,t',s')$$

$$\frac{\partial^2 S}{\partial B^\ell_{\nu\mu}(s,t)\partial G^{\ell+1}_{\alpha\beta}(t',s')} = \gamma\frac{\partial}{\partial G^{\ell+1}_{\alpha\beta}(t',s')}\left\langle \frac{\partial \phi(h^\ell_\mu(t))}{\partial u^\ell_\nu(s)}\right\rangle \equiv \gamma D^{A^\ell,G^{\ell+1}}_{\mu\nu\alpha\beta}(t,s,t',s') \qquad (37)$$

We note that terms such as $\frac{\partial}{\partial \Phi^{\ell-1}_{\alpha\beta}(t',s')}\left\langle \phi(h^\ell_\mu(t))\phi(h^\ell_\nu(s))\right\rangle$ can be further decomposed since the average over the $\{u^\ell_\mu(t)\} \sim \mathcal{GP}(0,\mathbf{\Phi}^{\ell-1})$ and $h^\ell$'s explicit dynamics both depend on $\Phi^{\ell-1}$

$$\frac{\partial}{\partial \Phi^{\ell-1}_{\alpha\beta}(t',s')}\left\langle \phi(h^\ell_\mu(t))\phi(h^\ell_\nu(s))\right\rangle = \frac{1}{2}\left\langle \frac{\partial^2}{\partial u^\ell_\alpha(t')\partial u^\ell_\beta(s')}\phi(h^\ell_\mu(t))\phi(h^\ell_\nu(s))\right\rangle$$

$$+ \left\langle \frac{\partial}{\partial \Phi^{\ell-1}_{\alpha\beta}(t',s')}\phi(h^\ell_\mu(t))\phi(h^\ell_\nu(s))\right\rangle \qquad (38)$$

where the first term comes from differentiating the Gaussian probability density for $u^\ell$ (e.g. Price's theorem) and the second term is an explicit derivative of the preactivation fields with $u^\ell$ treated as constant. Next we consider the nonvanishing terms which involve $\{\hat{\Delta}, \hat{K}, \Delta, K\}$ which give

$$\frac{\partial^2 S}{\partial \hat{\Delta}_\mu(t)\partial \Delta_\alpha(s)} = \delta_{\mu\alpha}\delta(t-s) + \Theta(t-s)K_{\mu\alpha}(s)$$

$$\frac{\partial^2}{\partial \hat{\Delta}_\mu(t)\partial K_{\alpha\beta}(s)} = \delta_{\mu\alpha}\Theta(t-s)\Delta_\beta(s)$$

$$\frac{\partial^2 S}{\partial \hat{K}_{\mu\nu}(t)\partial K_{\alpha\beta}(t')} = \delta_{\mu\alpha}\delta_{\nu\beta}\delta(t-t')$$

$$\frac{\partial^2 S}{\partial \hat{K}_{\mu\nu}(t)\partial \Phi^\ell_{\alpha\beta}(t',s')} = \delta_{\mu\alpha}\delta_{\nu\beta}G^{\ell+1}_{\alpha\beta}(t',s')\delta(t-t')\delta(t-s')$$

$$\frac{\partial^2 S}{\partial \hat{K}_{\mu\nu}(t)\partial G^\ell_{\alpha\beta}(t',s')} = \delta_{\mu\alpha}\delta_{\nu\beta}\Phi^{\ell-1}_{\alpha\beta}(t',s')\delta(t-t')\delta(t-s') \qquad (39)$$

This enumerates all possible non-vanishing terms in the Hessian. We can now construct a block matrix of these Hessians by partitioning our order parameters $\boldsymbol{q} = [\boldsymbol{q}_1, \boldsymbol{q}_2]^\top$ where

$$\boldsymbol{q}_1 = \text{Vec}\{\Phi^\ell_{\mu\nu}(t,s), G^\ell_{\mu\nu}(t,s), K_{\mu\nu}(t), \Delta_\mu(t), \hat{\Phi}^\ell_{\mu\nu}(t,s), \hat{G}^\ell_{\mu\nu}(t,s), \hat{K}_{\mu\nu}(t), \hat{\Delta}_\mu(t)\} \tag{40}$$

$$\boldsymbol{q}_2 = \text{Vec}\{A^\ell_{\mu\nu}(t,s), B^\ell_{\mu\nu}(t,s)\}. \tag{41}$$

This choice will become apparent shortly.

$$\nabla^2_{\boldsymbol{q}} S = \begin{bmatrix} \nabla^2_{\boldsymbol{q}_1} S & \nabla^2_{\boldsymbol{q}_1 \boldsymbol{q}_2} S \\ \nabla^2_{\boldsymbol{q}_2 \boldsymbol{q}_1} S & \nabla^2_{\boldsymbol{q}_2} S \end{bmatrix} \tag{42}$$

To calculate the full propagator $\boldsymbol{\Sigma} = -\left[\nabla^2_{\boldsymbol{q}} S\right]^{-1}$, we will assume invertibility of the upper block $\boldsymbol{\Sigma}^0 = -\left[\nabla^2_{\boldsymbol{q}_1} S\right]^{-1}$ and use this in the Schur complement

$$\boldsymbol{\Sigma} = -\left[\nabla^2_{\boldsymbol{q}} S\right]^{-1} = \begin{bmatrix} \boldsymbol{\Sigma}_{11} & \boldsymbol{\Sigma}_{12} \\ \boldsymbol{\Sigma}_{21} & \boldsymbol{\Sigma}_{22} \end{bmatrix}$$

$$\boldsymbol{\Sigma}_{11} = \boldsymbol{\Sigma}^0 - \boldsymbol{\Sigma}^0 \left[\nabla^2_{\boldsymbol{q}_1 \boldsymbol{q}_2} S\right] \left(\nabla^2_{\boldsymbol{q}_2} S + (\nabla^2_{\boldsymbol{q}_2 \boldsymbol{q}_1} S)\boldsymbol{\Sigma}^0(\nabla^2_{\boldsymbol{q}_1 \boldsymbol{q}_2} S)\right)^{-1} \left[\nabla^2_{\boldsymbol{q}_2 \boldsymbol{q}_1} S\right] \boldsymbol{\Sigma}^0$$

$$\boldsymbol{\Sigma}_{12} = \boldsymbol{\Sigma}_{21}^\top = -\boldsymbol{\Sigma}^0 \left[\nabla^2_{\boldsymbol{q}_1 \boldsymbol{q}_2} S\right] \left(\nabla^2_{\boldsymbol{q}_2} S + (\nabla^2_{\boldsymbol{q}_2 \boldsymbol{q}_1} S)\boldsymbol{\Sigma}^0(\nabla^2_{\boldsymbol{q}_1 \boldsymbol{q}_2} S)\right)^{-1}$$

$$\boldsymbol{\Sigma}_{22} = -\left(\nabla^2_{\boldsymbol{q}_2} S + (\nabla^2_{\boldsymbol{q}_2 \boldsymbol{q}_1} S)\boldsymbol{\Sigma}^0(\nabla^2_{\boldsymbol{q}_1 \boldsymbol{q}_2} S)\right)^{-1} \tag{43}$$

We now need to solve for $\boldsymbol{\Sigma}^0 = -\left[\nabla^2_{\boldsymbol{q}_1} S\right]^{-1}$. To perform this inverse, we again partition $\boldsymbol{q}_1$ into two sets of order parameters $\boldsymbol{q}_1 = [\boldsymbol{q}_1^1, \boldsymbol{q}_1^2]$ where $\boldsymbol{q}_1^1 = \text{Vec}\{\Phi^\ell_{\mu\nu}(t,s), G^\ell_{\mu\nu}(t,s), K_{\mu\nu}(t), \Delta_\mu(t)\}$ and $\boldsymbol{q}_1^2 = \text{Vec}\{\hat{\Phi}^\ell_{\mu\nu}(t,s), \hat{G}^\ell_{\mu\nu}(t,s), \hat{K}_{\mu\nu}(t), \hat{\Delta}_\mu(t)\}$

$$\nabla^2_{\boldsymbol{q}_1} S = \begin{bmatrix} \boldsymbol{0} & \boldsymbol{U}^\top \\ \boldsymbol{U} & \boldsymbol{\kappa} \end{bmatrix}, \; \boldsymbol{\kappa} \equiv \nabla^2_{\boldsymbol{q}_1^2} S, \; \boldsymbol{U} \equiv \nabla^2_{\boldsymbol{q}_1^2 \boldsymbol{q}_1^1} S \tag{44}$$

We seek a physically sensible inverse where the variance of $\boldsymbol{q}_1^2$ is vanishing [51, 53]. This leads to the following sub-propagator $\boldsymbol{\Sigma}^0$

$$\boldsymbol{\Sigma}^0 = -[\nabla^2_{\boldsymbol{q}_1} S]^{-1} = \begin{bmatrix} \boldsymbol{U}^{-1}\boldsymbol{\kappa}[\boldsymbol{U}^{-1}]^\top & -\boldsymbol{U}^{-1} \\ -[\boldsymbol{U}^\top]^{-1} & \boldsymbol{0} \end{bmatrix} \tag{45}$$

Thus given $\boldsymbol{\kappa}, \boldsymbol{U}$, we can solve for $\boldsymbol{\Sigma}^0$ and ultimately for the full propagator $\boldsymbol{\Sigma}$. The relevant entries in $\boldsymbol{\kappa}$ and $\boldsymbol{U}$ are given by those second derivatives calculated above. We note that each of the field derivatives needed for $\boldsymbol{U}$ can be computed implicitly from the field dynamics. For example, for the $\Delta_\mu(t)$ derivatives we have

$$\frac{\partial}{\partial \Delta_{\nu'}(t')} h^\ell_\mu(t) = \gamma \Theta(t - t') \Phi^{\ell-1}_{\mu\nu'}(t,t') g^\ell_\nu(t')$$

$$+ \gamma \int_0^t ds \sum_\nu \left[A^{\ell-1}_{\mu\nu}(t,s) + \Phi^{\ell-1}_{\mu\nu}(t,s)\Delta_\nu(s)\right] \frac{\partial g^\ell_\nu(s)}{\partial \Delta_{\nu'}(t')}$$

$$\frac{\partial}{\partial \Delta_\nu(t')} z^\ell_\mu(t) = \gamma \Theta(t - t') G^{\ell+1}_{\mu\nu}(t,t') \phi(h^\ell_\nu(t'))$$

$$+ \gamma \int_0^t ds \sum_\nu \left[B^\ell_{\mu\nu}(t,s) + G^{\ell+1}_{\mu\nu}(t,s)\Delta_\nu(s)\right] \frac{\partial \phi(h^\ell_\nu(s))}{\partial \Delta_{\nu'}(t')} \tag{46}$$

These can then be used in the averages such as $\left\langle \frac{\partial}{\partial \Delta_{\nu'}(t')} \phi(h^\ell_\mu(t))\phi(h^\ell_\nu(s)) \right\rangle$. Similarly, we can compute terms such as $\frac{\partial h^\ell_\mu(t)}{\partial \Phi^\ell_{\alpha\beta}(t',s')}$ through the following closed equations

$$\frac{\partial h^\ell_\mu(t)}{\partial \Phi^{\ell-1}_{\alpha\beta}(t',s')} = \gamma \delta(t - t')\delta_{\mu\alpha}\Theta(t - s')\Delta_\beta(s')$$

$$+ \gamma \int_0^t ds \sum_\nu \left[A^{\ell-1}_{\mu\nu}(t,s) + \Delta_\nu(s)\Phi^{\ell-1}_{\mu\nu}(t,s)\right] \frac{\partial g^\ell_\nu(s)}{\partial \Phi^\ell_{\alpha\beta}(t',s')}$$

$$\frac{\partial z^\ell_\mu(t)}{\partial \Phi^{\ell-1}_{\alpha\beta}(t',s')} = \gamma \int_0^t ds \sum_\nu \left[B^\ell_{\mu\nu}(t,s) + \Delta_\nu(s)G^{\ell+1}_{\mu\nu}(t,s)\right] \frac{\partial \phi(h^\ell_\nu(s))}{\partial \Phi^{\ell-1}_{\alpha\beta}(t',s')} \tag{47}$$

These terms can then be used to compute quantities like $D^{\Phi^\ell}$.

# F Solving for the Propagator

In this section we sketch out the required steps to obtain the propagator $\boldsymbol{\Sigma}$.

- Step 1: Solve the infinite width DMFT equations for $q_\infty$ which include the prediction error dynamics $\Delta_\mu(t)$, the feature kernels $\Phi_{\mu\nu}^\ell(t,s)$, gradient kernels $G_{\mu\nu}^\ell(t,s)$. This step corresponds to algorithm in Bordelon & Pehlevan '22 and defines the dynamics one would expect at infinite width [9]. See below for more detail.
- Step 2: Compute the entries of the Hessian of $S$ evaluated at the $q_\infty$ computed in the first step. Some of these entries look like fourth cumulants of features like $\kappa = \left\langle \phi(h)^4 \right\rangle - \left\langle \phi(h)^2 \right\rangle^2$ and some of them measure sensitivity of one order parameter to a perturbation in another order parameter $D^{\Phi^\ell} = \frac{\partial}{\partial \Phi^{\ell-1}} \left\langle \phi(h^\ell)^2 \right\rangle$. The averages $\langle \rangle$ used to calculate $\kappa$ and $D^{\Phi^\ell}$ should be performed over the infinite width stochastic processes for preactivations $h^\ell$ which are defined in equation (19).
- Step 3: After populating the entries of the block matrix for the Hesssian $\nabla^2 S$, we then calculate the propagator $\Sigma$ with a matrix inversion. Since we discretized time, this is a finite dimensional matrix.

The step 1 above demands a solution to the infinite width DMFT equations (solving for the saddle point $\boldsymbol{q}_\infty$). We will now give a detailed set of instructions about how the infinite width limit for $q_\infty$ is solved (step 1 above). This corresponds to the algorithm of Bordelon & Pehlevan 2022 to solve the saddle point equations $\frac{\partial}{\partial \boldsymbol{q}} S(\boldsymbol{q})|_{\boldsymbol{q}_\infty} = 0$ [9].

- Step 1: Start with a guess for the kernels $\Phi_{\mu\nu}^\ell(t,s), G_{\mu\nu}^\ell(t,s)$ and for the predictions through time $f_\mu(t)$. We usually use the lazy limit (e.g. $\Phi_{\mu\nu}^\ell(t,s) = \Phi_{\mu\nu}^\ell(0,0)$ ...) as an initial guess.
- Step 2: Sample Gaussian sources $u_\mu^\ell(t)$ and $r_\mu^\ell(t)$ based on the current covariances $\Phi^\ell$ and $G^\ell$.
- Step 3: For each sample, solve integral equations for $h(t)$ and $z(t)$.

$$h_\mu^\ell(t) = u_\mu^\ell(t) + \gamma \int_0^t ds \sum_\nu [A_{\mu\nu}^{\ell-1}(t,s) + \Phi_{\mu\nu}^{\ell-1}(t,s)][\dot{\phi}(h_\nu^\ell(s))z_\nu^\ell(s)]$$

$$z_\mu^\ell(t) = r_\mu^\ell(t) + \gamma \int_0^t ds \sum_\nu [B_{\mu\nu}^\ell(t,s) + G_{\mu\nu}^{\ell+1}(t,s)]\phi(h_\nu^\ell(s)) \qquad (48)$$

These will be samples from the single site distribution for $h,z$

- Step 4: Average over the Monte Carlo samples to produce a new estimate of the kernels: $\Phi^\ell(t,s) = \left\langle \phi(h^\ell(t))\phi(h^\ell(s)) \right\rangle$. A similar procedure is performed for $G^\ell$ and the response functions $A^\ell, B^\ell$.
- Step 5: Compute the NTK estimate $K(t) = \sum_\ell G^{\ell+1}(t,t)\Phi^\ell(t,t)$ and then integrate prediction dynamics from the dynamics of the NTK $\frac{d}{dt}f_\mu(t) = \sum_\nu K_{\mu\nu}(t)\Delta_\nu(t)$.
- Repeat steps 2-5 until the order parameters converge.

Below we provide a pseudocode algorithm to solve for the propagator elements.

---

**Algorithm 1:** Propagator Solver

---

**Data:** $\boldsymbol{K}^x, \boldsymbol{y}$, Initial Guesses $\{\boldsymbol{\Phi}^\ell, \boldsymbol{G}^\ell\}_{\ell=1}^L$, $\{\boldsymbol{A}^\ell, \boldsymbol{B}^\ell\}_{\ell=1}^{L-1}$, Sample count $\mathcal{S}$, Update Speed $\beta$
**Result:** Propagator Matrix $\boldsymbol{\Sigma}$

1 Solve DMFT equations with Algorithm 2 for order parameters $f_\mu(t), \Phi_{\mu\alpha}^\ell(t,s), ...$ ;
2 Draw $\mathcal{S}$ samples $\{u_{\mu,n}^\ell(t)\}_{n=1}^{\mathcal{S}} \sim \mathcal{GP}(0, \boldsymbol{\Phi}^{\ell-1})$, $\{r_{\mu,n}^\ell(t)\}_{n=1}^{\mathcal{S}} \sim \mathcal{GP}(0, \boldsymbol{G}^{\ell+1})$;
3 Integrate dynamics for each sample to get $\{h_{\mu,n}^\ell(t), z_{\mu,n}^\ell(t)\}_{n=1}^{\mathcal{S}}$;
4 Estimate $\kappa$ functions with Monte Carlo integration, for instance
5 $\kappa_{\mu\nu\alpha\beta}^{\Phi^\ell}(t,s,t',s') =$
  $\frac{1}{\mathcal{S}} \sum_{n \in [\mathcal{S}]} \phi(h_{\mu,n}^\ell(t))\phi(h_{\nu,n}^\ell(s))\phi(h_{\alpha,n}^\ell(t'))\phi(h_{\beta,n}^\ell(s')) - \Phi_{\mu\nu}^\ell(t,s)\Phi_{\alpha\beta}^\ell(t',s')$ ;
6 For each sample, compute field sensitivities to error signals, such as $\frac{\partial h_{\mu,n}^\ell(t)}{\partial \Delta_\nu(s)}$, and kernels
  $\frac{\partial h_{\mu,n}^\ell(t)}{\partial \Phi_{\alpha\beta}^\ell(t',s')}$ implicitly using equations (46) (47) ;
7 Use these sensitivities to compute the necessary $D$ tensors such as
  $D_{\mu\nu\alpha}^{\Phi^\ell \Delta} = \frac{1}{\mathcal{S}} \sum_{n \in [\mathcal{S}]} \frac{\partial}{\partial \Delta_\alpha(t')} \left[ \phi(h_{\mu,n}^\ell(t))\phi(h_{\nu,n}^\ell(s)) \right]$;
8 Invert $\boldsymbol{U}$ matrix and compute $\boldsymbol{\Sigma}_0$ in equation (45);
9 Compute the Schur-complement in equation (43) to handle the response functions ;

---

The above propagator solver builds on the solution to the DMFT equations which is provided below.

---

**Algorithm 2:** Alternating Monte Carlo Solution to Saddle Point Equations

---

**Data:** $\boldsymbol{K}^x, \boldsymbol{y}$, Initial Guesses $\{\boldsymbol{\Phi}^\ell, \boldsymbol{G}^\ell\}_{\ell=1}^L, \{\boldsymbol{A}^\ell, \boldsymbol{B}^\ell\}_{\ell=1}^{L-1}$, Sample count $\mathcal{S}$, Update Speed $\beta$

**Result:** Final Kernels $\{\boldsymbol{\Phi}^\ell, \boldsymbol{G}^\ell\}_{\ell=1}^L, \{\boldsymbol{A}^\ell, \boldsymbol{B}^\ell\}_{\ell=1}^{L-1}$, Network predictions through training $f_\mu(t)$

1 $\boldsymbol{\Phi}^0 = \boldsymbol{K}^x \otimes \boldsymbol{1}\boldsymbol{1}^\top, \boldsymbol{G}^{L+1} = \boldsymbol{1}\boldsymbol{1}^\top$ ;
2 **while** *Kernels Not Converged* **do**
3     From $\{\boldsymbol{\Phi}^\ell, \boldsymbol{G}^\ell\}$ compute $\boldsymbol{K}^{NTK}(t,t)$ and solve $\frac{d}{dt} f_\mu(t) = \sum_\alpha \Delta_\alpha(t) K_{\mu\alpha}^{NTK}(t,t)$;
4     $\ell = 1$;
5     **while** $\ell < L+1$ **do**
6        Draw $\mathcal{S}$ samples $\{u_{\mu,n}^\ell(t)\}_{n=1}^{\mathcal{S}} \sim \mathcal{GP}(0, \boldsymbol{\Phi}^{\ell-1}), \{r_{\mu,n}^\ell(t)\}_{n=1}^{\mathcal{S}} \sim \mathcal{GP}(0, \boldsymbol{G}^{\ell+1})$;
7        Integrate dynamics for each sample to get $\{h_{\mu,n}^\ell(t), z_{\mu,n}^\ell(t)\}_{n=1}^{\mathcal{S}}$;
8        Compute new $\boldsymbol{\Phi}^\ell, \boldsymbol{G}^\ell$ estimates:
9        $\tilde{\Phi}_{\mu\alpha}^\ell(t,s) = \frac{1}{\mathcal{S}} \sum_{n\in[\mathcal{S}]} \phi(h_{\mu,n}^\ell(t)) \phi(h_{\alpha,n}^\ell(s)), \tilde{G}_{\mu\alpha}^\ell(t,s) = \frac{1}{\mathcal{S}} \sum_{n\in[\mathcal{S}]} g_{\mu,n}^\ell(t) g_{\alpha,n}^\ell(s)$ ;
10        Solve for Jacobians on each sample $\frac{\partial \phi(\boldsymbol{h}_n^\ell)}{\partial \boldsymbol{r}_n^{\ell\top}}, \frac{\partial \boldsymbol{g}_n^\ell}{\partial \boldsymbol{u}_n^{\ell\top}}$ ;
11        Compute new $\boldsymbol{A}^\ell, \boldsymbol{B}^{\ell-1}$ estimates:
12        $\tilde{\boldsymbol{A}}^\ell = \frac{1}{\mathcal{S}} \sum_{n\in[\mathcal{S}]} \frac{\partial \phi(\boldsymbol{h}_n^\ell)}{\partial \boldsymbol{r}_n^{\ell\top}}, \tilde{\boldsymbol{B}}^{\ell-1} = \frac{1}{\mathcal{S}} \sum_{n\in[\mathcal{S}]} \frac{\partial \boldsymbol{g}_n^\ell}{\partial \boldsymbol{u}_n^{\ell\top}}$ ;
13        $\ell \leftarrow \ell + 1$;
14     **end**
15     $\ell = 1$;
16     **while** $\ell < L+1$ **do**
17        Update feature kernels: $\boldsymbol{\Phi}^\ell \leftarrow (1-\beta)\boldsymbol{\Phi}^\ell + \beta\tilde{\boldsymbol{\Phi}}^\ell, \boldsymbol{G}^\ell \leftarrow (1-\beta)\boldsymbol{G}^\ell + \beta\tilde{\boldsymbol{G}}^\ell$ ;
18        **if** $\ell < L$ **then**
19           Update $\boldsymbol{A}^\ell \leftarrow (1-\beta)\boldsymbol{A}^\ell + \beta\tilde{\boldsymbol{A}}^\ell, \boldsymbol{B}^\ell \leftarrow (1-\beta)\boldsymbol{B}^\ell + \beta\tilde{\boldsymbol{B}}^\ell$
20        **end**
21        $\ell \leftarrow \ell + 1$
22     **end**
23 **end**
24 **return** $\{\boldsymbol{\Phi}^\ell, \boldsymbol{G}^\ell\}_{\ell=1}^L, \{\boldsymbol{A}^\ell, \boldsymbol{B}^\ell\}_{\ell=1}^{L-1}, \{f_\mu(t)\}_{\mu=1}^P$

---

## G   Leading Correction to the Mean Order Parameters

In this section we use the propagator structure derived in the last section to reason about the leading finite size correction to $\langle \boldsymbol{q} \rangle$ at width $N$. Letting the indices $i, j, k, n$ enumerate all entries of the order parameters in $\boldsymbol{q}$ (technically this is a sum over samples and an integral over time for gradient flow), we find the leading Pade Approximant for the mean has the form (App D)

$$\langle q_i - q_i^\infty \rangle_N = \frac{N \langle (q_i - q_i^\infty)V \rangle_\infty + \frac{N^2}{2} \langle (q_i - q_i^\infty)V^2 \rangle_\infty \cdots}{1 + N \langle V \rangle_\infty + \frac{N^2}{2} \langle V^2 \rangle_\infty + \ldots}$$

$$\sim \frac{1}{3! N} \sum_{jkl} \frac{\partial^3 S}{\partial q_j \partial q_k \partial q_l} \langle \delta_i \delta_j \delta_k \delta_l \rangle_\infty + \mathcal{O}(N^{-2}). \tag{49}$$

$$= \frac{1}{2N} \sum_{jkl} \frac{\partial^3 S}{\partial q_j \partial q_k \partial q_l} \Sigma_{ij} \Sigma_{kl} + \mathcal{O}(N^{-2}) \tag{50}$$

where $\delta_j = \sqrt{N}(q_j - q_j^\infty)$ and the derivatives are computed at the saddle point. In the last line, we utilized Wick's theorem and the permutation symmetry of the third derivative $\frac{\partial^3 S}{\partial q_i \partial q_j \partial q_k}$ to evaluate the four point averages in terms of the propagator $\Sigma_{ij}$, which was provided in the preceding section E. In practice computing even the full set of second derivatives for the DMFT action to get $\Sigma$ is quite challenging. Despite the challenge of computing the mean order parameter correction, these corrections are relevant in practice and crucially distinguish the training timescales of deep networks at different widths as we show in Figures 7 and A.4.

### G.1 Correction to Mean Predictions and Full MSE Correction

Supposing that we solved for the propagator $\Sigma$, using the formalism in the preceeding section, we can compute the $\mathcal{O}(N^{-1})$ correction to the average network prediction error due to finite size. We let $\langle \Delta(t) \rangle$ represent the average of errors over an ensemble of width $N$ networks.

$$
\begin{aligned}
\frac{d}{dt} \langle \Delta_\mu(t) \rangle &= -\sum_\nu \langle K_{\mu\nu}(t) \Delta_\nu(t) \rangle \\
&= -\sum_\nu \langle K_{\mu\nu}(t) \rangle \langle \Delta_\nu(t) \rangle - \sum_\nu \mathrm{Cov}\left(K_{\mu\nu}(t), \Delta_\nu(t)\right) \\
&\sim -\sum_\nu \langle K_{\mu\nu}(t) \rangle \langle \Delta_\nu(t) \rangle - \frac{1}{N} \sum_\nu \Sigma^{K\Delta}_{\mu\nu\nu}(t,t) + \mathcal{O}(N^{-2})
\end{aligned}
\tag{51}
$$

where $\Sigma^{K\Delta}_{\mu\nu\nu}(t,t)$ is the leading covariance (propagator element) between the kernel $K_{\mu\nu}(t)$ and prediction error $\Delta_\nu(t)$. We see that the average kernel $\langle K_{\mu\nu}(t) \rangle$ (which depends on the finite width $N$) plays an important role in characterizing the timescales of the average prediction dynamics. Once this equation is solved for $\langle \Delta_\mu(t) \rangle$, the square loss at width $N$ and time $t$ has the form

$$
\sum_\mu \langle \Delta_\mu(t)^2 \rangle \sim \left(1 - \frac{2}{N}\right) \sum_\mu \Delta_\mu^\infty(t)^2 + \frac{2}{N} \sum_\mu \langle \Delta_\mu(t) \rangle_\infty \Delta_\mu^\infty(t) + \frac{1}{N} \sum_\mu \Sigma^\Delta_{\mu\mu}(t,t) + \mathcal{O}(N^{-2})
\tag{52}
$$

We will now comment on the structure of the cross term in this above solution. First, if $\langle K \rangle \succeq K^\infty$ and $\Sigma^{K\Delta}$ is negligible then the average errors at finite width will decay more rapidly than the infinite width model. However, we suspect that in general, $\langle K \rangle - K^\infty$ contains many negative eigenvalues since signal propagation at finite width tends to reduce the scale of feature kernels [14]. We suspect that this is the cause of the slower dynamics of ensembled predictors for narrower networks in Figure 7 and Figure A.4. Additionally, the term involving $\Sigma^{K\Delta}$ will generically increase the cross term since the dynamics of $\Delta$ cause its fluctuations to become anti-correlated with the fluctuations in $K$. In general, it is challenging to make strong definitive statements about the relative scale of these competing effects on the cross term. However, we can say more about this solution in the lazy limit, where we find that the cross term will generically be positive, leading to larger MSE (Appendix H.2).

### G.2 Perturbation Theory in Rates rather than Predictions

In experiments on deep CNNs trained on CIFAR-10 in 7 and A.4, we find that the loss curves for the ensemble averaged predictors are effectively time rescaled by a function of network width. In this section, we argue that a proper way to account for this is to compute a perturbation expansion in the *exponent* which defines the rate of decay of the training errors. To illustrate the point, we first consider the case of a single training example before describing larger datasets. In this case, we consider the change of variables $\Delta(t) = e^{-r(t)}y$. We now treat $r$ as an order parameter of the theory with dynamics

$$
\frac{d}{dt} r(t) = K(t)
\tag{53}
$$

Note that this equation is now a linear relation between two order parameters $(r(t), K(t))$, whereas the relation was previously quadratic. In the lazy limit, if $K \to K - \epsilon$ then $r \to r - \epsilon t$, giving an effective rescaling of training time by $1 - \frac{\epsilon}{K}$.

For multiple training examples, we introduce the notion of a transition matrix $\boldsymbol{T}(t) \in \mathbb{R}^{P \times P}$ which has dynamics

$$
\frac{d}{dt} \boldsymbol{T}(t) = -\boldsymbol{K}(t)\boldsymbol{T}(t) \,, \ \boldsymbol{T}(0) = \mathbf{I}.
\tag{54}
$$

The solution to the training prediction errors can be obtained at any time $t$ by multiplying the initial condition $\boldsymbol{\Delta}(0) = \boldsymbol{y}$ with the transition matrix $\boldsymbol{\Delta}(t) = \boldsymbol{T}(t)\boldsymbol{y}$, where $\boldsymbol{y}$ are the training targets. In this case, the relevant *rate matrix*, which would be an alternative order parameter is

$$
\boldsymbol{R}(t) = -\log \boldsymbol{T}(t)
\tag{55}
$$

where $\log$ is the matrix logarithm function. Note that in general $\boldsymbol{T}(t)$ admits a Peano-Baker series solution [62–64]. In the special case where $\boldsymbol{K}(t)$ commutes with $\bar{\boldsymbol{K}}(t) = \frac{1}{t}\int_0^t ds\,\boldsymbol{K}(s)$, we obtain the following simplified formula for the rate matrix $\boldsymbol{R}$

$$\boldsymbol{R}(t) = \int_0^t ds\ \boldsymbol{K}(s) \tag{56}$$

The benefit of this representation is the elimination of coupled order parameter dynamics which are quadratic in fluctuations (in $\boldsymbol{\Delta}$ and $\boldsymbol{K}$) into a linear dynamical relation between order parameters $\boldsymbol{R}$ and $\boldsymbol{K}$. An expansion in $\boldsymbol{R}$ will thus give better predictions at long times $t$ than a direct expansion in $\boldsymbol{\Delta}$. In the lazy $\gamma \to 0$ limit, the constancy of $\boldsymbol{K}(t) = \boldsymbol{K}$ gives the further simplification $\boldsymbol{R} = \boldsymbol{K}t$. Working with this representation, we have the following finite width expression for the training loss

$$\begin{aligned}
\left\langle |\boldsymbol{\Delta}(t)|^2 \right\rangle &= \boldsymbol{y}^\top \left\langle \exp\left(-2\boldsymbol{R}(t)\right) \right\rangle \boldsymbol{y} \\
&\sim \boldsymbol{y}^\top \exp\left(-2\left(\boldsymbol{R}_\infty(t) + \frac{1}{N}\boldsymbol{R}^1(t)\right)\right)\boldsymbol{y} \\
&+ \frac{1}{2}\sum_{\mu\nu\alpha\beta} \Sigma_{\mu\nu\alpha\beta}^R(t,t)\frac{\partial^2}{\partial R_{\mu\nu}\partial R_{\alpha\beta}}\boldsymbol{y}^\top \exp\left(-2\boldsymbol{R}\right)\boldsymbol{y}|_{\boldsymbol{R}=\boldsymbol{R}_\infty(t)+\frac{1}{N}\boldsymbol{R}^1(t)} + \mathcal{O}(N^{-2})
\end{aligned} \tag{57}$$

where $\langle \boldsymbol{R} \rangle \sim \boldsymbol{R}_\infty + \frac{1}{N}\boldsymbol{R}^1 + \mathcal{O}(N^{-2})$ is the leading correction to the mean $\boldsymbol{R}$. In this representation, it is clear that finite width can alter the timescale of the dynamics through a correction to the mean of $\boldsymbol{R}$, as well as contribute an additive correction from fluctuations. This justifies the study perturbation analysis of rates $R_N$ as a function of $1/N$ in Figures 7 and A.4.

## H  Variance in the Lazy Limit

We can simplify the propagator equations in the lazy $\gamma \to 0$ limit. To demonstrate how to use our formalism, we go through the complete process of inverting the Hessian, however, for this case, this procedure is a bit cumbersome. A simplified derivation for the lazy limit can be found below in section H.1 which relies only on linearizing the dynamics around the infinite width solution. In the $\gamma \to 0$ limit, all of the $D$ tensors vanish and the $\kappa$ tensors are constant in time. Thus, it suffices to analyze the kernels restricted to $t = 0$ and study the evolution of the prediction variance $\boldsymbol{\Delta}(t)$.

$$\begin{aligned}
S &= \int dt \sum_\mu \hat{\Delta}_\mu(t)\left(\Delta_\mu(t) - y_\mu + \int ds \sum_\nu \Theta(t-s)K_{\mu\nu}\Delta_\nu(s)\right) \\
&+ \sum_\ell \sum_{\mu\nu}\left[\hat{\Phi}_{\mu\nu}^\ell \Phi_{\mu\nu}^\ell + G_{\mu\nu}^\ell \hat{G}_{\mu\nu}^\ell\right] + \sum_{\mu\nu}\hat{K}_{\mu\nu}\left[K_{\mu\nu} - \sum_\ell G_{\mu\nu}^{\ell+1}\Phi_{\mu\nu}^\ell\right] + \sum_\ell \ln \mathcal{Z}_\ell \\
\mathcal{Z}_\ell &= \mathbb{E}_{\{u_\mu^\ell\},\{r_\mu^\ell\}} \exp\left(-\sum_{\mu\nu}\hat{\Phi}_{\mu\nu}^\ell \phi(u_\mu^\ell)\phi(u_\nu^\ell) - \sum_{\mu\nu}\hat{G}_{\mu\nu}^\ell g_\mu^\ell g_\nu^\ell\right), \quad g_\mu^\ell = r_\mu^\ell \dot{\phi}(u_\mu^\ell)
\end{aligned} \tag{58}$$

where $\{u_\mu^\ell\} \sim \mathcal{N}(0, \boldsymbol{\Phi}^{\ell-1}), \{r_\mu^\ell\} \sim \mathcal{N}(0, \boldsymbol{G}^{\ell+1})$. Taking two derivatives with respect to $\{\hat{\Phi}^\ell, \hat{G}^\ell\}$ give terms of the form

$$\begin{aligned}
\kappa_{\mu\nu\alpha\beta}^{\Phi^\ell} &= \left\langle \phi(u_\mu^\ell)\phi(u_\nu^\ell)\phi(u_\alpha^\ell)\phi(u_\beta^\ell)\right\rangle - \Phi_{\mu\nu}^\ell \Phi_{\alpha\beta}^\ell \\
\kappa_{\mu\nu\alpha\beta}^{G^\ell} &= \left\langle g_\mu^\ell g_\nu^\ell g_\alpha^\ell g_\beta^\ell\right\rangle - G_{\mu\nu}^\ell G_{\alpha\beta}^\ell \\
\kappa_{\mu\nu\alpha\beta}^{\Phi^\ell,G^\ell} &= \left\langle \phi(u_\mu^\ell)\phi(u_\nu^\ell)g_\alpha^\ell g_\beta^\ell\right\rangle - \Phi_{\mu\nu}^\ell G_{\alpha\beta}^\ell
\end{aligned} \tag{59}$$

Given these we also have the relevant non-vanishing sensitivity tensors

$$D_{\mu\nu\alpha\beta}^{\Phi^{\ell+1}\Phi^{\ell}} = \frac{\partial^2}{\partial\Phi_{\alpha\beta}^{\ell}}\left\langle\phi(u_\mu^{\ell+1})\phi(u_\nu^{\ell+1})\right\rangle \;,\; D_{\mu\nu\alpha\beta}^{G^{\ell}G^{\ell+1}} = \frac{\partial}{\partial G_{\alpha\beta}^{\ell+1}}\left\langle g_\mu^\ell g_\nu^\ell\right\rangle$$

$$D_{\mu\nu\alpha\beta}^{G^{\ell}\Phi^{\ell-1}} = \frac{\partial}{\partial\Phi_{\alpha\beta}^{\ell-1}}\left\langle g_\mu^\ell g_\nu^\ell\right\rangle \tag{60}$$

$$D_{\mu\nu\alpha\beta}^{K\Phi^{\ell}} = \delta_{\mu\alpha}\delta_{\nu\beta}G_{\mu\nu}^{\ell+1} \;,\; D_{\mu\nu\alpha\beta}^{KG^{\ell}} = \delta_{\mu\alpha}\delta_{\nu\beta}\Phi_{\mu\nu}^{\ell-1}$$

$$D_{\mu\alpha\beta}^{\Delta K}(t) = \int ds\Theta(t-s)\delta_{\mu\alpha}\Delta_\beta(s) \tag{61}$$

As before we let $q_1 = \mathrm{Vec}\{\Delta_\mu(t), \Phi_{\mu\nu}^\ell, G_{\mu\nu}^\ell, K_{\mu\nu}\}$ and $q_2 = \mathrm{Vec}\{\hat{\Delta}_\mu(t), \hat{\Phi}_{\mu\nu}^\ell, \hat{G}_{\mu\nu}^\ell, \hat{K}_{\mu\nu}\}$. The propagator has the form

$$U \equiv \nabla_{q_2 q_1}^2 S = \begin{bmatrix} \mathbf{I}+\boldsymbol{\Theta}_K & \mathbf{0} & \mathbf{0} & D^{\Delta K} \\ \mathbf{0} & \mathbf{I}-D^{\Phi,\Phi} & \mathbf{0} & \mathbf{0} \\ \mathbf{0} & -D^{G\Phi} & \mathbf{I}-D^{GG} & \mathbf{0} \\ \mathbf{0} & -D^{K\Phi} & -D^{KG} & \mathbf{I} \end{bmatrix}, \quad \nabla_{q_2 q_2}^2 S = \begin{bmatrix} \mathbf{0} & \mathbf{0} & \mathbf{0} & \mathbf{0} \\ \mathbf{0} & \boldsymbol{\kappa}^{\Phi,\Phi} & \boldsymbol{\kappa}^{\Phi G} & \mathbf{0} \\ \mathbf{0} & \boldsymbol{\kappa}^{G\Phi} & \boldsymbol{\kappa}^{GG} & \mathbf{0} \\ \mathbf{0} & \mathbf{0} & \mathbf{0} & \mathbf{0} \end{bmatrix}$$

$$\tag{62}$$

The propagator of interest is $\boldsymbol{\Sigma}_{q_1} = U^{-1}\left[\nabla_{q_2 q_2}^2 S\right]U^{-1\top}$. We can exploit the block structure of $U$ to find an inverse

$$U^{-1} = \begin{bmatrix} U_{\Delta\Delta}^{-1} & U_{\Delta\Phi}^{-1} & U_{\Delta G}^{-1} & U_{\Delta K}^{-1} \\ \mathbf{0} & U_{\Phi\Phi}^{-1} & \mathbf{0} & \mathbf{0} \\ \mathbf{0} & U_{G\Phi}^{-1} & U_{GG}^{-1} & \mathbf{0} \\ \mathbf{0} & U_{K\Phi}^{-1} & U_{KG}^{-1} & \mathbf{I} \end{bmatrix} \tag{63}$$

where each sub-block can be computed with the Schur-complement formula. Altogether, we multiply through to get the propagator

$$\boldsymbol{\Sigma} = \begin{bmatrix} \mathbf{0} & U_{\Delta\Phi}^{-1}\boldsymbol{\kappa}^{\Phi\Phi}+U_{\Delta G}^{-1}\boldsymbol{\kappa}^{G\Phi} & U_{\Delta\Phi}^{-1}\boldsymbol{\kappa}^{\Phi G}+U_{\Delta G}^{-1}\boldsymbol{\kappa}^{GG} & \mathbf{0} \\ \mathbf{0} & U_{\Phi\Phi}^{-1}\boldsymbol{\kappa}^{\Phi\Phi} & U_{\Phi\Phi}^{-1}\boldsymbol{\kappa}^{\Phi G} & \mathbf{0} \\ \mathbf{0} & U_{G\Phi}^{-1}\boldsymbol{\kappa}^{\Phi\Phi}+U_{GG}^{-1}\boldsymbol{\kappa}^{G\Phi} & U_{G\Phi}^{-1}\boldsymbol{\kappa}^{\Phi G}+U_{GG}^{-1}\boldsymbol{\kappa}^{GG} & \mathbf{0} \\ \mathbf{0} & U_{K\Phi}^{-1}\boldsymbol{\kappa}^{\Phi\Phi}+U_{KG}^{-1}\boldsymbol{\kappa}^{G\Phi} & U_{K\Phi}^{-1}\boldsymbol{\kappa}^{\Phi G}+U_{KG}^{-1}\boldsymbol{\kappa}^{GG} & \mathbf{0} \end{bmatrix}$$

$$\times \begin{bmatrix} U_{\Delta\Delta}^{-1} & \mathbf{0} & \mathbf{0} & \mathbf{0} \\ [U_{\Delta\Phi}^{-1}]^\top & U_{\Phi\Phi}^{-1} & [U_{G\Phi}^{-1}]^\top & [U_{K\Phi}^{-1}]^\top \\ [U_{\Delta G}^{-1}]^\top & \mathbf{0} & U_{GG}^{-1} & [U_{KG}^{-1}]^\top \\ [U_{\Delta K}^{-1}]^\top & \mathbf{0} & \mathbf{0} & \mathbf{I} \end{bmatrix} \tag{64}$$

Two of these blocks corresponding to $K, \Delta$ are especially important for characterizing the fluctuations of network predictions. The covariance structure for $K$ has the form

$$\boldsymbol{\Sigma}_K = U_{K\Phi}^{-1}\boldsymbol{\kappa}^{\Phi\Phi}[U_{K\Phi}^{-1}]^\top + U_{KG}^{-1}\boldsymbol{\kappa}^{G\Phi}[U_{K\Phi}^{-1}]^\top + U_{K\Phi}^{-1}\boldsymbol{\kappa}^{\Phi G}[U_{KG}^{-1}]^\top + U_{KG}^{-1}\boldsymbol{\kappa}^{GG}[U_{KG}^{-1}]^\top \tag{65}$$

Next we use the fact that $U_{\Delta\Phi}^{-1} = U_{\Delta K}^{-1}U_{K\Phi}^{-1}$ and that $U_{\Delta G}^{-1} = U_{\Delta K}^{-1}U_{KG}^{-1}$, which follows from the block structure of $U$. Consequently we arrive at the identity

$$\boldsymbol{\Sigma}_\Delta = U_{\Delta\Phi}^{-1}\boldsymbol{\kappa}^{\Phi\Phi}[U_{\Delta\Phi}^{-1}]^{-1} + U_{\Delta G}^{-1}\boldsymbol{\kappa}^{G\Phi}[U_{\Delta\Phi}^{-1}]^{-1} + U_{\Delta G}^{-1}\boldsymbol{\kappa}^{G\Phi}[U_{\Delta\Phi}^{-1}]^{-1} + U_{\Delta G}^{-1}\boldsymbol{\kappa}^{GG}[U_{\Delta G}^{-1}]^{-1}$$

$$= U_{\Delta K}^{-1}\boldsymbol{\Sigma}_K[U_{K\Delta}^{-1}]^\top. \tag{66}$$

Lastly, we note that, by the Schur-complement formula that $U_{\Delta K}^{-1} = -\left(\mathbf{I}+\boldsymbol{\Theta}_K\right)^{-1}D^{\Delta K}$. Thus, writing $\left(\mathbf{I}+\boldsymbol{\Theta}_K\right)\boldsymbol{\Sigma}_\Delta\left(\mathbf{I}+\boldsymbol{\Theta}_K\right)^\top = D^{\Delta K}\boldsymbol{\Sigma}_K[D^{\Delta K}]^\top$ as an integral equation, we find

$$\Sigma_{\mu\nu}^\Delta(t,s) + \int_0^t dt'\sum_\alpha K_{\mu\alpha}\Sigma_{\alpha\nu}^\Delta(t',s) + \int_0^s ds'\sum_\beta K_{\nu\beta}\Sigma_{\mu\beta}^\Delta(t,s')$$

$$+ \int_0^t dt'\int_0^s ds'\sum_{\alpha\beta}K_{\mu\alpha}K_{\nu\beta}\Sigma_{\alpha\beta}^\Delta(t',s') = \sum_{\alpha\beta}\int_0^t\Delta_\alpha(t')\int_0^s ds'\Delta_\beta(s')\Sigma_{\mu\alpha,\nu\beta}^K \tag{67}$$

Differentiation with respect to $t$ and $s$ gives a simple differential equation

$$\frac{\partial^2}{\partial t \partial s}\Sigma_{\mu\nu}^{\Delta}(t,s) + \sum_{\alpha} K_{\mu\alpha}\frac{\partial}{\partial s}\Sigma_{\alpha\nu}^{\Delta}(t,s) + \sum_{\beta} K_{\nu\beta}\frac{\partial}{\partial t}\Sigma_{\mu\beta}^{\Delta}(t,s)$$

$$+ \sum_{\alpha\beta} K_{\mu\alpha}K_{\nu\beta}\Sigma_{\alpha\beta}^{\Delta}(t,s) = \sum_{\alpha\beta}\Delta_{\alpha}(t)\Delta_{\beta}(s)\Sigma_{\mu\alpha,\nu\beta}^{K} \tag{68}$$

Let $\{\boldsymbol{\psi}_k\}$ be the eigenvectors of the kernel matrix $\boldsymbol{K}$. Projecting these dynamics on the eigenspace $\Sigma_{k\ell}(t,s) = \boldsymbol{\psi}_k^{\top}\boldsymbol{\Sigma}(t,s)\boldsymbol{\psi}_{\ell}$ recovers the equation in the main text

$$\left(\frac{\partial}{\partial t}+\lambda_k\right)\left(\frac{\partial}{\partial s}+\lambda_{\ell}\right)\Sigma_{k\ell}(t,s) = \sum_{k'\ell'}\Delta_{k'}(t)\Delta_{\ell'}(s)\Sigma_{kk'\ell\ell'}^{K} \tag{69}$$

Replacing $\Sigma^K = \kappa$ recovers the equation (7) in the main text.

## H.1 Perturbed Linear System

In this section, we provide a simpler derivation of the lazy limit training error variance dynamics. In this case, we merely perturb the dynamics around its infinite width value $\boldsymbol{\Delta}(t) = \boldsymbol{\Delta}_{\infty}(t) + \boldsymbol{\epsilon}^{\Delta}(t)$ and $\boldsymbol{K} = \boldsymbol{K}_{\infty} + \boldsymbol{\epsilon}^K$, and keep terms only linear in these perturbations. The perturbation $\boldsymbol{\epsilon}^K$ is fixed in time and the dynamics of $\boldsymbol{\epsilon}^{\Delta}(t)$ are

$$\frac{d}{dt}\boldsymbol{\epsilon}^{\Delta}(t) = -\boldsymbol{K}_{\infty}\boldsymbol{\epsilon}^{\Delta}(t) - \boldsymbol{\epsilon}^K\boldsymbol{\Delta}_{\infty}(t) \tag{70}$$

Projecting this equation on the eigenspace of $\boldsymbol{K}_{\infty}$ gives

$$\frac{d}{dt}\epsilon_k^{\Delta}(t) = -\lambda_k\epsilon_k(t) - \sum_{k'}\epsilon_{kk'}^K\Delta_{k'}^{\infty}(t) \tag{71}$$

This immediately recovers the final result of the last section

$$N\left(\frac{\partial}{\partial t}+\lambda_k\right)\left(\frac{\partial}{\partial t}+\lambda_k\right)\langle\epsilon_k^{\Delta}(t)\epsilon_{\ell}^{\Delta}(s)\rangle = \left(\frac{\partial}{\partial t}+\lambda_k\right)\left(\frac{\partial}{\partial t}+\lambda_k\right)\Sigma_{k\ell}^{\Delta}(t,s)$$

$$= \sum_{k'\ell'}\Sigma_{kk'\ell\ell'}^K\Delta_{k'}^{\infty}(t)\Delta_{\ell'}^{\infty}(s) \tag{72}$$

Qualitatively, the process of computing this linear correction (in $\boldsymbol{\epsilon}^K$) to the dynamics of $\boldsymbol{\Delta}$ is identical to the argument utilized in prior work on perturbative feature learning corrections [11]. In that context, the perturbation is caused by small amounts of feature learning, rather than initialization fluctuations.

## H.2 Mean Prediction Error Correction in the Lazy Limit

Using a similar heuristic as in the preceeding section, we now consider the correction to the mean predictor $\langle\Delta_{\mu}(t)\rangle$ in the lazy limit. Taylor expanding $\langle\boldsymbol{\Delta}(t)\rangle$ in powers of $1/N$, we find

$$\frac{d}{dt}\langle\boldsymbol{\Delta}(t)\rangle = \frac{d}{dt}\boldsymbol{\Delta}^{\infty}(t) + \frac{1}{N}\frac{d}{dt}\boldsymbol{\Delta}^1(t) + \dots$$

$$= -\langle(\boldsymbol{K}-\boldsymbol{K}^{\infty}+\boldsymbol{K}^{\infty})(\boldsymbol{\Delta}-\boldsymbol{\Delta}^{\infty}+\boldsymbol{\Delta}^{\infty})\rangle$$

$$= -\boldsymbol{K}^{\infty}\boldsymbol{\Delta}^{\infty} - \boldsymbol{K}^{\infty}\langle\boldsymbol{\Delta}-\boldsymbol{\Delta}^{\infty}\rangle$$

$$- \langle(\boldsymbol{K}-\boldsymbol{K}^{\infty})\rangle\boldsymbol{\Delta}^{\infty} - \langle(\boldsymbol{K}-\boldsymbol{K}^{\infty})(\boldsymbol{\Delta}-\boldsymbol{\Delta}^{\infty})\rangle$$

$$\sim -\boldsymbol{K}^{\infty}\boldsymbol{\Delta}^{\infty} - \frac{1}{N}\boldsymbol{K}^{\infty}\boldsymbol{\Delta}^1 - \frac{1}{N}\boldsymbol{K}^1\boldsymbol{\Delta}^{\infty} - \frac{1}{N}\langle\boldsymbol{\epsilon}^K\boldsymbol{\epsilon}^{\Delta}\rangle_{\infty} + \mathcal{O}(N^{-2}) \tag{73}$$

From the previous section we have that

$$\frac{d}{dt}\boldsymbol{\epsilon}^{\Delta} = -\boldsymbol{K}^{\infty}\boldsymbol{\epsilon}^{\Delta} - \boldsymbol{\epsilon}^K\boldsymbol{\Delta}^{\infty} \implies \boldsymbol{\epsilon}^{\Delta}(t) = -\int_0^t ds\,\exp\left(-\boldsymbol{K}^{\infty}(t-s)\right)\boldsymbol{\epsilon}^K\exp\left(-\boldsymbol{K}^{\infty}s\right)\boldsymbol{y} \tag{74}$$

Projecting these dynamics onto the eigenspace of the kernel gives

$$\epsilon_k^\Delta(t) = -\sum_\ell \epsilon_{k\ell}^K \frac{e^{-\lambda_\ell t} - e^{-\lambda_k t}}{\lambda_k - \lambda_\ell} y_\ell \tag{75}$$

where $\ell = k$ should be seen as the limit where $\lambda_k \to \lambda_\ell$ of the above. Thus we find that the leading mean correction to the error solves the following differential equation

$$\left(\frac{d}{dt} + \lambda_k\right) \Delta_k^1(t) = -\sum_\ell K_{k\ell}^1 y_\ell e^{-\lambda_\ell t} + \sum_{\ell\ell'} \Sigma_{k\ell\ell\ell'}^K \frac{e^{-\lambda_{\ell'} t} - e^{-\lambda_\ell t}}{\lambda_\ell - \lambda_{\ell'}} y_{\ell'}.$$

$$= \sum_\ell y_\ell e^{-\lambda_\ell t} \left[-K_{k\ell}^1 + \Sigma_{k\ell\ell\ell}^K t\right] + \sum_{\ell \neq \ell'} \Sigma_{k\ell\ell\ell'}^K \frac{e^{-\lambda_{\ell'} t} - e^{-\lambda_\ell t}}{\lambda_\ell - \lambda_{\ell'}} y_{\ell'} \tag{76}$$

We see that at late sufficiently large $t$, that the terms involving $\Sigma^K$ will dominate. We can gain more intuition by considering the special case of a single training data point where the mean error correction has the form

$$\left(\frac{d}{dt} + \lambda\right) \Delta^1(t) = y e^{-\lambda t} \left[-K^1 + t\Sigma^K\right] \implies \Delta^1(t) = y \left[-tK^1 + \frac{1}{2} t^2 \Sigma^K\right] e^{-\lambda t}$$

$$\implies \langle \Delta(t)^2 \rangle \sim \Delta^\infty(t)^2 + \frac{1}{N} \left[2y^2 t e^{-2\lambda t} \left[-K^1 + \frac{1}{2} t\Sigma^K\right] + \Sigma^\Delta(t,t)\right] + \mathcal{O}(N^{-2})$$

$$\sim \Delta^\infty(t)^2 + \frac{2}{N} y^2 t e^{-2\lambda t} \left[-K^1 + \Sigma^K t\right] + \mathcal{O}(N^{-2}) \tag{77}$$

While the term involving $\Sigma^K$ is positive for all $t$, $K^1$ could be positive or negative for a given architecture. If $K^1$ is positive, then MSE is initially improved at early times but after $t > \frac{K^1}{\Sigma^K}$ the MSE is worse than the infinite width. On the other hand, if $K^1$ is negative (as we suspect is typically the case), then the MSE will strictly decrease with network width for any time $t$.

## I  Two Layer Equations and Time/Time Diagonal

In this section, we analyze two layer networks in greater detail. Unlike the deep network case, two layer networks can be analyzed on the time-time diagonal: ie the dynamics only depend on $\Phi(t,t)$ and $G(t,t)$ rather than on all possible off-diagonal pairs of time points. Further, there are no response functions $A^\ell, B^\ell$ which complicate the recipe for calculating the propagator (Appendix E).

### I.1  A Single Training Point

For a two layer network trained on a single training point with norm constraint $|\boldsymbol{x}|^2 = D$, we have the following DMFT action

$$S[\{K(t), \hat{K}(t), \Delta(t), \hat{\Delta}(t)\}] \tag{78}$$

$$= \int dt \left[K(t)\hat{K}(t) + \hat{\Delta}(t)\left(\Delta(t) - y + \int ds\, \Theta(t-s)\Delta(s)K(s)\right)\right]$$

$$+ \ln \mathcal{Z}[\hat{K}, f], \; \mathcal{Z} = \mathbb{E}_{h,g} \exp\left(-\int dt \hat{K}(t)[\phi(h(t))^2 + g(t)^2]\right).$$

The saddle point equations are

$$\frac{\partial S}{\partial \hat{K}(t)} = K(t) - \langle [\phi(h(t))^2 + g(t)^2] \rangle = 0$$

$$\frac{\partial S}{\partial \hat{\Delta}(t)} = \Delta(t) - y + \int ds\, \Theta(t-s)\Delta(s)K(s) = 0$$

$$\frac{\partial S}{\partial K(s)} = \hat{K}(s) + \Delta(s) \int dt\, \hat{\Delta}(t)\Theta(t-s) = 0$$

$$\frac{\partial S}{\partial \Delta(s)} = \hat{\Delta}(s) + K(s) \int dt\, \hat{\Delta}(t)\Theta(t-s) = 0 \tag{79}$$

From these equations, we can compute the entries in the Hessian of the DMFT action $S$. Letting $q(t) = \begin{bmatrix} \Delta(t) \\ K(t) \end{bmatrix}$ and $\hat{q}(t) = \begin{bmatrix} \hat{\Delta}(t) \\ \hat{K}(t) \end{bmatrix}$

$$\frac{\partial^2 S}{\partial q(t) \partial q(s)^\top} = \mathbf{0}$$

$$\frac{\partial^2 S}{\partial \hat{q}(t) \partial q(s)^\top} = \begin{bmatrix} \delta(t-s) + \Theta(t-s)K(s) & \Theta(t-s)\Delta(s) \\ -\left\langle \frac{\partial}{\partial \Delta(s)}(\phi(h(t))^2 + g(t)^2) \right\rangle & \delta(t-s) \end{bmatrix}$$

$$\frac{\partial^2 S}{\partial \hat{q}(t) \partial \hat{q}(s)^\top} = \begin{bmatrix} 0 & 0 \\ 0 & \kappa(t,s) \end{bmatrix} \tag{80}$$

where $\kappa(t,s) = \left\langle (\phi(h(t))^2 + g(t)^2)(\phi(h(s))^2 + g(s)^2) \right\rangle - K(t)K(s)$ is the NTK's fourth cumulant. We now vectorize our order parameters over time $q = \text{Vec}\{q(t)\}_{t \in \mathbb{R}_+}$ and $\hat{q} = \text{Vec}\{\hat{q}(t)\}_{t \in \mathbb{R}_+}$ and express the full Hessian

$$\nabla^2 S = \begin{bmatrix} \mathbf{0} & \frac{\partial^2 S}{\partial q \partial \hat{q}^\top} \\ \frac{\partial^2 S}{\partial \hat{q} \partial q^\top} & \frac{\partial^2 S}{\partial \hat{q} \partial \hat{q}^\top} \end{bmatrix} \implies -[\nabla^2 S]^{-1} = \begin{bmatrix} (\frac{\partial^2 S}{\partial \hat{q} \partial q^\top})^{-1} \frac{\partial^2 S}{\partial \hat{q} \partial \hat{q}}(\frac{\partial^2 S}{\partial q \partial \hat{q}^\top})^{-1} & -(\frac{\partial^2 S}{\partial \hat{q} \partial q^\top})^{-1} \\ -(\frac{\partial^2 S}{\partial q \partial \hat{q}^\top})^{-1} & \mathbf{0} \end{bmatrix} \tag{81}$$

The covariance matrix of interest (for $q(t)$) is thus

$$\Sigma_q = \begin{bmatrix} \mathbf{I} + \mathbf{\Theta}_K & \mathbf{\Theta}_\Delta \\ -\mathbf{D} & \mathbf{I} \end{bmatrix}^{-1} \begin{bmatrix} 0 & 0 \\ 0 & \kappa \end{bmatrix} \begin{bmatrix} \mathbf{I} + \mathbf{\Theta}_K & \mathbf{\Theta}_\Delta \\ -\mathbf{D} & \mathbf{I} \end{bmatrix}^{-1\top} . \tag{82}$$

where $[\mathbf{\Theta}_K](t,s) = \Theta(t-s)K(s)$ and $[\mathbf{\Theta}_\Delta](t,s) = \Theta(t-s)\Delta(s)$. The above equations allow one to use the infinite width DMFT dynamics for $K(t), \Delta(t)$ to compute the finite size fluctuation dynamics of the kernel $K$ and the error signal $\Delta$.

### I.1.1 Computing Field Sensitivities

In this section, we compute $D(t,s)$ by solving for the sensitivity of order parameters. We start with the DMFT field equations

$$h(t) = u + \gamma \int_0^t ds \Delta(s) g(s) , \quad z(t) = r + \gamma \int_0^t ds \Delta(s) \phi(h(t)). \tag{83}$$

Now, differentiating both sides with respect to $\Delta(s')$ gives

$$\frac{\partial h(t)}{\partial \Delta(s')} = \gamma \Theta(t-s') g(s') + \gamma \int_0^t ds \Delta(s) \frac{\partial g(s)}{\partial \Delta(s')}$$

$$\frac{\partial z(t)}{\partial \Delta(s')} = \gamma \Theta(t-s') \phi(h(s')) + \gamma \int_0^t ds \Delta(s) \frac{\partial \phi(h(s))}{\partial \Delta(s')}. \tag{84}$$

We can compute $D$ Monte carlo by iteratively solving the above equations for each sampled trajectory $\{h(t), z(t)\}$ [65, 46]. Averaging the necessary fields over the Monte Carlo samples will give us the final expressions for $D(t,s)$.

$$D(t,s) = \left\langle \frac{\partial}{\partial \Delta(s)}(\phi(h(t))^2 + g(t)^2) \right\rangle \tag{85}$$

Similarly, the uncoupled kernel variance $\kappa(t,s)$ can be evaluated via Monte Carlo sampling for nonlinear networks.

### I.2 Test Point Fluctuation Dynamics

We now are in a position to calculate the test/train kernel and test prediction fluctuations. To do this systematically, we augment $S$ with the test point prediction $f_\star$ and field $h_\star$ and introduce the kernel

$K_\star(t) = \langle \phi(h(t))\phi(h_\star(t)) + g(t)g_\star(t) \rangle$. The test prediction $f_\star$ and field $h_\star$ have dynamics

$$h_\star(t) = u_\star + \gamma \int_0^t ds \Delta(s) \dot{\phi}(h_\star(s))z(s)K_\star^x \ , \ \langle u_\star u \rangle = K_\star^x$$

$$\frac{\partial}{\partial t} f_\star(t) = K_\star(t)\Delta(t) \ , \ K_\star(t) = \langle \phi(h(t))\phi(h_\star(t)) + g(t)g_\star(t) \rangle \tag{86}$$

The augmented action for this DMFT has the form

$$S = \int dt \, \hat{f}_\star(t) \left( f_\star(t) - \int ds \Theta(t-s)\Delta(s)K_\star(s) \right) + \int dt \, \hat{K}_\star(t)K_\star(t)$$

$$+ \int dt \, \hat{\Delta}(t) \left( \Delta(t) - y + \int ds \, \Theta(t-s)\Delta(s)K(s) \right) + \int dt \, \hat{K}(t)K(t)$$

$$+ \ln \mathbb{E} \exp \left( - \int \hat{K}(t)(\phi(h(t))^2 + g(t)^2) - \int \hat{K}_\star(t)(\phi(h(t))\phi(h_\star(t)) + g(t)g_\star(t)) \right) \tag{87}$$

We let $\boldsymbol{q}(t) = [\Delta(t), f_\star(t), K(t), K_\star(t)]^\top$

$$\nabla^2_{\hat{q}\hat{q}} S[\boldsymbol{q}, \hat{\boldsymbol{q}}] = \begin{bmatrix} 0 & 0 & 0 & 0 \\ 0 & 0 & 0 & 0 \\ 0 & 0 & \boldsymbol{\kappa} & \boldsymbol{\kappa}_\star^\top \\ 0 & 0 & \boldsymbol{\kappa}_\star & \boldsymbol{\kappa}_{\star\star} \end{bmatrix} \ , \ \nabla^2_{\hat{q},q} S[\boldsymbol{q}, \hat{\boldsymbol{q}}] = \begin{bmatrix} \mathbf{I} + \boldsymbol{\Theta}_K & 0 & \boldsymbol{\Theta}_\Delta & 0 \\ -\boldsymbol{\Theta}_{K_\star} & \mathbf{I} & 0 & -\boldsymbol{\Theta}_\Delta \\ -\boldsymbol{D} & 0 & \mathbf{I} & 0 \\ -\boldsymbol{D}_\star & 0 & 0 & \mathbf{I} \end{bmatrix}$$

$$D(t, s) = \left\langle \frac{\partial}{\partial \Delta(s)} (\phi(h(t))^2 + g(t)^2) \right\rangle \tag{88}$$

$$D_\star(t, s) = \left\langle \frac{\partial}{\partial \Delta(s)} (\phi(h(t))\phi(h_\star(t)) + g(t)g_\star(t)) \right\rangle \tag{89}$$

Our total covariance matrix / propagator is thus

$$\boldsymbol{\Sigma} = \begin{bmatrix} \mathbf{I} + \boldsymbol{\Theta}_K & 0 & \boldsymbol{\Theta}_\Delta & 0 \\ -\boldsymbol{\Theta}_{K_\star} & \mathbf{I} & 0 & -\boldsymbol{\Theta}_\Delta \\ -\boldsymbol{D} & 0 & \mathbf{I} & 0 \\ -\boldsymbol{D}_\star & 0 & 0 & \mathbf{I} \end{bmatrix}^{-1} \begin{bmatrix} 0 & 0 & 0 & 0 \\ 0 & 0 & 0 & 0 \\ 0 & 0 & \boldsymbol{\kappa} & \boldsymbol{\kappa}_\star^\top \\ 0 & 0 & \boldsymbol{\kappa}_\star & \boldsymbol{\kappa}_{\star\star} \end{bmatrix} \begin{bmatrix} \mathbf{I} + \boldsymbol{\Theta}_K & 0 & \boldsymbol{\Theta}_\Delta & 0 \\ -\boldsymbol{\Theta}_{K_\star} & \mathbf{I} & 0 & -\boldsymbol{\Theta}_\Delta \\ -\boldsymbol{D} & 0 & \mathbf{I} & 0 \\ -\boldsymbol{D}_\star & 0 & 0 & \mathbf{I} \end{bmatrix}^{-1\top} \tag{90}$$

This is the equation provided in the main text Equation (8).

## I.3 Two Layer Linear Network Closed Form

For a linear network on a single data point, we can compute $D(t, s)$ and $\kappa(t, s)$ analytically. We start from the field equations

$$\frac{dh(t)}{dt} = \gamma \Delta(t)z(t) \ , \ \frac{dz(t)}{dt} = \gamma \Delta(t)h(t) \tag{91}$$

We can make a change of variables $v_+(t) = \frac{1}{\sqrt{2}}(h(t) + z(t))$ and $v_-(t) = \frac{1}{\sqrt{2}}(h(t) - z(t))$. We note that $v_+(0) = \frac{1}{\sqrt{2}}(u + r)$ and $v_-(0) = \frac{1}{\sqrt{2}}(u - r)$ are independent Gaussians. These functions $v_+(t), v_-(t)$ satisfy dynamics

$$\frac{dv_+}{dt} = \gamma \Delta(t)v_+(t) \ , \ \frac{dv_-(t)}{dt} = -\gamma \Delta(t)v_-(t)$$

$$\implies v_+(t) = \exp \left( \gamma \int_0^t ds \Delta(s) \right) v_+(0) \implies \frac{\partial v_+(t)}{\partial \Delta(s)} = \gamma v_+(t)\Theta(t-s)$$

$$\implies v_-(t) = \exp \left( -\gamma \int_0^t ds \Delta(s) \right) v_-(0) \implies \frac{\partial v_+(t)}{\partial \Delta(s)} = -\gamma v_-(t)\Theta(t-s) \tag{92}$$

Now, we use the fact that $v_+(0) = \frac{1}{\sqrt{2}}(u + r)$ and $v_-(0) = \frac{1}{\sqrt{2}}(u - r)$ are independent standard normal random variables to compute $K(t) = \langle h(t)^2 + z(t)^2 \rangle = \langle v_+(t)^2 + v_-(t)^2 \rangle$

$$D(t, s) = \frac{\partial}{\partial \Delta(s)} \langle h(t)^2 + z(t)^2 \rangle = 2\gamma \left[ \langle v_+(t)^2 \rangle - \langle v_-(t)^2 \rangle \right] \Theta(t - s)$$

$$= 2\gamma \left[ \exp\left( 2\gamma \int_0^t ds \Delta(s) \right) - \exp\left( -2\gamma \int_0^t ds \Delta(s) \right) \right] \Theta(t - s) \qquad (93)$$

This operator is causal ($D(t, s) = 0$ for $s > t$) as expected and vanishes as $t \to 0$. If we take $\gamma \to 0$, we have $D(t, s) \to 0$ which agrees with our reasoning that fields $h, z$ only depend on $\Delta$ in the feature learning regime. Since all fields are Gaussian in the linear network case, we can use Wick's theorem to obtain the exact uncoupled kernel variance in the two layer case.

$$\kappa(t, s) = \langle (h(t)^2 + z(t)^2)(h(s)^2 + z(s)^2) \rangle - K(t)K(s)$$

$$= 2 \langle h(t)h(s) \rangle^2 + 2 \langle h(t)z(s) \rangle^2 + 2 \langle z(t)h(s) \rangle^2 + 2 \langle z(t)z(s) \rangle^2$$

$$= \langle v_+(t)v_+(s) + v_-(t)v_-(s) \rangle^2 + \langle v_+(t)v_+(s) - v_-(t)v_-(s) \rangle^2 \qquad (94)$$

The $v_\pm(t)$ functions are those given above. Using the fact that $\langle v_+(0)^2 \rangle = \langle v_-(0)^2 \rangle = 1$ allows us to easily compute the single site average above.

## J   Multiple Samples with Whitened Data

In this section, we analyze the role that sample number plays in dynamics in a simplified model of a two layer linear network trained on whitened data. Concretely, we assume that $\frac{\boldsymbol{x}_\mu \cdot \boldsymbol{x}_\nu}{D} = \delta_{\mu\nu}$. The field equations for preactivations $h_\mu(t)$ and pregradients $z(t)$ obey

$$\frac{d}{dt} h_\mu(t) = \gamma \Delta_\mu(t) z(t) \,, \quad \frac{d}{dt} z(t) = \gamma \sum_{\mu=1}^P \Delta_\mu(t) h_\mu(t) \qquad (95)$$

We will assume the targets have unit norm $|\boldsymbol{y}|^2 = 1$ and we define the projection of $\boldsymbol{\Delta}$ onto the target as $\Delta_y(t) = \boldsymbol{y} \cdot \boldsymbol{\Delta}(t)$. The other $P - 1$ orthogonal components are denoted $\boldsymbol{\Delta}_\perp(t)$ so that $\boldsymbol{\Delta} = \Delta_y(t)\boldsymbol{y} + \boldsymbol{\Delta}_\perp(t)$ with $\boldsymbol{\Delta}_\perp(t) \cdot \boldsymbol{y} = 0$. At infinite width, $\boldsymbol{\Delta}_\perp = 0$ and our field equations become

$$\frac{d}{dt} h_y(t) = \Delta_y(t) z(t) \,, \quad \frac{d}{dt} z(t) = \Delta_y(t) h_y(t) \,, \quad \boldsymbol{\Delta}_\perp(t) = 0 \,, \quad h_\perp \sim \mathcal{N}(0, 1) \qquad (96)$$

However, at finite width $N$, the off-target predictions $\boldsymbol{\Delta}_\perp$ fluctuate over random initialization. To model all of the fluctuations simultaneously, we consider the following action

$$S = \gamma \int dt \sum_\mu \hat{\Delta}_\mu(t)(\Delta_\mu(t) - y_\mu) + \ln \mathbb{E} \exp\left( \int dt \sum_\mu \hat{\Delta}_\mu(t) z(t) h_\mu(t) \right) \qquad (97)$$

which enforces the constraint that $\Delta_\mu(t) = y_\mu - \frac{1}{\gamma} \langle z(t)h_\mu(t) \rangle$ at infinite width. The Hessian over order parameters $\boldsymbol{q} = \text{Vec}\{\Delta_\mu(t), \hat{\Delta}_\mu(t)\}$ has the form

$$\nabla_{\boldsymbol{q}}^2 S = \begin{bmatrix} \boldsymbol{0} & (\gamma \mathbf{I} + \boldsymbol{D})^\top \\ \gamma \mathbf{I} + \boldsymbol{D} & \boldsymbol{\kappa} \end{bmatrix} \,, \quad D_{\mu\nu}(t, s) = \left\langle \frac{\partial}{\partial \Delta_\nu(s)} z(t) h_\mu(t) \right\rangle \qquad (98)$$

We thus get the following covariance for predictions $\boldsymbol{\Sigma}_\Delta = (\gamma \mathbf{I} + \boldsymbol{D})^{-1} \boldsymbol{\kappa} \left[ (\gamma \mathbf{I} + \boldsymbol{D})^{-1} \right]^\top$. We now compute the necessary components of the $D$ tensor

$$\frac{\partial h_\mu(t)}{\partial \Delta_\nu(s)} = \gamma \delta_{\mu\nu} \Theta(t - s) z(s) + \gamma \int_0^t dt' \Delta_\mu(t') \frac{\partial z(t')}{\partial \Delta_\nu(s)}$$

$$\frac{\partial z(t)}{\partial \Delta_\nu(s)} = \gamma \Theta(t - s) h_\nu(s) + \gamma \int_0^t dt' \sum_\mu \Delta_\mu(t') \frac{\partial h_\mu(t')}{\partial \Delta_\nu(s)}$$

$$= \gamma \Theta(t - s) h_\nu(s) + \gamma \int_0^t dt' \Delta_y(t') \frac{\partial h_y(t')}{\partial \Delta_\nu(s)} \qquad (99)$$

In the last line, we used the fact that these equations are to be evaluated at the mean field infinite width stochastic process where $\Delta_\perp(t) = 0$. To compute the sensitivity tensor $D$, we find the following equations for our correlators of interest:

$$\left\langle \frac{\partial h_\mu(t)}{\partial \Delta_\nu(s)} z(t) \right\rangle = \delta_{\mu\nu} \gamma \Theta(t-s) \langle z(s) z(t) \rangle \ , \ \mu, \nu \neq y$$

$$\left\langle \frac{\partial z(t)}{\partial \Delta_\nu(s)} h_\mu(t) \right\rangle = \gamma \Theta(t-s) \delta_{\mu\nu} \ , \ \mu, \nu \neq y \tag{100}$$

$$\left\langle \frac{\partial h_y(t)}{\partial \Delta_y(s)} z(t) \right\rangle = \gamma \Theta(t-s) \langle z(s) z(t) \rangle + \gamma \int_0^t dt' \Delta_y(t') \left\langle \frac{\partial z(t')}{\partial \Delta_y(s)} z(t) \right\rangle$$

$$\left\langle \frac{\partial z(t)}{\partial \Delta_y(s)} z(t') \right\rangle = \gamma \Theta(t-s) \langle h_y(s) z(t) \rangle + \gamma \int_0^t dt'' \Delta_y(t'') \left\langle \frac{\partial h_y(t'')}{\partial \Delta_y(s)} z(t') \right\rangle$$

We therefore see that the components of $D$ decouple over indices. In the $y$ direction, we have the following equations

$$D_y(t, s) = \left\langle \frac{\partial h_y(t)}{\partial \Delta_y(s)} z(t) \right\rangle + \left\langle \frac{\partial z(t)}{\partial \Delta_y(s)} h_y(t) \right\rangle \tag{101}$$

where the correlators must be solved self-consistently. We will provide this solution in one moment, but first, we will look at the orthogonal directions. For the $P - 1$ orthogonal directions, we obtain the explicit formula for $D$ in each of these directions

$$D_\perp(t, s) = \left\langle \frac{\partial h_\perp(t)}{\partial \Delta_\perp(s)} z(t) \right\rangle + \left\langle \frac{\partial z(t)}{\partial \Delta_\perp(s)} h_\perp(t) \right\rangle$$

$$= \gamma \Theta(t-s) \langle z(t) z(s) \rangle + \gamma \Theta(t-s) \tag{102}$$

Now, we return to $D_y$. To solve these equations we utilize the change of variables employed in the single sample case $v_+(t) = \frac{1}{\sqrt{2}}(h_y(t) + z(t)), v_-(t) = \frac{1}{\sqrt{2}}(h_y(t) - z(t))$ (see Appendix I.3). This orthogonal transformation decouples the dynamics

$$\frac{d}{dt} v_+(t) = \gamma \Delta_y(t) v_+(t) \ , \ \frac{d}{dt} v_-(t) = -\gamma \Delta_y(t) v_-(t) \tag{103}$$

As a consequence, the field derivatives close

$$\frac{\partial v_+(t)}{\partial \Delta_y(s)} = \gamma \Theta(t-s) v_+(s) + \int_0^t dt' \Delta_y(t') \frac{\partial v_+(t')}{\partial \Delta_y(s)}$$

$$\frac{\partial v_-(t)}{\partial \Delta_y(s)} = -\gamma \Theta(t-s) v_-(s) - \int_0^t dt' \Delta_y(t') \frac{\partial v_-(t')}{\partial \Delta_y(s)} \tag{104}$$

The correlator of interest is

$$\langle h_y(t) z(t) \rangle = \frac{1}{2} \langle [v_+(t) + v_-(t)][v_+(t) - v_-(t)] \rangle = \frac{1}{2} \langle v_+(t)^2 - v_-(t)^2 \rangle \tag{105}$$

So we get that

$$D_y(t, s) = \frac{1}{2} \left\langle \frac{\partial}{\partial \Delta_y(s)} \left( v_+(t)^2 - v_-(t)^2 \right) \right\rangle$$

$$= \left\langle v_+(t) \frac{\partial v_+(t)}{\partial \Delta_y(s)} \right\rangle - \left\langle v_-(t) \frac{\partial v_-(t)}{\partial \Delta_y(s)} \right\rangle \tag{106}$$

Similarly, we can derive the on-target and off-target uncoupled variances $\kappa_y(t, s)$ and $\kappa_\perp(t, s)$, which satisfy

$$\kappa_y(t, s) = \langle v_+(t) v_+(s) + v_-(t) v_-(s) \rangle^2 + \langle v_+(t) v_+(s) - v_-(t) v_-(s) \rangle^2$$

$$\kappa_\perp(t, s) = \frac{1}{2} \langle v_+(t) v_+(s) + v_-(t) v_-(s) \rangle \tag{107}$$

Using these functions, we arrive at the following variance for each of the $P$ dimensions

$$\boldsymbol{\Sigma}_{\Delta_y} = (\gamma \mathbf{I} + \boldsymbol{D}_y)^{-1} \boldsymbol{\kappa}_y (\gamma \mathbf{I} + \boldsymbol{D}_y)^{-1}$$

$$\boldsymbol{\Sigma}_{\Delta_\perp} = (\gamma \mathbf{I} + \boldsymbol{D}_\perp)^{-1} \boldsymbol{\kappa}_\perp (\gamma \mathbf{I} + \boldsymbol{D}_\perp)^{-1} \tag{108}$$

Using the fact that all $\Delta_\perp$ variables are independent and identically distributed under the leading order picture, the expected training loss has the form

$$\langle |\boldsymbol{\Delta}|^2 \rangle \approx \Delta_y^\infty(t)^2 + \frac{2}{N} \Delta_y^1(t) \Delta_y^\infty(t) + \frac{1}{N} \Sigma_{\Delta_y}(t, t) + \frac{(P-1)}{N} \Sigma_{\Delta_\perp}(t, t) + \mathcal{O}(N^{-2}). \tag{109}$$

where $\langle \Delta_y - \Delta_y^\infty \rangle = \frac{1}{N} \Delta_y^1(t) + \mathcal{O}(N^{-2})$. We note that the bias correction if $\mathcal{O}(N^{-1})$ while the variance is $\mathcal{O}(P/N)$. We compare the above leading order theory with and without the bias correction in Appendix Figure A.2.

# K   Online Learning

Our technology for computing finite size effects can easily be translated to a setting where the neural network is trained in an online fashion, disregarding the effect of SGD noise. At each step, we compute the gradient over the full data distribution $p(\boldsymbol{x})$. Focusing on MSE loss, we study the following equation

$$\frac{d}{dt} \Delta(\boldsymbol{x}, t) = -\mathbb{E}_{\boldsymbol{x}' \sim p(\boldsymbol{x}')} K(\boldsymbol{x}, \boldsymbol{x}'; t) \Delta(\boldsymbol{x}', t) \tag{110}$$

where $K(\boldsymbol{x}, \boldsymbol{x}'; t)$ is the dynamic NTK and $\Delta(\boldsymbol{x}, t) = y(\boldsymbol{x}) - f(\boldsymbol{x}, t)$ is the prediction error. In general the distribution involves integration over an uncountable set of possible inputs $\boldsymbol{x}$. To remedy this, we utilize a countable orthonormal basis of functions for the data distribution $\{\psi_k(\boldsymbol{x})\}_{k=1}^\infty$. For example, if $p(\boldsymbol{x})$ were the isotropic Gaussian density for $\mathcal{N}(0, \mathbf{I})$, then $\psi_k$ could be Hermite polynomials. We expand $\Delta$ and $K$ in this basis $\psi_k$, and arrive at the following differential equation

$$\frac{d}{dt} \Delta_k(t) = - \sum_\ell K_{k\ell}(t) \Delta_\ell(t) \tag{111}$$

By orthonormality, the average turned into a sum over all possible orthonormal functions $\{\psi_k\}$. We note that since $K$ is evolving in time, there is not generally a fixed basis of functions that diagonalize $K$, resulting in the couplings across eigenmodes in Equation (111). Since, in online learning, there is no distinction between the training and test distribution, our error of interest is simply $\mathcal{L}(t) = \sum_k \Delta_k(t)^2$. To obtain the finite size corrections to this quantity, we compute the joint propagator for all variables $\{K_{k\ell}(t), \Delta_k(t)\}$. If we wanted to pursue a perturbation theory in rates (Appendix G.2), we could again define a transition matrix $\boldsymbol{T}$ and rate matrix $\boldsymbol{R}(t)$ as

$$\boldsymbol{R}(t) = -\log \boldsymbol{T}(t) \ , \ \frac{d}{dt} T_{k\ell}(t) = - \sum_{k'} K_{kk'}(t) T_{k'\ell}(t) \ , \ T_{k\ell}(0) = \delta_{k\ell} \tag{112}$$

We can then obtain $\boldsymbol{\Delta} = \exp(-\boldsymbol{R}(t)) \boldsymbol{y}$, where $y_k = \mathbb{E}_{\boldsymbol{x}} \psi_k(\boldsymbol{x}) y(\boldsymbol{x})$. Since $\boldsymbol{R}$ has a finite size mean correction and finite size fluctuations, so too does the error $\Delta_k(t)$ and the loss $\mathcal{L}$ (Appendix G.2).

## K.1   Two Layer Networks

In the two layer case, instead of tracking kernels, we could instead deal with the distribution over read-in vectors $\boldsymbol{w} \in \mathbb{R}^D$ and readout scalars $a \in \mathbb{R}$ as in the original works on mean field networks [6, 66]. When training on the population risk equations for $\boldsymbol{x} \sim \mathcal{N}(0, \mathbf{I})$

$$\frac{d}{dt} \boldsymbol{w} = a \mathbb{E}_{\boldsymbol{x}} \Delta(\boldsymbol{x}) \dot{\phi}(\boldsymbol{w} \cdot \boldsymbol{x}) \boldsymbol{x} = \mathbb{E}_{\boldsymbol{x}} \frac{\partial \Delta(\boldsymbol{x})}{\partial \boldsymbol{x}} \dot{\phi}(\boldsymbol{w} \cdot \boldsymbol{x}) + \mathbb{E} \Delta(\boldsymbol{x}) \ddot{\phi}(\boldsymbol{w} \cdot \boldsymbol{x}) \boldsymbol{w}$$

$$\frac{d}{dt} a = \mathbb{E}_{\boldsymbol{x}} \Delta(\boldsymbol{x}) \phi(\boldsymbol{w} \cdot \boldsymbol{x}) \tag{113}$$

The action has the form

$$S = \gamma \int dt d\boldsymbol{x} \hat{\Delta}(t, \boldsymbol{x}) (\Delta(t, \boldsymbol{x}) - y(\boldsymbol{x})) + \ln \mathbb{E}_{a, \boldsymbol{w}} \exp \left( \int dt d\boldsymbol{x} \hat{\Delta}(t, \boldsymbol{x}) a(t) \phi(\boldsymbol{w}(t) \cdot \boldsymbol{x}) \right) \tag{114}$$

The Hessian over $\boldsymbol{q} = \{\Delta_\mu(t), \hat{\Delta}_\mu(t)\}$ is

$$\nabla^2 S = \begin{bmatrix} 0 & \mathbf{I} + \boldsymbol{D}_\Delta \\ \mathbf{I} + \boldsymbol{D}_\Delta & \boldsymbol{\kappa} \end{bmatrix}.$$ (115)

where $D_\Delta(t, \boldsymbol{x}; s, \boldsymbol{x}') = \left\langle \frac{\partial}{\partial \Delta(s, \boldsymbol{x}')} a(t) \phi(\boldsymbol{w}(t) \cdot \boldsymbol{x}) \right\rangle$ We can use the following implicit rule

$$\frac{\partial a(t)}{\partial \Delta(s, \boldsymbol{x})} = \gamma \Theta(t - s) p(\boldsymbol{x}) \phi(\boldsymbol{w}(s) \cdot \boldsymbol{x}) + \gamma \mathbb{E}_{\boldsymbol{x}'} \int_0^t dt' \Delta(t', \boldsymbol{x}') \dot{\phi}(\boldsymbol{w} \cdot \boldsymbol{x}') \boldsymbol{x}' \cdot \frac{\partial \boldsymbol{w}(t)}{\partial \Delta(s, \boldsymbol{x})}$$

$$\frac{\partial \boldsymbol{w}(t)}{\partial \Delta(s, \boldsymbol{x})} = \gamma \Theta(t - s) p(\boldsymbol{x}) a(s) \dot{\phi}(\boldsymbol{w}(s) \cdot \boldsymbol{x}) \boldsymbol{x}$$

$$+ \gamma \mathbb{E}_{\boldsymbol{x}'} \int_0^t dt' \Delta(t', \boldsymbol{x}') \left[ \frac{\partial a(t')}{\partial \Delta(s, \boldsymbol{x})} \dot{\phi}(\boldsymbol{w} \cdot \boldsymbol{x}') + a(t') \ddot{\phi}(\boldsymbol{w} \cdot \boldsymbol{x}') \frac{\partial \boldsymbol{w}(t')}{\partial \Delta(s, \boldsymbol{x})} \cdot \boldsymbol{x}' \right] \quad (116)$$

The above equations could be solved and then used to compute $D_\Delta(t, \boldsymbol{x}; s, \boldsymbol{x}')$ which must then be inverted to get the observed prediction variance.

## K.2 Linear Activations

Using the ideas in the preceding sections, we can make more progress in the case of a two layer linear network in the online learning setting. The key idea is to track the kernel and prediction error projections onto the space of linear functions. In this case we get the following DMFT over the order parameter $\boldsymbol{\beta}(t) = \frac{1}{N} \boldsymbol{W}^\top \boldsymbol{a} \in \mathbb{R}^D$.

$$\frac{d}{dt} a(t) = \gamma(\boldsymbol{\beta}_\star - \boldsymbol{\beta}(t)) \cdot \boldsymbol{w}(t)$$

$$\frac{d}{dt} \boldsymbol{w}(t) = \gamma a(t)(\boldsymbol{\beta}_\star - \boldsymbol{\beta}(t))$$

$$\boldsymbol{\beta}(t) = \frac{1}{\gamma} \langle a(t) \boldsymbol{w}(t) \rangle$$ (117)

At infinite width, we see that the dynamics can be reduced to tracking the projection of the weights $\boldsymbol{w}$ and $\boldsymbol{\beta}$ on the $\boldsymbol{\beta}_\star$ direction. The $D - 1$ off-target dimensions vanish $\boldsymbol{\beta}_\perp(t) = 0$. At infinite width, we arrive at the alignment dynamics studied in prior work [64, 9]

$$\frac{d}{dt} \boldsymbol{\beta}(t) = \boldsymbol{M}(t)(\boldsymbol{\beta}_\star - \boldsymbol{\beta}(t))$$

$$\frac{d}{dt} \boldsymbol{M}(t) = \gamma^2 \boldsymbol{\beta}(t)(\boldsymbol{\beta}_\star - \boldsymbol{\beta}(t))^\top + \gamma^2 \boldsymbol{\beta}(t)(\boldsymbol{\beta}_\star - \boldsymbol{\beta}(t))^\top$$

$$+ 2\gamma^2 (\boldsymbol{\beta}_\star - \boldsymbol{\beta}(t)) \cdot \boldsymbol{\beta}(t) \mathbf{I}$$ (118)

We note that $\boldsymbol{\beta}(t) = \beta(t) \boldsymbol{\beta}_\star$ and that $\boldsymbol{M}$ has only one special eigenvector $\boldsymbol{\beta}_\star$ with eigenvalue $m_\star(t)$. It thus suffices to track evolution in this single direction

$$\frac{d}{dt} \beta(t) = m_\star(t)(\beta_\star - \beta(t)) \, , \quad \frac{d}{dt} m_\star(t) = 4\gamma^2 \beta(t)(\beta_\star - \beta(t))$$ (119)

We note that this equation is identical to the differential equation for a single training example in Appendix J. Here $\beta_\star - \beta(t)$ plays the role of $\Delta_y(t)$ and $m_\star(t)$ plays the role of the kernel $K_y(t)$. A key observation is the conservation law $4\gamma^2 \frac{d}{dt} \beta(t)^2 = \frac{d}{dt} m_\star(t)^2$, from which it follows that $m_\star(t)^2 - 4 = 4\gamma^2 \beta(t)$ [9]

$$\frac{d}{dt} \beta(t) = 2\sqrt{1 + \gamma^2 \beta(t)^2}(\beta_\star - \beta(t))$$ (120)

This is identical to the differential equations for a single sample (producing prediction $f(t)$ and kernel $K(t)$) if the following substitutions are made

$$f(t) \leftrightarrow \beta(t) \, , \quad K(t) \leftrightarrow m_\star(t)$$ (121)

We now proceed to compute finite size corrections starting from the action

$$S = \gamma \int dt \hat{\boldsymbol{\beta}}(t) \cdot \boldsymbol{\beta}(t) + \ln \mathbb{E} \exp \left( - \int dt \hat{\boldsymbol{\beta}}(t) \cdot \boldsymbol{w}(t) a(t) \right) \tag{122}$$

The necessary ingredients are

$$\boldsymbol{\kappa}(t,s) = \left\langle a(t) a(s) \boldsymbol{w}(t) \boldsymbol{w}(s)^\top \right\rangle - \gamma^2 \boldsymbol{\beta}(t) \boldsymbol{\beta}(s)$$
$$= \langle a(t) a(s) \rangle \left\langle \boldsymbol{w}(t) \boldsymbol{w}(s)^\top \right\rangle + \left\langle a(s) \boldsymbol{w}(t) \right\rangle \left\langle a(t) \boldsymbol{w}(s)^\top \right\rangle \in \mathbb{R}^{D \times D} \tag{123}$$

Similarly we have to compute the sensitivity tensor

$$\boldsymbol{D}(t,s) = \left\langle \frac{\partial}{\partial \boldsymbol{\beta}(s)^\top} a(t) \boldsymbol{w}(t) \right\rangle \in \mathbb{R}^{D \times D} \tag{124}$$

We start from the dynamics

$$\frac{d}{dt} \boldsymbol{w}(t) = \gamma a(t)(\boldsymbol{\beta}_\star - \boldsymbol{\beta}(t)) \ , \quad \frac{d}{dt} a(t) = \gamma (\boldsymbol{\beta}_\star - \boldsymbol{\beta}(t)) \cdot \boldsymbol{w}(t) \tag{125}$$

Next, we have to calculate causal derivatives for fields

$$\frac{\partial}{\partial \boldsymbol{\beta}(s)^\top} \boldsymbol{w}(t) = -\gamma \Theta(t-s) a(s) \mathbf{I} + \gamma \int_0^t dt' (\boldsymbol{\beta}_\star - \boldsymbol{\beta}(t')) \frac{\partial a(t')}{\partial \boldsymbol{\beta}(s)^\top}$$
$$\frac{\partial}{\partial \boldsymbol{\beta}(s)} a(t) = -\gamma \Theta(t-s) \boldsymbol{w}(s) + \gamma \int_0^t dt' (\boldsymbol{\beta}_\star - \boldsymbol{\beta}(t')) \cdot \frac{\partial \boldsymbol{w}(t')}{\partial \boldsymbol{\beta}(s)} \tag{126}$$

Following an identical argument as in J, we see that $\boldsymbol{D}$ has block diagonal structure with $D_{\beta_\star}(t,s)$ on the $\boldsymbol{\beta}_\star \boldsymbol{\beta}_\star^\top$ direction and $D_\perp(t,s)$ in any of the $D-1$ remaining directions

$$D_{\beta_\star}(t,s) = \left\langle \frac{\partial}{\partial \beta(s)} a(t) w_{\beta_\star}(t) \right\rangle \ , \quad D_\perp(t,s) = \left\langle \frac{\partial}{\partial \beta_\perp(s)} a(t) w_\perp(t) \right\rangle \tag{127}$$

Similarly, $\boldsymbol{\kappa}(t,s)$ has a similar decomposition

$$\kappa_{\beta_\star}(t,s) = \langle a(t) a(s) \rangle \langle w_{\beta_\star}(t) w_{\beta_\star}(s) \rangle + \langle a(s) w_{\beta_\star}(t) \rangle \langle a(t) w_{\beta_\star}(s) \rangle$$
$$\kappa_\perp(t,s) = \langle a(t) a(s) \rangle \langle w_\perp(t) w_\perp(s) \rangle + \langle a(s) w_\perp(t) \rangle \langle a(t) w_\perp(s) \rangle \tag{128}$$

The processes have the following equations at infinite width

$$\frac{d}{dt} w_{\beta_\star}(t) = \gamma a(t)(\beta_\star - \beta(t)) \ , \quad \frac{d}{dt} a(t) = \gamma w_{\beta_\star}(t)(\beta_\star - \beta(t)) \ , \quad \frac{d}{dt} w_\perp(t) = 0 \tag{129}$$

As a consequence we note that $\langle w_\perp(t) a(s) \rangle = 0$ so that $\kappa_\perp(t,s) = \langle a(t) a(s) \rangle$. Letting $v_+(t) = \frac{1}{\sqrt{2}}(w_{\beta_\star}(t) + a(t))$ and $v_-(t) = \frac{1}{\sqrt{2}}(w_{\beta_\star}(t) + a(t))$, we find the same decoupled stochastic processes as in Appendix I.3.

$$\frac{d}{dt} v_+(t) = \gamma(\beta_\star - \beta(t)) v_+(t) \ , \quad \frac{d}{dt} v_-(t) = -\gamma(\beta_\star - \beta(t)) v_-(t) \tag{130}$$

We can use these equations to perform the necessary averages for $\kappa_{\beta_\star}$ and $D_{\beta_\star}$. Lastly, we use

$$\frac{\partial}{\partial \beta_\perp(s)} w_\perp(t) = -\gamma \Theta(t-s) a(s) \tag{131}$$

to evaluate $D_\perp(t,s)$. The observed covariances are just

$$\boldsymbol{\Sigma}_{\beta_\star} = (\gamma \mathbf{I} - \boldsymbol{D}_{\beta_\star})^{-1} \boldsymbol{\kappa}_{\beta_\star} (\gamma \mathbf{I} - \boldsymbol{D}_{\beta_\star})^{-1\top} \ , \quad \boldsymbol{\Sigma}_\perp = (\gamma \mathbf{I} - \boldsymbol{D}_\perp)^{-1} \boldsymbol{\kappa}_\perp (\gamma \mathbf{I} - \boldsymbol{D}_\perp)^{-1\top} \tag{132}$$

We note that these expressions are identical to those in Appendix J under the substitution $\beta_\star - \beta(t) \to \Delta(t)$ and $D \to P$. Thus the expected test risk is

$$\langle |\boldsymbol{\beta}(t) - \boldsymbol{\beta}_\star|^2 \rangle \sim (\beta(t) - \beta_\star)^2 + \frac{1}{N} \Sigma_{\beta_\star}(t,t) + \frac{(D-1)}{N} \Sigma_{\beta_\perp}(t,t) + \mathcal{O}(N^{-2}) \tag{133}$$

This recovers the variance we obtained in the multiple-sample whitened data case J.

## K.3 Connections to Offline Learning in Linear Model

**Remark 1** *The finite size variance of generalization error in an online learning setting with linear target function $y = \boldsymbol{\beta}^* \cdot \boldsymbol{x}$ has an identical form as the model described above. In this setting, we sample infinitely many fresh data points $\boldsymbol{x} \sim \mathcal{N}(0, \mathbf{I})$ at each step leading to the flow $\frac{d}{dt}\boldsymbol{w}_i(t) = \gamma a_i(t)\mathbb{E}_{\boldsymbol{x}}\Delta(\boldsymbol{x})\boldsymbol{x}$ and $\frac{d}{dt}a_i(t) = \gamma \boldsymbol{w}_i(t) \cdot \mathbb{E}_{\boldsymbol{x}}\Delta(\boldsymbol{x})\boldsymbol{x}$. The order parameter of interest in this setting is $\boldsymbol{\beta}(t) = \frac{1}{\gamma N}\sum_{i=1}^{N}\boldsymbol{w}_i(t)a_i(t)$. The precise correspondence between this setting and the offline setting is summarized in Table 2. We note that this argument could be extended to higher degree monomial activations as well, at the cost of tracking higher degree tensors (eg for quadratic activations $\boldsymbol{M} = \frac{1}{N}\sum_{i=1}^{N}a_i\boldsymbol{w}_i\boldsymbol{w}_i^\top \in \mathbb{R}^{D \times D}$ is sufficient).*

| Setting | Order Params. | Target | Off-target Dims. | Loss | Variance | Infinite Quantity |
|---------|---------------|--------|------------------|------|----------|-------------------|
| Offline | $\boldsymbol{\Delta} = \boldsymbol{y} - \boldsymbol{f}$ | $\boldsymbol{y}$ | $P-1$ | Train | $\mathcal{O}(\frac{P}{N})$ | $D$ |
| Online | $\boldsymbol{\beta}_\star - \boldsymbol{\beta}$ | $\boldsymbol{\beta}_\star$ | $D-1$ | Test | $\mathcal{O}(\frac{D}{N})$ | $P$ |

Table 2: Summary of the equivalence between the leading $1/N$ correction in the offline setting and the online setting for two layer linear networks. In the offline training setting, the order parameters are the errors $\boldsymbol{\Delta} = \boldsymbol{y} - \boldsymbol{f} \in \mathbb{R}^P$ while in the online case they are $\boldsymbol{\beta}_\star - \boldsymbol{\beta} \in \mathbb{R}^D$.

As in the offline case, in Fig. 4 (c) and (d) we see that the variance contribution to test loss $|\boldsymbol{\beta} - \boldsymbol{\beta}_\star|^2$ increases with input dimension $D$. We note that this perturbative effect to the loss dynamics is reminiscent of the deviations from mean field behavior studied in SGD [43, 44], though this present work concerns fluctuations driven by initialization variance rather than stochastic sampling of data. In Fig. 4 (e) we show that richer networks have lower variance at fixed $N$. Similarly, leading order theory for richer networks more accurately captures their dynamics as $D/N$ increases (Fig. 4 (f)).

# L Deep Linear Networks

For deep linear networks, the fields $h_\mu^\ell(t), g_\mu^\ell(t)$ are Gaussian and have the following self-consistent equations

$$h_\mu^\ell(t) = u_\mu^\ell(t) + \gamma \int_0^t ds \sum_\nu \left[ A_{\mu\nu}^{\ell-1}(t,s) + \Delta_\nu(s)H_{\mu\nu}^{\ell-1}(t,s) \right] g_\nu^\ell(s) \,, \; u_\mu^\ell(t) \sim \mathcal{GP}(0, \boldsymbol{H}^{\ell-1})$$

$$g_\mu^\ell(t) = r_\mu^\ell(t) + \gamma \int_0^t ds \sum_\nu \left[ B_{\mu\nu}^\ell(t,s) + \Delta_\nu(s)G_{\mu\nu}^{\ell+1}(t,s) \right] h_\nu^\ell(s) \,, \; r_\mu^\ell(t) \sim \mathcal{GP}(0, \boldsymbol{G}^{\ell+1}).$$

$$(134)$$

where $H_{\mu\nu}^\ell(t,s) = \langle h_\mu^\ell(t)h_\nu^\ell(s) \rangle$ and $G_{\mu\nu}^\ell(t,s) = \langle g_\mu^\ell(t)g_\nu^\ell(s) \rangle$ and $A_{\mu\nu}^\ell(t,s) = \left\langle \frac{\partial h_\mu^\ell(t)}{\partial r_\nu(s)} \right\rangle$ and

$B_{\mu\nu}^\ell(t,s) = \left\langle \frac{\partial h_\mu^\ell(t)}{\partial r_\nu(s)} \right\rangle$ [9]. Therefore, we express the action as a differentiable function of the order parameters by integrating over the Gaussian field distribution. For concreteness, we vectorize our fields over time and samples $\boldsymbol{h}^\ell = \text{Vec}\{h_\mu^\ell(t)\}_{\{\mu \in [P], t \in \mathbb{R}_+\}}, \boldsymbol{g}^\ell = \text{Vec}\{g_\mu^\ell(t)\}_{\{\mu \in [P], t \in \mathbb{R}_+\}}$ we consider the contribution of a single hidden layer.

$$\mathcal{Z}_\ell = \int d\hat{\boldsymbol{h}}^\ell d\hat{\boldsymbol{g}}^\ell d\boldsymbol{h}^\ell d\boldsymbol{g}^\ell \exp\left( -\frac{1}{2}\hat{\boldsymbol{h}}^\ell \boldsymbol{\Sigma}_u \hat{\boldsymbol{h}}^\ell + i\hat{\boldsymbol{h}}^\ell \cdot (\boldsymbol{h}^\ell - \boldsymbol{C}^\ell \boldsymbol{g}^\ell) - \frac{1}{2}\boldsymbol{h}^{\ell\top}\hat{\boldsymbol{H}}^\ell \boldsymbol{h}^\ell \right)$$

$$\exp\left( -\frac{1}{2}\hat{\boldsymbol{g}}^\ell \boldsymbol{\Sigma}_r^\ell \hat{\boldsymbol{g}}^\ell + i\hat{\boldsymbol{g}}^\ell \cdot (\boldsymbol{g}^\ell - \boldsymbol{D}^\ell \boldsymbol{h}^\ell) - \frac{1}{2}\boldsymbol{g}^{\ell\top}\hat{\boldsymbol{G}}^\ell \boldsymbol{g}^\ell \right)$$

where $C_{\mu\nu}^\ell(t,s) = \gamma\Theta(t-s)\left[ A_{\mu\nu}^{\ell-1}(t,s) + H_{\mu\nu}^{\ell-1}(t,s)\Delta_\nu(s) \right]$ and $D_{\mu\nu}^\ell(t,s) = \gamma\Theta(t-s)\left[ B_{\mu\nu}^\ell(t,s) + G_{\mu\nu}^{\ell+1}(t,s)\Delta_\nu(s) \right]$. Performing the joint Gaussian integrals over $(\boldsymbol{h}^\ell, \boldsymbol{g}^\ell, \hat{\boldsymbol{h}}^\ell, \hat{\boldsymbol{g}}^\ell)$ we find

$$\ln \mathcal{Z}_\ell = -\frac{1}{2}\ln\det \begin{bmatrix} -\hat{\boldsymbol{H}}^\ell & 0 & \mathbf{I} & -\boldsymbol{D}^{\ell\top} \\ 0 & -\hat{\boldsymbol{G}}^\ell & -\boldsymbol{C}^{\ell\top} & \mathbf{I} \\ \mathbf{I} & -\boldsymbol{C}^\ell & \boldsymbol{\Sigma}_u & 0 \\ -\boldsymbol{D}^\ell & \mathbf{I} & 0 & \boldsymbol{\Sigma}_r \end{bmatrix} \quad (135)$$

We can then automatically differentiate the DMFT action to get the propagator. For example, for a three layer linear network, the full DMFT action has the form

$$S = \frac{1}{2}\text{Tr}\left[\hat{\boldsymbol{H}}^1\boldsymbol{H}^1 + \hat{\boldsymbol{H}}^2\boldsymbol{H}^2 + \hat{\boldsymbol{G}}^1\boldsymbol{G}^1 + \hat{\boldsymbol{G}}^2\boldsymbol{G}^2\right] - \gamma^2\text{Tr}\boldsymbol{A}\boldsymbol{B}$$

$$-\frac{1}{2}\ln\det\begin{bmatrix} -\hat{\boldsymbol{H}}^1 & 0 & \boldsymbol{I} & -\boldsymbol{D}^{1\top} \\ 0 & -\hat{\boldsymbol{G}}^1 & -\boldsymbol{C}^{1\top} & \boldsymbol{I} \\ \boldsymbol{I} & -\boldsymbol{C}^1 & \boldsymbol{1}\boldsymbol{1}^\top & 0 \\ -\boldsymbol{D}^1 & \boldsymbol{I} & 0 & \boldsymbol{G}^2 \end{bmatrix}$$

$$-\frac{1}{2}\ln\det\begin{bmatrix} -\hat{\boldsymbol{H}}^2 & 0 & \boldsymbol{I} & -\boldsymbol{D}^{2\top} \\ 0 & -\hat{\boldsymbol{G}}^2 & -\boldsymbol{C}^{2\top} & \boldsymbol{I} \\ \boldsymbol{I} & -\boldsymbol{C}^2 & \boldsymbol{H}^1 & 0 \\ -\boldsymbol{D}^2 & \boldsymbol{I} & 0 & \boldsymbol{1}\boldsymbol{1}^\top \end{bmatrix} \tag{136}$$

where $\boldsymbol{C}^1 = \gamma\boldsymbol{\Theta}_\Delta$ and $\boldsymbol{C}^2 = \gamma\boldsymbol{\Theta}_\Delta \odot \boldsymbol{H}^1 + \gamma\boldsymbol{A}$ and $\boldsymbol{D}^1 = \gamma\boldsymbol{\Theta}_\Delta \odot \boldsymbol{G}^2 + \gamma\boldsymbol{B}$ and $\boldsymbol{D}^2 = \gamma\boldsymbol{\Theta}_\Delta$. This above example can be extended to deeper networks. The total size of the block matrices which we compute determinants over is $4PT \times 4PT$ for a dataset of size $P$ trained for $T$ steps.

## M   Discrete Time Dynamics and Edge of Stability Effects

Large step size effects can induce qualitatively different dynamics in neural network training. For instance, if the step size exceeds that required for linear stability with the initial kernel, the kernel can decrease in order to stabilize the dynamics [57]. Alternatively, during training the kernel may exhibit a "progressive sharpening" phase where its top eigenvalue grows before reaching a stability bound set by the learning rate [19]. It is therefore well motivated to study how dynamics in this regime alter finite size effects in neural networks. We will first solve a special model which was considered in prior work [57]: a two layer linear network trained on a single training point. We will then provide the full DMFT equations for the discrete time case and provide an outline for how one could obtain finite size effects in that picture.

### M.1   Two Layer Linear Equations

In a two layer linear network, the DMFT equations are

$$h(t+1) = h(t) + \eta\gamma\Delta(t)z(t)\,,\ z(t+1) = z(t) + \eta\gamma\Delta(t)h(t)$$

$$f(t) = \frac{1}{\gamma}\langle z(t)h(t)\rangle \tag{137}$$

The NTK has the form $K(t) = \langle h(t)^2 + z(t)^2\rangle$. We can easily show that the kernel and error have coupled dynamics

$$f(t+1) = f(t) + \eta\langle h(t)^2 + z(t)^2\rangle\Delta(t) + \eta^2\gamma\Delta(t)^2\langle h(t)z(t)\rangle$$
$$= f(t) + \eta K(t)\Delta(t) + \eta^2\gamma^2\Delta(t)^2 f(t) \tag{138}$$
$$K(t+1) = K(t) + 4\eta\gamma\Delta(t)\langle h(t)z(t)\rangle + \eta^2\gamma^2\Delta(t)^2\langle h(t)^2 + z(t)^2\rangle$$
$$= K(t) + 4\eta\gamma^2\Delta(t)f(t) + \eta^2\gamma^2\Delta(t)^2 K(t) \tag{139}$$

These equations define the infinite width evolution of $\Delta(t)$ and $K(t)$. Already at this level of analysis, we can reason about the evolution of $K(t)$. In the small $\eta$ limit, we could disregard terms of order $\mathcal{O}(\eta^2)$ and arrive at the following gradient flow approximation for $K(t) \sim 2\sqrt{1 + \gamma^2 f(t)^2}$ [9]. This evolution will not reach the edge of stability provided that $\eta < \frac{1}{\sqrt{1+\gamma^2 y^2}}$. For large $\gamma$ and $y = 1$, this leads to the constraint $\eta\gamma < 1$. However, if $\eta$ exceeds this bound, the gradient flow approximation is no longer reasonable and the system reaches an edge of stability effect as shown in Figure 6.

To calculate the finite size effects, we need to compute $\kappa$ and $D(t,s) = \frac{\partial}{\partial\Delta(s)}\langle h(t)^2 + z(t)^2\rangle$. To evaluate these quantities we utilize the same change of variables employed in Appendix I.3. In discrete time, these decoupled equations are

$$v_+(t+1) = v_+(t) + \eta\gamma\Delta(t)v_+(t)\,,\ v_-(t+1) = v_-(t) - \eta\gamma\Delta(t)v_-(t). \tag{140}$$

Given $\Delta(t)$, these can be expressed as linear systems of equations. Now, we can easily compute the uncoupled kernel variance

$$\kappa(t,s) = 2\langle h(t)h(s)\rangle^2 + 2\langle z(t)z(s)\rangle^2 + 2\langle h(t)z(s)\rangle^2 + 2\langle z(t)h(s)\rangle^2$$
$$= \langle v_+(t)v_+(s) + v_-(t)v_-(s)\rangle^2 + \langle v_+(t)v_+(s) - v_-(t)v_-(s)\rangle^2. \qquad (141)$$

Similarly, we can calculate $D(t,s)$ by using the fact $\langle h(t)^2 + z(t)^2\rangle = \langle v_+(t)^2 + v_-(t)^2\rangle$

$$D(t,s) = 2\left\langle v_+(t)\frac{\partial v_+(t)}{\partial\Delta(s)}\right\rangle + 2\left\langle v_-(t)\frac{\partial v_-(t)}{\partial\Delta(s)}\right\rangle$$
$$\frac{\partial v_+(t)}{\partial\Delta(s)} = \gamma\Theta(t-s)v_+(s) + \sum_{t'<t}\Delta(t')\frac{\partial v_+(t')}{\partial\Delta(s)}$$
$$\frac{\partial v_-(t)}{\partial\Delta(s)} = -\gamma\Theta(t-s)v_-(s) - \sum_{t'<t}\Delta(t')\frac{\partial v_-(t')}{\partial\Delta(s)} \qquad (142)$$

These can be directly solved as a linear system of equations.

## N  Computing Details

Experiments for Figures 3, 6 and 2 were conducted on a Google Colab GPU with JAX. Experiments for Figures 5, A.3, 7 were performed on a NVIDIA SMX4-A100-80GB GPU. The total compute required for all Figures in the paper took around 4 hours. Jupyter Notebooks to reproduce plots can be found at `https://github.com/Pehlevan-Group/dmft_fluctuations`.

