# OpenReview forum: "Dynamics of Finite Width Kernel and Prediction Fluctuations in Mean Field Neural Networks"
_NeurIPS.cc/2023/Conference — NeurIPS 2023 spotlight_

### Official Review · Reviewer_fkFE · 2023-07-05

**Soundness:** 3 good
**Presentation:** 3 good
**Contribution:** 3 good
**Rating:** 7
**Confidence:** 3

**Summary:**

The authors study a finite width correct to the Dynamical Mean Field Theory (DMFT) of finite depth neural networks in the feature learning regime. While I will be the first to admit that I am not an expert on the DMFT calculations, the authors did produce very convincing simulations capturing interesting properties of finite size networks. In particular, the edge of stability behaviour at large learning rates seems to be modelled by the finite width correction.

I hope the authors can help me clarify some questions regarding the implementation of the solver, after which I would be happy to raise my score to accept.

**Strengths:**

The authors develop a strong theory of finite width neural networks capable of making accurate predictions.

**Weaknesses:**

N/A

**Questions:**

I would like the authors to provide some intuitions about how the DMFT equations are being solved numerically. I understand this may be the contents of a previous paper (Bordelon and Pehlevan, 2022), but I am having trouble understanding this part.

1. Based on my understanding, all the elements of $q$ are processes index by time $t$. So what does it mean to compute elements of $q$ or $\Sigma$? Are the authors time discretizing these processes?
2. How should I understand the role of the propagator $\Sigma$, and how the authors solve for this object first independently of the elements of $q$?
3. How should I understand the saddle point solver in Algorithm 2? Here I am just asking for what is happening within the algorithm, as I find reading the pseudocode quite difficult to understand.

I would be happy to continue this discussion during the rebuttal period, and once again I would raise my score once I find these questions addressed.

---

> ### Author Rebuttal · Authors · 2023-08-08
>
> We thank their reviewer for their support and good questions. We hope to make our methods more understandable and self-contained in the paper. Below we provide some more explanations about how we solve our self-consistent equations.
>
> ### Response to Questions
>
> 1. This is a great question that we will address more in the work (Appendix E). First, the $\mathbf q$ vector represents the collection of the order parameters (kernels $\Phi^\ell_{\mu\nu}(t,s)$, predictions $f_\mu(t)$, response functions $A^\ell_{\mu\nu}(t,s)$, etc) at *all possible times through training*. Theoretically, for gradient flow this is an uncountably infinite set. For discrete time gradient descent, this is a countable set. In a practical numerical solution, we need to discretize time to compute it. For subtle theoretical reasons, the path integral is also initially derived using a discretization of all the dynamics in time (with the Ito convention). After integration over the initial weights, we can then take a continuous time limit to arrive at the action $S$ provided in Appendix D. Concretely we will add the following text to Appendix E
>     *For nonlinear deep networks, we build on the Monte-carlo approach developed by Bordelon & Pehlevan 2022 which computes the saddle point equations for all order parameters $q_\infty$. For a practical numerical algorithm, we discretize the time steps so that we store finite matrices (such as $\Phi^\ell_{\mu\nu}(t,s)$) with layer, sample, and time indices.  Our approach allows us to estimate the entries of the action's Hessian $\nabla^2 S$ and ultimately invert it to obtain the propagator $\Sigma = \left[ - \nabla^2 S(q_\infty) \right]^{-1}$. To do this, we need to use sampling to estimate the fourth feature moments $\kappa$ and the sensitivity blocks $D$. We evaluate these averages over the stochastic process defined by $q_\infty$.*
> For a detailed set of steps to compute $q_\infty$ and $\Sigma$, see the third bullet point below.
> 3. The propagator $\Sigma$ can be thought of as the covariance of the order parameters near infinite width. Its name in physics derives from the fact that fluctuations in the order parameters such as the kernel $\Phi^\ell$ at time $t$ can propagate noise to other order parameters (such as $\Phi^{\ell+1}$) at time $t' > t$. When we say we compute $q_\infty$ it means that we are solving the saddle point equations, which can be solved using the methods of Bordelon & Pehlevan 2022. **To compute $\Sigma$, we need to have already first solved for $q_\infty$**. We evaluate the second derivative of $S$ at this value of the order parameters $q_\infty$. This involves using similar methods (either monte carlo for nonlinear networks or closed form expressions for linear networks.) Again, in practice, we discretize time in a numerical algorithm to evaluate these expressions. (See below for more detail.)
> 4. The Algorithm pseudocode can be summarized by the following high level instructions
> * Step 1: First solve the infinite width DMFT equations for $q_\infty$ which include the prediction error dynamics $\Delta_\mu(t)$, the feature kernels $\Phi^\ell_{\mu\nu}(t,s)$, gradient kernels $G^\ell_{\mu\nu}(t,s)$. This step corresponds to algorithm in Bordelon & Pehlevan 22 and defines the dynamics one would expect at infinite width.
> * Step 2: Compute the entries of the Hessian of $S$ evaluated at the $q_\infty$ computed in the first step. Some of these entries look like fourth cumulants of features like $\kappa = \left< \phi(h)^4 \right> - \left< \phi(h)^2 \right>^2$ and some of them measure sensitivity of one order parameter to a perturbation in another order parameter $D^{\Phi^\ell} = \frac{\partial}{\partial \Phi^{\ell-1}} \left< \phi(h^\ell)^2 \right>$. The averages $\left< \right>$ used to calculate $\kappa$ and $D^{\Phi^\ell}$ should be performed over the infinite width stochastic processes for preactivations $h^\ell$ which are defined in equation (19).
> * Step 3: After populating the entries of the block matrix for the Hesssian $\nabla^2 S$, we then calculate the propagator $\Sigma$ with a matrix inversion. Since we discretized time, this is a finite dimensional matrix.
>
> The above text will also be added to the Appendix of our work.
>
> 4. Now we will give a detailed set of intuitions about how the infinite width limit for $q_\infty$ is solved (step 1 above). This corresponds to the algorithm of Bordelon & Pehlevan 2022 to solve the saddle point equations $\frac{\partial}{\partial q} S(q)|_{q_\infty} = 0$.
> * Step 1: Start with a guess for the kernels $\Phi^\ell_{\mu\nu}(t,s), G^\ell_{\mu\nu}(t,s)$ and for the predictions through time $f_\mu(t)$. We usually use the lazy limit as an initial guess.
> * Step 2: Sample gaussian sources $u^\ell_\mu(t) \sim \mathcal{GP}(0,\Phi^{\ell-1})$ and $r^\ell_\mu(t) \sim \mathcal{GP}(0,G^{\ell+1})$ based on the current estimated covariances $\Phi^\ell$ and $G^\ell$ respectively.
> * Step 3: For each sample, solve integral equations for $h^\ell_\mu(t)$ and $z^\ell_\mu(t)$.
> These will be samples from the single site distribution for $h^\ell, z^\ell$. In a discretization, the integrals will be replaced with sums.
> * Step 4: Average over the Monte-carlo samples to produce a new estimate of the kernels, for instance $\Phi^\ell_{\mu\nu}(t,s) = \left< \phi(h^\ell_\mu(t)) \phi(h^\ell_\nu(s)) \right>$. A similar procedure is performed for $G^\ell$ and the response functions $A,B$.
> * Step 5: Compute the NTK estimate $K(t) = \sum_\ell G^{\ell+1}(t,t) \Phi^\ell(t,t)$ and then integrate prediction dynamics from the dynamics of the NTK $\frac{d}{dt} f_\mu(t) = \sum_\nu K_{\mu\nu}(t) \Delta_\nu(t)$.
> * Repeat steps 2-5 until the order parameters converge.
>
> Please let us know if these explanations helped clarify the numerical methods and if there are any remaining questions or concerns.
>
> As we mentioned in the global response, we are also going to add a more self-contained derivation of the DMFT action (similar to Bordelon & Pehlevan 2022).

---

> > ### Comment · Reviewer_fkFE · 2023-08-16
> > **Response**
> >
> > I apologize for the delayed response. I have been traveling and wrestling with a heavy review load this year.
> >
> > I believe my concerns have been addressed, and I will raise my score to accept.

---

### Official Review · Reviewer_2Zma · 2023-07-05

**Soundness:** 3 good
**Presentation:** 3 good
**Contribution:** 4 excellent
**Rating:** 7
**Confidence:** 3

**Summary:**

Building on past work which set up a DMFT (Dynamical Mean Field Theory) for fully connected networks in the infinite width limit (where the width of each layer tends to infinity), this paper reasons about the *fluctuations* around the infinite width limit. This is important because for finite sized neural networks, these fluctuations are large enough that they are an important part of the network training dynamics. The description of the DMFT is theoretically complicated and cannot be solved exactly, but they enable simulations which confirm that the DMFT captures the behaviour of finite sized real networks quite accurately. As an application, this DMFT theory is used to understand bias, training rates, and variance in realistic tasks and the special case of 2-layer networks (where the theory is quite a bit simpler) is investigated in more detail. This theory has the potential to open the door to many potential future uses that explain how neural networks learn.

**Strengths:**

This paper lays out a framework for theoretical understanding of deep neural networks that incorporates the effect of finite width. This is a  problem that has received a lot of attention since the original NTK/infinite width limits came out, and as far as I know, the approach here is novel and powerful. The paper is well written for the most part although familiarity with the infinite width DMFTs is assumed.

Overall, the fact that the theory and the simulations agree very well is quite impressive, and I think the ideas in the paper are quite ambitious because they can be used for almost any kind of question one might have about the theoretical evolution of the DNN. This paper has the potential to be the basis for future work which uses the theory developed here to investigate questions about how DNNs learn.

**Weaknesses:**

The main weakness of the paper is that its a bit spread thin at times: both the theory and a few different applications are covered, but it seems like the authors were trying to make it all fit and I would have liked more detail in a few spots. This is largely due to the page limit of the submission. I personally would have found it to be a stronger paper if a single really clear example was presented in a lot of detail. (Although again, I completely understand that this is largely pressure from the conference format to try and do a lot of stuff)

The other main "weakness" of the paper (which is strictly speaking a limitation of the audience of the paper) is that to understand it, you need to be familiar with the previous DMFT on which this paper is built. The authors include a very short section called "Review of Dynamical Mean Field Theory" citing [9],[46] as a review, but this section is extremely sparse for actually understanding what is going on. I essentially had to read [9] in its entirety first to understand what was going on in this paper. (Also the reference [46] could not be found since only authors and title are given...where would one find this reference?) In my view, this weakness could be mitigated by just being more honest with the reader up front about this...for example [9],[46] should be cited at point 1 in the list of contributions to make it more clear the dependence and what is/isn't actually explained in this paper.

Another (related) "weakness" is that the paper relies quite heavily on physics technology and jargon to reach its conclusions. The fact that the results are so heavily entrenched in physics jargon like "order parameters" or "propagator" makes this paper less likely to have a broad impact on the deep learning community. The authors would add a lot of value to the work by attempting to make a "translation guide" to help people who don't have the same physics background understand what is going on in more detail.

**Questions:**

* Suggestion: Eqn (3) seems like a very important main result: a bit more discussion on the next order term and why its actually size 1/N (even though it starts with an N) would be helpful.
* I am wondering if you had any numerical simulations where you checked the rate of covergence between the theory here and simulations (which should presumably be like 1/N^2?). Something like Figure 2a but comparing to the fluctuation predictions rather than to $q_\infty$.
* One high-level question I had: It seems that if the learning rate is fixed and not scaled, the the fluctuations and the effect of the learning rate are both on the same scale $1/\sqrt{N}$. 1. Is that correct? and 2. Does your theory work to analyze what's going on in that case?

Here is a list of other minor errors/suggestions I found while reading:]
* Eqn (2): use definition equal with three lines to be consistent with definitions later on
* Line 179 vs 185: Is there a difference between $K$ and $K_\infty$? If so what is it?
* Line 188: What does the subscript 0 in "$Cov_0$" mean here?
*  Section 6.1: I was able to more-or-less piece this together, but I think it would be a lot more understandable if you gave explit definitions for $K(t)$ and $K_\ast(t)$
* Line 232-235: I think it would be a lot clearer to write out the definitions here of the two new $\Delta$'s in terms of $\Delta_\mu$.
* References [15],[21],[33], [46], [53] has only author/title but not where published/where to find.
* Reference [22] missing a title?
* I would also check the arxiv only referenes e.g. [11],[13],[20],[27],[34],[57],[63] to make sure there isnt a conference or journal version that is now published.

**Limitations:**

A potential criticism is the physics level of rigor used in Appendix C which is used to establish the main results. The manipulations carried out in the proof of Appendix C certainly seem plausible, and I believe the community as a whole is ok with this level of rigour, but the authors could be a bit more clear about what they mean by "proof" in the main paper. It is not a mathematically rigour proof (which would involve all sorts of techincal assumptions), but rather a physics-type statement that holds assuming the usual expansions can be carried out without obstructions. To reiterate: I think the actual work is fine, but they could be a bit more honest about how it is "proven" and the level of rigour in the main paper.

---

> ### Author Rebuttal · Authors · 2023-08-08
>
>
> We thank the reviewer for appreciating the strengths of our approach and its applicability to wide DNN dynamics. Below we provide some responses to the weaknesses, questions and limitations.
>
> **Responses to Weaknesses**
>
> We agree that the paper is spread a bit thin at times. Based on the detailed comments and questions of this reviewer, we have added more detailed definitions of the terms which appear in the main text equations. Given an additional page for a final draft, we will expand our comments on the setup of the calculation, the statement of the main results and their implications.
>
> Based on the comments of all reviewers, we will add a more self-contained derivation of DMFT in the Appendix which will introduce the main concepts and derive the action $S$ which plays such a central role in this paper. We also will add a table, as the first reviewer suggested which translates the physics jargon for the objects into more ML theory friendly language.
>
> We agree with the reviewer that we do not provide formally rigorous proofs of our results, but derive them at a physical level of rigor. We will add a sentence in the disussion acknowledging this limitation and will leave open for future works to provide rigorous proofs of these dynamical expansions.
>
> **Responses to Questions**
>
> 1. This is a good question/suggestion. Indeed the raw covariance of the order parameters is size $\mathcal{O}(1/N)$. We define the propagator $\Sigma \sim N \text{Cov}(q) \sim \mathcal{O}(1)$ entries to be $N$ times this covariance so that it behaves as an $\mathcal{O}(1)$ quantity. Thus equation 7 can be solved for $\Sigma^{\Delta} \sim \mathcal{O}(1)$ once for all possible $N$. When we want to compare to simulations of a finite width $N$ network, we could multiply the empirically observed covariance by $N$ and compare to $\Sigma$. As the reviewer points out, the covariance $N \text{Cov}(q) \sim \Sigma + \mathcal{O}(N^{-1})$ is correct up to a subleading term. This is established in Appendix 3.1 and 3.3, specifically equations 15 and 17 (disregard the typo in the sentence above equation 17). We will comment on this near equation 3.
> 2. When we submitted the paper, we did not have simulations showing that the covariance predicted $\frac{1}{N}\Sigma$ is correct up to $\mathcal{O}(N^{-2})$. To accurately measure the deviations between the order parameter covariance and the theoretically predicted propagator $\Sigma$, we add a simulation in the attached PDF which shows that this rate is accurate. Estimating this error rate requires a very large number of neural networks (many more than to estimate $\text{Cov}(q)$) so we focus on the variance of ReLU feature and gradient kernels at initialization, where we can exactly compute $q_\infty$ and $\Sigma$.
> 3.  On the learning rate scaling with $N$ question, we have a few comments. First, if the reviewer is asking about the learning rate in the gradient flow $\frac{d}{dt}\theta = N \gamma^2 \nabla \mathcal{L}$, if the factor of $N$ is removed it will take a time $t \sim \mathcal{O}(N)$ to make progress on the training loss, but in gradient flow this is just a rescaling of the time axis in all our plots (so our theory still applies). What happens if other parameters (such as feature learning rate $\gamma$ or discrete time step size $\eta$) depend on $N$? First, what if $\gamma$ is depenent on width $N$? In the classic NTK parameterization where $\gamma = \frac{1}{\sqrt N}$, we can still compute a predicted dynamics for the kernels and fluctuations. First, each finite width $N$ network has a corresponding infinite width (feature learning) network with order parameters $q_{\infty}(\gamma = \frac{1}{\sqrt N})$. Second, the propagator can be computed as $\Sigma(\gamma= \frac{1}{\sqrt N})$, in both cases evaluating the dynamics at feature learning velocity $\gamma$ which depends on the width. The predicted covariance would be $\text{Cov}(q) \sim \frac{1}{N} \Sigma(\gamma =\frac{1}{\sqrt N})$. This makes understanding the effect of width possible but more complicated in NTK parameterization since decreasing $N$ has two effects (increase feature learning and fluctuation variance). In mean field parameterization/ $\mu P$, the feature learning parameter $\gamma$ is fixed with width so that the same $q_{\infty}(\gamma)$ and $\Sigma(\gamma)$ can be used to estimate finite width effects for different widths $N$ simply by multiplying by $1/N$. What if the raw learning rate $\eta$ in discrete time is scaled differently with $N$? This will also change the dynamics, but it will lead to a badly behaved $q_\infty$. For instance, if $\eta$ is rescaled by $1/\sqrt{N}$, the neural network will not fit the data in finite time at infinite width and if $\eta$ is multiplied by a higher power of width $N$, the dynamics will become unstable in discrete time. We add a comment on this in section 4.
>
> **Minor Comments**
>
> 1. We will be sure to use $\equiv$ when we define new terms (such as in Eq 2).
> 2. Yes, there is a difference between $K$ and $K_\infty$. The $K$ should be thought of as the random finite width NTK while $K_\infty$ is the infinite width kernel. We will add a sentence clarifying this in this section.
> 3. In this section by $\text{Cov}_0$, we meant the leading order covariance (neglecting $1/N^2$ and smaller terms). We will remove this notation and simply explain that we are computing the asymptotic covariance.
> 4. We will add an explicit definition of $K$ and $K_\star$ which represent the train-train NTK and the train-test NTK respectively.
> 5. We will explicitly define $\Delta_y$ and $\Delta_{\perp}$. In words, $\Delta_y$ is the projection of the vector $\mathbf \Delta$ on the label direction $\mathbf y$ and $\Delta_{\perp}$ is the projection on orthogonal directions.
> 6. We thank the reviewer for catching the issues with the references we will go through them and add the most recent (journal versions) citations for the papers.

---

### Official Review · Reviewer_VrD5 · 2023-07-10

**Soundness:** 3 good
**Presentation:** 3 good
**Contribution:** 4 excellent
**Rating:** 7
**Confidence:** 3

**Summary:**

The paper addresses the problem of analytical description of the rich (feature learning) dynamics of neural networks. To achieve this, the authors use previously introduced dynamical mean field theory (DMFT), which identifies several key characteristics of the problem - order parameters - defines their probability distribution throughout the training using a path integral representation. The paper considers $\mu P$ parametrization of the network, which is known to display feature learning behavior at infinite width (in contrast to popular NTK and Standard parametrizations) with feature learning strength controlled by parameter $\gamma$.

In the paper, the authors focus on leading width $O(N^{-1})$ corrections to the infinite width limit. Technically, this is achieved by taking into account quadratic (Gaussian) fluctuation around saddle point $\mathbf{q}_\infty$ of DMFT action $S(\mathbf{q})$. After deriving general equations governing $O(N^{-1})$ corrections, the authors consider a number of simplified scenarios where these equations can be solved analytically. Also, the authors validate qualitative conclusions of their theory in the non-synthetic experiments with CNN trained on CIFAR10.

**Strengths:**

The approach chosen by the authors - identifying saddle point $\mathbf{q}_\infty$ of the system's action $S(\mathbf{q})$ and investigating it together with Gaussian fluctuation around $\mathbf{q}_\infty$ - is a workhorse of various branches of theoretical physics where behavior of a complex system needs to be analyzed. In physics, this approach not only provides a very sizable portion of available analytical results but, for example, also provides SOTA results for numerical modeling of realistic strongly-correlated materials [Kotliar 2004](https://pubs.aip.org/physicstoday/article-abstract/57/3/53/755526/Strongly-Correlated-Materials-Insights-From?redirectedFrom=fulltext), [Vollhardt 2019](https://arxiv.org/abs/1910.12650). Thus, realizing this general strategy is one of the fundamental directions within deep learning theory. Also, as noted by the authors, the DMFT approach is non-perturbative w.r.t. feature learning strength  - a feature that is mostly absent in other approaches to NN dynamics away from kernel regime.

**Weaknesses:**

* In many cases, solving DMFT equations analytically seems to be intractable. This significantly renders the main purpose of the proposed theory - obtaining analytical insights into the network dynamics.
* The DMFT equations are bulky, which could make working with them quite exhaustive.
* I believe it is hard to understand main DMFT ingredients - order parameters $\mathbf{q}$ and their action $S(\mathbf{q})$ - from the current paper alone. Most probably, a careful reading of the original DMFT paper [9] is required to understand this paper. However, this is not the author's fault but rather a consequence of the chosen approach.

**Questions:**

* What dataset size $P$ were used for main CIFAR10 experiment of fig. 10? Due to mentioned $O(P^4T^4)$ memory requirements of storing the propagators, it seems to be impossible to work with the full ($P=50000$) CIFAR10 size. Also, what is the computational complexity of solving DMFT equations? Both for saddle point $\mathbf{q}_\infty$ and $O(N^{-1})$ corrections if they are different.
* While in most of the presented plots DMFT accurately describes the experiments, for the big sample sizes (fig. 3) there is a significant disagreement between theory and experiment. Do you think this a fundamental limitation DMFT on the level of leading order correction (ignoring $O(N^{-2})$ and beyond)? Or maybe it is because in the experiment you considered whitened data, whereas in the more realistic scenarios, the data typically has a low effective dimension (e.g. measured by the decay of data covariance eigenvalues)?

**Limitations:**

The authors discuss the limitations of their current approach. The limitations mainly come from 1) the inability to numerically solve DMFT equations for large-scale problems (e.g. due to $O(P^4 T^4)$ propagator size) and  2) the need to consider higher order expansion around the saddle point $\mathbf{q}_\infty$.

---

> ### Author Rebuttal · Authors · 2023-08-08
>
> We thank the reviewer for the careful reading and supportive comments. Below we address the weaknesses pointed out and attempt to answer the reviewer's questions.
>
> **Response to Weaknesses**
>
> The reviewer is correct to point out that the DMFT equations are difficult to solve numerically and that the equations are bulky in the general case. We acknowledge these issues in the limitations section and will stress them further in the updated draft. Despite their complexity, we think it is still useful to derive the equations in full generality to show that, in principle, the finite size corrections to DMFT can be computed with similar methods to compute the infinite limit.
>
> That being said, we think that a good deal of intuition can be gained by studying the special cases that are tractable numerically to analyze how various aspects of the problem (feature learning, sample size, depth etc) alter the dynamics and fluctuations.
>
> Since some of the reviewers point out that we use physics jargon without comparison to ML terminology to describe some of the objects that appear in our study, we will add a table (see attached PDF in global response) which compares the terms (order parameter, propagator, action etc) which show up in the calculation to terms more familiar to a ML audience (concentrating variable, asymptotic covariance, log density, etc).
>
> We will also add a short self-contained primer and derivation of the DMFT action in the Appendix so that readers do not need to refer to [9] to understand the present paper.
>
> **Response to Questions**
> 1. We apologize for not being clear about this section. The CIFAR-10 plots are purely empirical. In these plots, we do not attempt solving the DMFT equations or the propagator for this setting as both timesteps and samples are prohibitively large. The purpose of this section was to illustrate that the qualitative findings from the simpler analytically tractable models carry over in more realistic cases, specifically the accumulation of finite size effects over training and that the finite size effects depend on a low degree polynomial in $1/N$. We will be sure to clarify this in the draft.
> 2. This is a great question. We are mainly showing in this toy example (which has a very simplified whitened data model) that the leading order corrections can be accurate for some problem settings ($P<N$ in this problem) but can underestimate finite size effects in other settings ($P>N$). We suspect that in this example, higher order corrections (like $P^2/N^2, ...$ etc) may be necessary to accurately capture it. We agree with the reviewer's intuition about the correlations in natural data could be reducing the scale of finite size noise in more realistic settings, making finite size networks closer to the infinite width limit. For instance, if the data matrix were low rank with rank $P_{eff} < P$ then the finite width effect in this two-layer, linear network example is actually $P_{eff} / N$. We will add a proof of this and discuss it in the main text. We leave open for future works how the realistic structure of natural data alters the scale of finite width corrections.

---

### Author Rebuttal · Authors · 2023-08-08

We thank all of the reviewers for their detailed reading and comments. We appreciate the general support for this paper and the comments on the paper's strengths and weaknesses. Many concerns were shared among reviewers which has caused us to make the following updates to the paper
1. We spend more space defining all of the mathematical terms which appear in our paper and relate the physics terminology with more standard machine learning terminology. A table summarizing the map between our terminology and more standard terms is provided in the PDF. This will hopefully reduce the obscurity of our writing.
2. We provided a new experiment (for ReLU kernels at initialization) that the asymptotic covariance $\frac{1}{N} \Sigma$ predicted by our theory is accurate up to order $1/N^2$ (see attached PDF). This provides additional support for our approach, as we can characterize the error of our theoretical covariance predictions (though empirical estimates of the covariance error require simulating a very large number of networks).
3. We are now spending more space to define each of the terms (like $K_\infty, K, K_\star, f, f_\star, \Delta_y, \Delta_{\perp}$, etc) which appear in our equations. This will hopefully make the paper easier to parse and allow the reader to more easily interpret our results.
4. We will provide a self-contained derivation of the DMFT action $S$ in the Appendix so that we do not force the reviewer to read [9] Bordelon & Pehlevan 2022. We will also show how one can use $S$ to find the saddle point equations for $q_\infty$.
5. We expand in the Appendix our section which explains how to numerically solve the self-consistent equations for the saddle point $q_\infty$ and the propagator $\Sigma$ (see response to reviewer fkFE).
6. We will acknowledge in the limitations section that our paper operates at the level of rigor of a physics calculation rather than a fully rigorous proof which would need several additional assumptions to make the expansion properly defined.
7. We will fix all of the issues with the citations to make sure they are up to date and contain the appropriate journal and update paper citations to Arxiv preprints which are now published.
8. We will clarify that the CIFAR-10 experiments are purely experimental to see whether the dynamics in a more realistic setting has qualitatively similar dynamics to our solveable examples.

Overall, we aim to make the paper more understandable and readable. With the additional page, we will be able to expand the writing and exposition in the main text.

---

### Decision · Program_Chairs · 2023-09-21

**Decision:**

Accept (spotlight)

**Comment:**

This paper develops a new DMFT (Dynamical Mean Field Theory) theory that admits a finite width approximation around the infinite width limit while the existing work mainly focused on the infinite width limit. This enables us to describe the behavior of the network precisely such as bias, variance, and training rates. Overall, this paper provides a solid theoretical contribution on the DMFT analysis and the theory yields a impressively precise estimate of the actual behavior. All the reviewers are positive on this paper. I also think that this is a good paper. Hence I am happy to recommend this paper to accept.